# Functional symmetrization of neuromotor modules during locomotor development in human infants

Jiayin Lin [1], Sophia C. W. Ha[2], Janet H. Zhang-Lea[3], Zoe Y. S. Chan[4], Kaiduo Fang [5], Xiaoyu Guo[6], Borong He [1], Kelvin Y. S. Lau[1], Rosa H. M. Chan[6], Roy T. H. Cheung[7], Chao-Ying Chen[8] & Vincent C. K. Cheung [1,9] ✉

During human locomotor development, neonatal neuromotor control modules known as muscle synergies are continuously modified into their mature forms to enable independent walking. How the early muscle synergies, developing neuromusculoskeletal systems, and sensorimotor plasticity interact to regulate this process remains unknown. To address this, we investigated concurrent changes in muscle synergies, kinematic synergies, and lower-limb biomechanical properties across 4 stages of supported and independent walking through longitudinal bilateral multi-muscle recording, kinematic tracking, and personalized neuro-musculoskeletal modelling in 11 human infants, while incorporating additional data from adults and elders for a whole-lifespan analysis. Our results argue that the initially bilaterally asymmetrical muscle synergies and limb biomechanical properties co-evolve to ultimately result in symmetrical kinematic synergies that may stabilize gait. Functional symmetrization of neuromotor modules may be a reflection of the co-development of muscle synergies, their associated kinematic functions, and limb biomechanical properties for achieving gait stability and control efficiency throughout the lifespan.

The theory of motor modularity has been used in many recent studies to understand how the mammalian central nervous system (CNS) coordinates the activation of numerous muscles to perform diverse motor tasks[1–5]. This theory suggests that instead of controlling each muscle individually, the CNS modulates a limited number of motor modules that activate multiple muscles together as discrete units, thus enhancing the efficiency of control. The motor module, commonly identified as a time-invariant muscle synergy (MS), represents a constant muscle co-activation pattern that is temporally scaled by its activation coefficient (AC). In humans, adult locomotion is generated by the combination of at least four MSs for each lower limb[6–8]. By what principle, and through what neural mechanisms, the MSs are developmentally assembled into their mature forms in the adult has remained an important question in motor neuroscience.

For sure, the locomotor MSs undergo long-term changes throughout the human lifespan. As neonates develop into toddlers, the MSs for neonatal stepping fractionate into sparser MSs composed of fewer muscles to better fulfill the biomechanical demands of independent walking[9–11]. As toddlers grow into adults, the MSs continue to be refined, but with the dominant contributing muscles in each MS preserved in this process[12]. As adults age, the extent of MS changes may become smaller, but the AC of the MSs display greater temporal variability both within and across individuals[13]. While these age-related MS changes are well described, the functional significance of such changes is less understood. Presumably, to meet task-specific biomechanical demands, the CNS selects and adjusts which MSs to activate based on their biomechanical functions[14–18]. This perspective implies that modification in MSs across the life stages may stem from

[1]School of Biomedical Sciences and Gerald Choa Neuroscience Institute, The Chinese University of Hong Kong, Hong Kong, China. [2]School of Health and Sports Science, Regent College London, London, UK. [3]Department of Human Physiology, University of Oregon, Eugene, OR, USA. [4]Faculty of Kinesiology, University of Calgary, Calgary, AB, Canada. [5]Department of Electrical and Electronic Engineering, The Hong Kong Polytechnic University, Hong Kong, China. [6]Department of Electrical Engineering, and State Key Laboratory of Terahertz and Millimeter Waves, City University of Hong Kong, Hong Kong, China. [7]School of Health Sciences, Western Sydney Universities, Sydney, NSW, Australia. [8]School of Physical Therapy and Graduate Institute of Rehabilitation Science, Chang Gung University, Taoyuan City, Taiwan. [9]Joint Laboratory of Bioresources and Molecular Research of Common Diseases, The Chinese University of Hong Kong and Kunming Institute of Zoology of the Chinese Academy of Sciences, Hong Kong, China. ✉e-mail: vckc@cuhk.edu.hk

age-related changes of the task or limb biomechanical properties and/or shifts in the MSs' functional roles.

More specifically, one possibility is that during development, as the limb biomechanical properties such as limb length and muscle strength change, the MSs are modified to achieve the required biomechanical functions critical for task performance. Indeed, MSs appear to be structured to perform constant functions. Activations of particular MSs are associated with specific gait phases of human locomotion[14,15], and with specific kinetic or kinematic features during feline postural tasks[16,17] and frog kicking[18]. If we consider kinematic synergy (KS)—the coordinated variation of joint angles—as a functional representation of MS[19–21], the functional stability of MSs has been demonstrated in adult grasping and walking of humans[22,23], which showed a robust correspondence between MSs and stable KSs.

Another possibility is that the developmentally modified set of MS executes different sets of biomechanical functions over time. Such changes in functions may reflect how the MSs are gradually fine-tuned, as the limb biomechanical properties change, to enable a better match between the properties of the plant and its neural controller; the altered MS functions then emerge as an inevitable consequence of this tuning process. Indeed, accompanying the rapid growth in physical characteristics of infants and children[24–27] are significant age-related fluctuations in limb biomechanical properties and joint movement patterns. The previous finding that age-related changes in KSs were more pronounced than changes in MSs[28] further suggests an unstable correspondence between the MSs and their kinematic functions during gait development. As the MSs change, their altered functions may ultimately permit the task to be better performed by fulfilling specific biomechanical needs[11]. For example, during gait development, two neonatal MSs initially provide weight support during stance and facilitate acceleration during swing, respectively, but at a later age, two additional, new MSs emerge at initial contact and push-off to help the toddler walk more smoothly and stably[29]. Thus, according to this view, the changing MSs and their associated functions reflect a developmental learning process that should enable better task execution and skilled performance[3].

Regardless of whether the MSs' biomechanical functions change during development, exactly what factors ultimately drive the changes of MSs during locomotor development have remained obscure. Beyond genetically determined developmental programs[29], it is plausible that MSs on both sides are shaped to result in the bilateral symmetry of certain functional variables, thereby enhancing locomotor stability and postural balance. A previous study has highlighted that as children mature, the drive for symmetrical gait patterns would shape the structures of the locomotor central pattern generators and the neuro-fascia-musculoskeletal system[30]. The emergence of gait symmetry has even been recognized as a hallmark of gait development[28], with 5-month-old infants demonstrating greater capability to perceive bilateral motions and achieve stable posture than 3-month-old infants[31]. The potential role of gait symmetry as a locomotor developmental drive is as well supported by studies of motor adaptation, a process that might share certain common mechanisms with locomotor development, given that early crawling and walking development involves countless motor adaptation and decision-making steps[32]. For motor adaptation, locomotor stability has been identified as a primary driving force, and gait symmetry has been reported to play a more significant role than arm-swing symmetry in maintaining gait stability[33]. Indeed, adults adapt to perturbations in ways that prioritize locomotor stability over minimizing mechanical effort[34] or restoring baseline kinematics[35].

We therefore hypothesize that as infants acquire independent walking, the MSs are shaped, in response to the changing limb biomechanical properties and the initial lack of kinematic symmetry, to ultimately result in bilateral symmetry of the MSs' kinematic/biomechanical functions for enhancing locomotor stability. To test this hypothesis, we collected bilateral electromyographic (EMG) and kinematic data from infants ($n = 11$) longitudinally across four stages of gait development, and reanalyzed previous data[12] from adults ($n = 7$) and elders ($n = 7$) in our comparisons. We first tracked the changes, across the six time points of the entire lifespan in the three groups of participants, of both the MSs and their associated kinematic functions quantified as KSs. We then examined the MS-KS relationship using the kinematic-muscular synergy (KMS) model[36], which directly links any MS with its KS in the same synergy vector. Such a formulation not only reduces MS and KS variability that may arise from noise, but also offers robust MS-to-KS alignment. Next, to characterize the changes of the individuals' limb biomechanical properties, we constructed a personalized neuromusculoskeletal (NMS) model for each infant and estimated several muscle-tendon parameters of the four infantile stages from data-driven simulations. In our NMS model, a default set of muscle-tendon parameters from a generic adult model was adjusted to match the infant's experimental EMG and kinematic data[37–40], thus ensuring robustness and physiological reliability of our pediatric biomechanical simulations[41–45].

Overall, our analytic goals then were to examine how the subjects' MSs and their KSs varied across the different life stages, between the two limbs, and within the subject cohorts; to identify the NMS model-derived muscle-tendon parameters that changed significantly across the infantile stages; and to assess how these parameter changes influenced the MS-KS relationship. Our hypothesis predicts that as the MSs change across the stages, both their associated KSs and specific muscle-tendon parameters also change; that the KS variability across subjects and time points can be attributed to the variability of both the MS and certain muscle-tendon parameters; and that the KSs of both sides become more bilaterally symmetric as the subjects become older.

## Results

To reveal how infants acquire the motor patterns needed for independent walking, we longitudinally recorded EMG and lower-limb kinematics from 11 infants at 4 walking stages (Table 1), which we will refer to as the following infantile stages: weight-supported walking (CH1), hand-supported walking (CH2), early independent walking (CH3), and later independent walking (CH4). Besides, previous walking data from Adults ($n = 7$) and Elders ($n = 7$)[12] were analyzed for a more complete characterization of gait development over the entire lifespan. To identify the motor modules and their functions in driving joint movements, we extracted kinematic-muscular synergies (KMS) from time-synchronized EMG and joint acceleration (JAC) data of each subject at each stage (Fig. 2A). Each KMS comprises components of a MS and the MS's associated KS, both of which are modulated by the same time-varying AC. We then examined how the MS, KS, and AC changed across the 6 life stages (Figs. 1 and 2B).

### Age-related changes of KMS, MS, and KS across infantile stages

To see how the KMSs may change across the infantile walking stages, for each subject, we first matched each KMS of each stage to those of the other stages by maximizing the overall Pearson correlation ($r$) between all matched KMS pairs. As an example, we show the KMSs of one infant, aligned from CH1 to CH4, in Fig. 2B. At CH1, there were manifest co-contractions across many muscles in all four MSs (Fig. 2B, blue bars). At CH2, all MSs started to show activations in more specific muscle groups, but their associated KSs (red bars) also changed from those of CH1, with extents of change larger than those of their corresponding MSs for KMS1 and KMS2. At CH3, the MSs maintained similar shapes from those of CH2, but again, for KMS1 and KMS2, the KSs changed more noticeably than the MSs. From CH3 to CH4, the MSs for KMS2 and KMS3 remained similar, but the MS for KMS1 and all KSs changed to various degrees. This initial examination suggests that the KS, MS, and KMS changed between walking stages, though the KSs appeared to change more prominently than the MSs.

We proceeded to measure their between-stage changes for each subject of the entire infant cohort (Fig. 2C). For all three stage transitions, both the KS and MS exhibited low average between-stage similarity ($r = 0.3 − 0.5$; Fig. 2C). Specifically, KS similarity was even lower than MS similarity at each stage transition, and also generally lower when all three transitions were considered together ($p = 0.003$; Fig. S1A). Between CH1 and CH4, the similarity of KS (at 0.17) was significantly lower than that of MS (at 0.4) ($p = 0.002$, Fig. S1B). In all transitions, KMS demonstrated higher between-stage similarity than MS and KS ($p < 0.01$; Figs. 2C and S1A), but the longer

**Table 1 | Basic information of subjects**

| | | Infant | | | | | | | | | | | |
|---|---|---|---|---|---|---|---|---|---|---|---|---|---|
| Number | Gender | Weight-support (CH1) | | | Hand-support (CH2) | | | Earlier walking (CH3) | | | Later walking (CH4) | | |
| | | Age (yr) | Weight (kg) | Height (cm) | Age | Weight | Height | Age | Weight | Height | Age | Weight | Height |
| 1 | F | 0.54 | 8.2 | 63 | 0.85 | 11.7 | 66 | 1.03 | 9.0 | 71 | 3.47 | 14.5 | 97 |
| 2 | F | 0.52 | 7.8 | 69 | 0.95 | 9.6 | 75 | 1.20 | 10.1 | 80 | 2.43 | 12.5 | 93 |
| 3 | M | 1.21 | 10.3 | 77 | 1.30 | 10.5 | 79 | 1.47 | 11.0 | 81 | 3.12 | 15.3 | 99 |
| 4 | M | 0.92 | 9.5 | 70 | 1.16 | 10.5 | 75 | 1.16 | 10.5 | 75 | 2.72 | 13.9 | 93 |
| 5 | F | 0.96 | 6.6 | 67 | 1.28 | 8.7 | 73 | 1.28 | 8.7 | 73 | 2.45 | 9.6 | 85 |
| 6 | M | 0.93 | 9.4 | 70 | 0.93 | 9.4 | 70 | 1.14 | 9.6 | 73 | 2.21 | 11.6 | 85 |
| 7 | F | 0.68 | 8.0 | 64 | 1.05 | 9.1 | 70 | 1.05 | 9.1 | 70 | 1.76 | 11.0 | 81 |
| 8 | F | 0.85 | 8.3 | 69 | 1.05 | 8.7 | 71 | 1.05 | 8.7 | 71 | 1.44 | 9.4 | 80 |
| 9 | F | 0.87 | 7.6 | 70 | 1.08 | 8.7 | 76 | 1.39 | 9.6 | 80 | 1.84 | 10.1 | 86 |
| 10 | M | 0.89 | 8.7 | 71 | 1.05 | 9.0 | 73 | 1.35 | 9.5 | 77 | 1.85 | 10.3 | 84 |
| 11 | F | 0.76 | 8.8 | 67 | 0.97 | 9.3 | 73 | 1.24 | 10.3 | 76 | 1.64 | 10.5 | 81 |
| | Adult | | | | | Elder | | | | | | | |
| Number | Gender | Independent walking | | | Number | Gender | Independent walking | | | | | | |
| | | Age | Weight | Height | | | Age | Weight | Height | | | | |
| 1 | M | 22 | 83.7 | 183 | 1 | M | 74 | 48.9 | 171 | | | | |
| 2 | M | 25 | 66.2 | 179 | 2 | M | 62 | 65.3 | 159 | | | | |
| 3 | F | 26 | 61.2 | 156 | 3 | F | 73 | 56.5 | 164 | | | | |
| 4 | F | 24 | 60.5 | 166 | 4 | F | 65 | 53.0 | 149 | | | | |
| 5 | M | 25 | 91.1 | 181 | 5 | F | 60 | NA | 156 | | | | |
| 6 | F | 19 | 58.7 | 164 | 6 | F | 61 | NA | 158 | | | | |
| 7 | M | NA | 73.5 | 176 | 7 | M | 65 | NA | 174 | | | | |

the time elapsed between any two stages, the smaller the KMS similarity ($p = 0.01$, Fig. 2D). The between-stage MS and KS similarities were less affected by the between-stage duration, however (MS, $p = 0.31$; KS, $p = 0.62$; Fig. 2D). Besides, we quantified the reduced muscle co-contraction observed in MS across the stages. From the CH1 to the Adult stages, the average sparseness of the MS vectors across all subjects significantly increased (Fig. S2). The data relevant to the above comparisons are provided in the supplementary file, Supplementary Data 1-BetweenStage Comparison.xlsx.

Beyond measuring the extent of MS and KS changes between stages, we also characterized their rates of change by dividing the between-stage synergy differences by the duration elapsed between the stages. At only the CH1–CH2 but not the other transitions, KS exhibited a higher rate of change than MS ($p = 0.043$; Fig. S3A). Furthermore, the KS rate of change itself changed with age, in that it decreased significantly as the infants grew up ($p = 0.047$; Fig. S3B). The MS rate of change, however, remained relatively stable over the infantile age range ($p = 0.438$; Fig. S3B).

## Age-related changes in bilateral KS and MS symmetry over the lifespan

While the KS rate of change was age-dependent, we also wondered whether the similarity of the KS and MS between the two legs may also be age-dependent, noting the previous result that gait symmetry as measured by muscle activities and kinematic parameters increased differently with the children's age[28,46]. We calculated the average bilateral KS and MS similarities of all subjects in each age group. The bilateral KS similarities increased gradually from CH1 to CH4 (Fig. 3A), while the bilateral MS similarities remained stable (Fig. 3B). Both the KS and MS similarities reached a maximum at Adult, but decreased from Adult to Elder. To accurately reveal how the KS and MS symmetry varied with age, we regressed the bilateral KS and MS similarities on the subject's age (in logarithmic scale) with quadratic linear relationships (KS, $p < 0.001$; MS, $p = 0.23$). As indicated by the fitted curves, KS symmetry peaked at about 11 years old (Fig. 3C), but MS

symmetry did not exhibit any well-defined peak (Fig. 3D). Thus, the bilateral symmetry of KS increased through the infantile walking stages, reaching its peak during adolescence and early adulthood, while MS symmetry increased at a relatively slow and steady pace throughout development. The data relevant to the above analyses are provided in the supplementary file, Supplementary Data 2-BetweenLimb Comparison.xlsx.

## Changes of within-cluster subject variability of KS and MS over the lifespan

Based on the above analysis at the individual level, we further examined the within-cluster subject similarity of MS and KS among the whole population. We applied k-means to group the MSs of each stage into clusters, but visualized the clusters by plotting both the MSs and their associated KSs together so that the variability of the KSs associated with each MS pattern can be revealed. For the left leg, in all 6 stages, we identified 3 MS clusters which involved muscles of the crus, thigh, and back, respectively (Fig. 4A; similar results for the right leg are shown in Fig. S4). Within some clusters, we observed some KS components that showed consistent activations across subjects (e.g., Cluster 1 at Adult), some of which even remained stable across the stages (e.g., Cluster 1 of Adult and Elder were similar). However, other KS components in the centroids of certain clusters showed near-zero amplitudes, which likely resulted from cancellations between the positive and negative values that arose from the KS variability within the clusters (e.g., Cluster 2 at CH4). To quantify such within-cluster variation of KMS components throughout development, we calculated the within-cluster standard deviation (SD) for each KS and MS component, and averaged the SD values from all KS or dominant muscular components for each cluster at each stage. Across all stages, the KS components had significantly higher within-cluster SD than the MS components ($p < 0.001$; Fig. 4B). This SD difference is not a trivial consequence of the clustering being performed on the MS because the same result was obtained even when the clustering was performed on the KS instead ($p = 0.003$, Fig. S5). With the clustering on

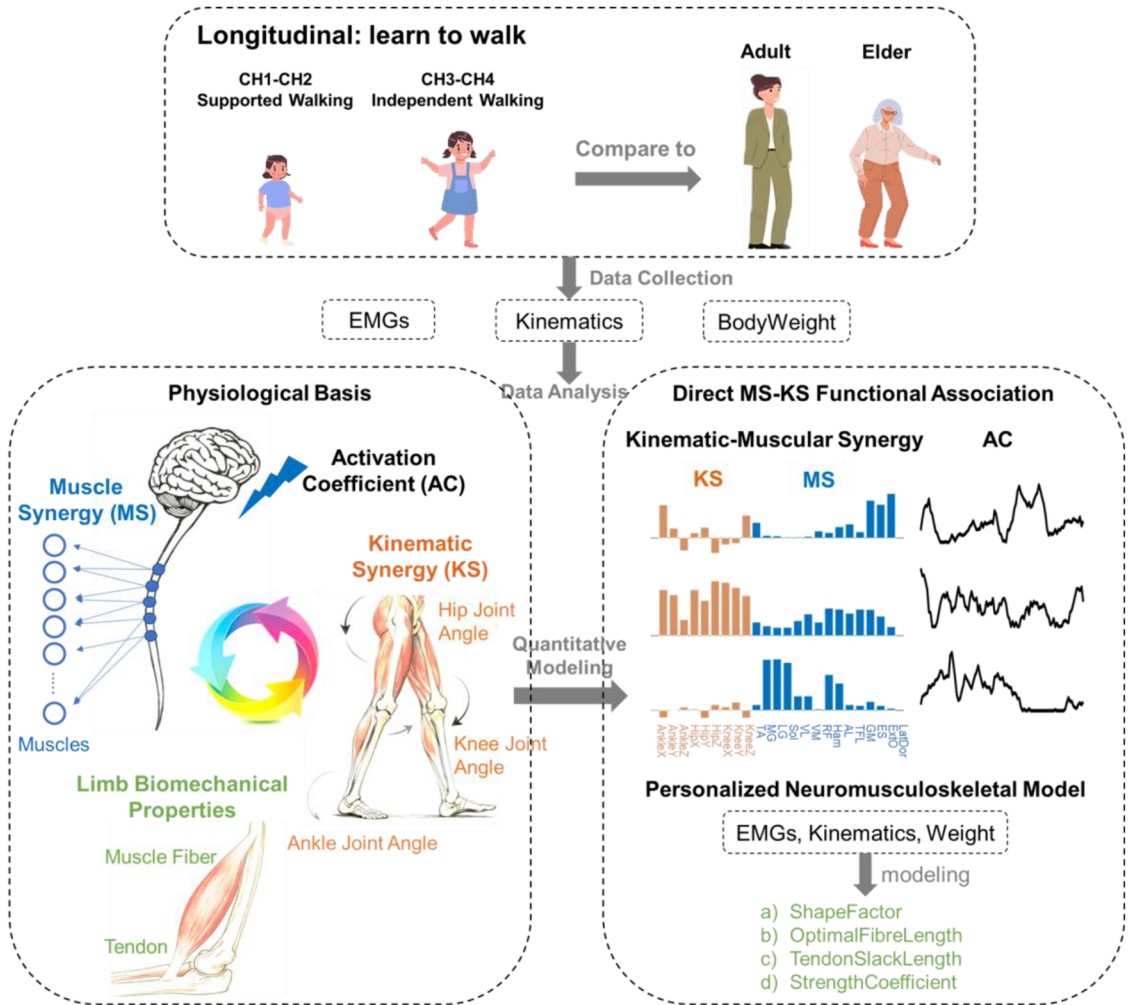

**Fig. 1 | Schematic of experimental and analytical approaches.** This longitudinal study collected surface electromyographic data (EMG), kinematic data, and body weight from 11 infants across four walking stages: CH1 (weight-supported walking), CH2 (hand-supported walking), CH3 (early independent walking), and CH4 (later independent walking). For lifespan comparisons, previously acquired data from 7 adults and 7 elders during independent walking were also analyzed. Body weight was also measured for all subjects. The bottom left panel illustrates the physiological basis of muscle synergy (MS), kinematic synergy (KS), and activation coefficient (AC) while highlighting their interactions with limb biomechanical properties in supporting the development of bilaterally symmetric gait in infants. Quantification of the synergies was achieved through mathematical models (bottom right panel). To link MSs to their kinematic functions, the kinematic-muscular synergy (KMS) model was employed, combining MS (blue bars) and the corresponding KS (red bars) into a single KMS vector, temporally modulated by AC. The MSs here included

activations from the following muscles: tibialis anterior (TA), medial gastrocnemius (MG), lateral gastrocnemius (LG), soleus (Sol), vastus lateralis (VL), vastus medialis (VM), rectus femoris (RF), hamstrings (Ham), adductor longus (AL), tensor fascia latae (TFL), gluteus maximus (GM), erector spinae at L2 (ES), external oblique (ExtO), and latissimus dorsi (LatDor). The KSs capture joint angle co-variation across the ankle, knee, and hip joints on three planes (X-, Y-, and Z-axis) where the X-axis represents extension (+) and flexion (−), the Y-axis, abduction (+) and adduction (−), and the Z-axis, external (+) and internal (−) rotation. To examine developmental changes in infants' limb biomechanical properties across the four stages, we customized a neuromusculoskeletal (NMS) model for each infant at each stage, estimating key muscle-tendon parameters including ShapeFactor, OptimalFibreLength, TendonSlackLength, and StrengthCoefficient (Table 3).

MS (Fig. 4B), the KS SD increased through the four infantile stages and then decreased from CH4 to the Adult and Elder stages. The MS SD, however, remained relatively stable across the six stages except for a significant decrease at Adult ($p < 0.05$). Thus, there was considerable among-subject and across-stage KS variability associated even with the same MS pattern, with this KS variability reaching maximum at the last infantile stage. The data relevant to the above analyses are provided in the supplementary file, Supplementary Data 3-Within-cluster Subject Comparison.xlsx.

### KS variability could not be explained by MS variability alone between stages, limbs, and subjects

If both KS and MS change between developmental stages, it is natural to wonder whether their extents of change are related to each other, in that within the same matched pairs of KMSs from different stages, larger

(or smaller) KS changes would be accompanied by larger (smaller) MS changes. We first regressed the between-stage KS similarity on the between-stage MS similarity and found no statistically significant relationship between them for all three infantile stage transitions ($p > 0.05$; Fig. 5A). Then, we regressed the bilateral KS similarity on the bilateral MS similarity for each of the six developmental stages. Notably, only in Adult was a significant positive correlation found ($p = 0.02$; Fig. 5B). To further reveal any potential association between the KS and MS at the population level, we categorized the KMSs of each stage using the infants' clusters (Figs. 4A and S4) as templates for grouping, and regressed the within-cluster SD of the KS (averaged over the JAC components) on that of MS (averaged over the dominant muscle components) using SD values from the clusters of both limbs. We found no correlation at any stage ($p > 0.05$) except for CH1 ($p = 0.03$, Fig. 5C).

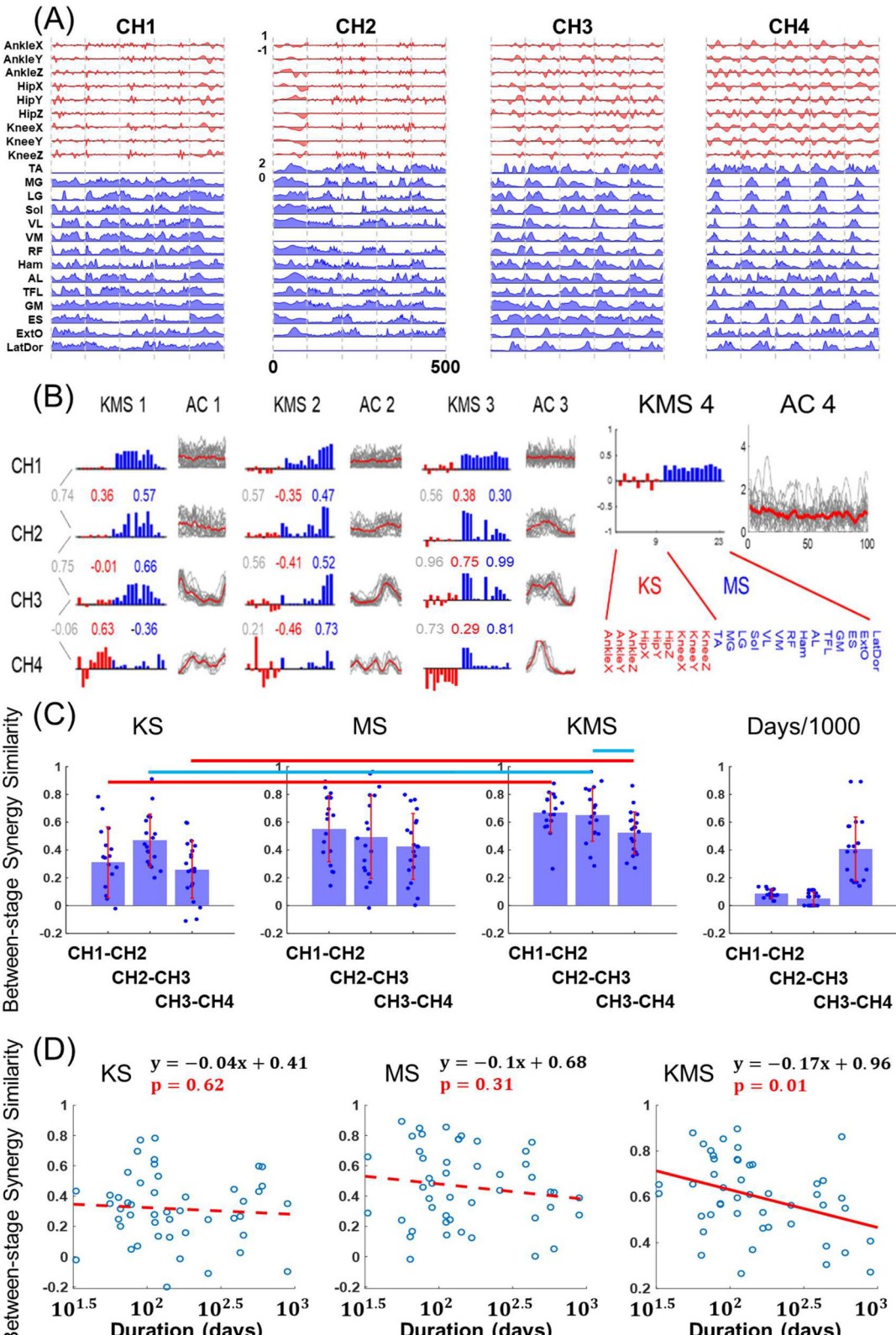

## Changes of limb biomechanical properties across infantile stages

Our analytic results above indicate that in all infantile walking stages, there was substantial across-stage, across-limb, and across-subject KS variability that could not be readily associated with the variability of their associated MSs alone. Only at Adult did we identify a significant correlation between the

bilateral KS similarity and bilateral MS similarity (Fig. 5B), which implies that in the adults, the limb biomechanical properties of the two sides may be similar enough[47] for any inter-limb (dis)similarity of the MSs to be reflected as (dis)similarity of their KSs. This observation raises the possibility that the KS variability arises, at least partly, from the variable biomechanical properties of the infants' limbs. To study the potential impact of the infants' limb

**Fig. 2 | Age-related changes in between-stage synergy similarity. A** Representative examples of preprocessed EMG and joint angular acceleration (JAC) signals from the left lower limb of a single subject, with data of five strides shown in each of four infantile walking stages. **B** For the same subject in (**A**), KMSs were extracted from the concatenated EMG and JAC matrices at each stage and aligned across the four stages. CH1 included four KMSs, while each subsequent stage included three. To quantify developmental changes in MS and KS for each infant, Pearson correlations (*r*) between corresponding KMSs, MSs, or KSs at each stage transition were calculated (values for KMS in gray, KS in red, MS in blue). The ACs of multiple gait cycles for each KMS are displayed on the right of the KMS, with individual cycles in gray and the mean profile in red. **C** Group means of between-stage synergy similarities (KS,

MS, KMS) and the duration (in days) between adjacent stages are presented as bar plots with standard deviations, covering the three stage transitions (CH1–CH2: *n* = 17, CH2–CH3: *n* = 18, CH3–CH4: *n* = 20). Statistically significant differences between conditions are indicated by a red line for *p* < 0.001 or a blue line for *p* < 0.05. **D** To assess whether between-stage synergy changes were associated with longer durations between stages, synergy similarities were regressed on the number of days between two consecutive stages for all subjects. In each subplot, observations are shown as blue circles (*n* = 47) and the fitted regression curve is shown in red. A significant effect was observed for KMS (*p* = 0.01, solid line), but not for KS (*p* = 0.62, dotted line) or MS (*p* = 0.31, dotted line).

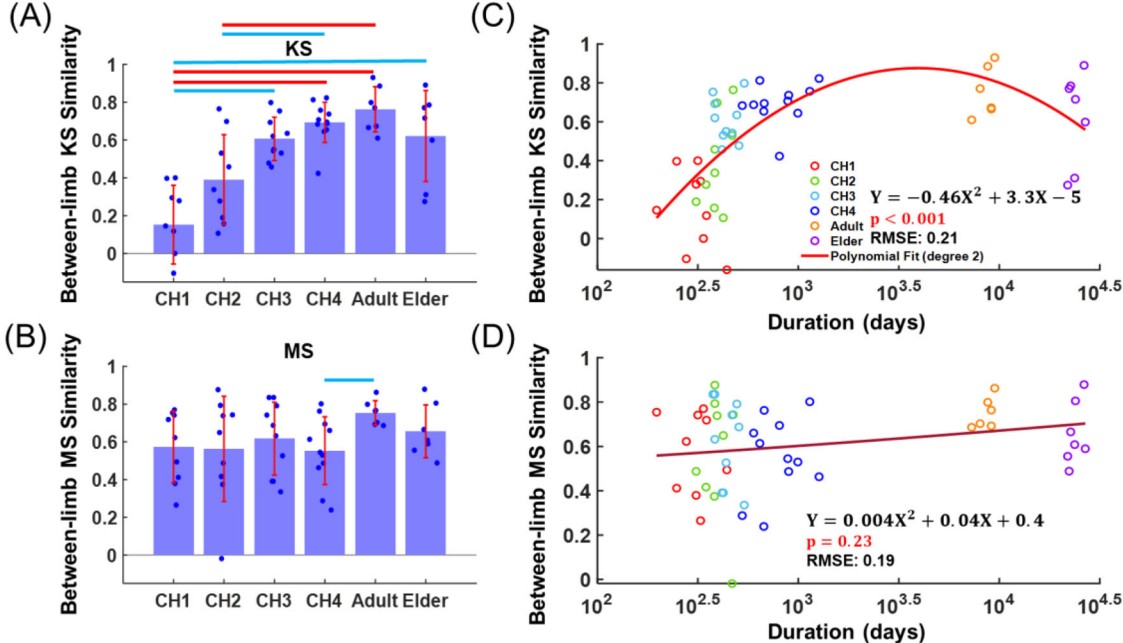

**Fig. 3 | Age-related changes in bilateral KS and MS symmetry.** To assess changes in the bilateral symmetry of synergies across the lifespan, KMSs from the left and right lower limbs were aligned for each subject and stage by maximizing Pearson correlation, after which the similarity of the corresponding KSs and MSs were calculated. **A**, **B** Population means of bilateral KS and MS similarity across six stages (*n* = 9, 9, 10, 11, 7, 7) are shown with error bars, with statistically significant differences between stages indicated above the bars (red: *p* < 0.001; blue: *p* < 0.05). For both KS

and MS, bilateral symmetry increased significantly throughout the infantile stages, reaching its peak in adulthood. **C**, **D** Bilateral similarities in KS and MS were regressed on the subject's age (in days, log10-transformed, *n* = 52) using a quadratic model. The fitted formula, root mean squared error (RMSE), and *p*-value for the whole model are provided in each subplot.

biomechanical features on the shaping of the KS, we built personalized NMS models for each infant at each stage to estimate multiple biomechanical parameters from model-simulated walking, so that how these parameters changed across developmental stages and influenced the MS-KS association can be systematically examined.

With our NMS model, for each infant at each stage, we estimated 10 parameters of 13 muscles on each of both sides related to the muscles' anatomical shape, degree of neuromuscular coupling, force generation capacity, and maximum contraction velocity (Table 3). We then regressed each parameter from each side on the infants' age in logarithmic scale. We observed significant age-dependent changes in 4 parameters (Table S1). Specifically, the ShapeFactor showed an age-dependent decrease for gluteus maximus (GM), rectus femoris (RF), and medial gastrocnemius (MG) (Fig. 6A, *p* < 0.05); the OptimalFibreLength, a decrease for adductor longus (AL), vastus lateralis (VL), vastus medialis (VM), RF, and tibialis anterior (TA) (Fig. 6B; *p* < 0.05); the TendonSlackLength, an increase for AL, RF, TA, but a decrease for hamstrings (HAM) and GM (Fig. 6C; *p* < 0.01); the StrengthCoefficient, an increase for HAM and muscle GM (Fig. 6D; *p* < 0.05). In addition, we also observed a reduction in the absolute bilateral difference in the ShapeFactor of lateral gastrocnemius (LG) (*p* = 0.01) and TendonSlackLength of VL (*p* = 0.009) as age increased (Fig. 6E), indicating that these parameters

became more bilaterally symmetric as limb biomechanical properties changed over time. The data relevant to the above analysis are provided in the supplementary file, Supplementary Data 4-Muscle-tendon Parameters.xlsx.

As a validation of the accuracy of our NMS model and the model-estimated muscle-tendon parameters, we used the model to predict the muscle activities that could produce the recorded kinematics, and evaluate whether the predicted activities aligned with the experimental EMGs. At each developmental stage, the Pearson correlation values between the time-aligned experimental and model-predicted EMGs across infants and validation trials were approximately 0.6 (Fig. S6A), and importantly, among 74/104 cases (26 muscles for both limbs × 4 stages), the correlations in >80% of the trials were statistically significant (*p* < 0.05) (Fig. S6B). This result argues for the validity of our simulations.

## KS variability could be associated with variability of MS and biomechanical parameters

Since ultimately, any regular synergistic joint motion results from the interaction between synergistic muscular activities and limb biomechanical properties, the age-related changes of the muscle parameters described above (Fig. 6) suggest that the KS variability, especially that across the walking stages, can only be understood in light of the variability of both the

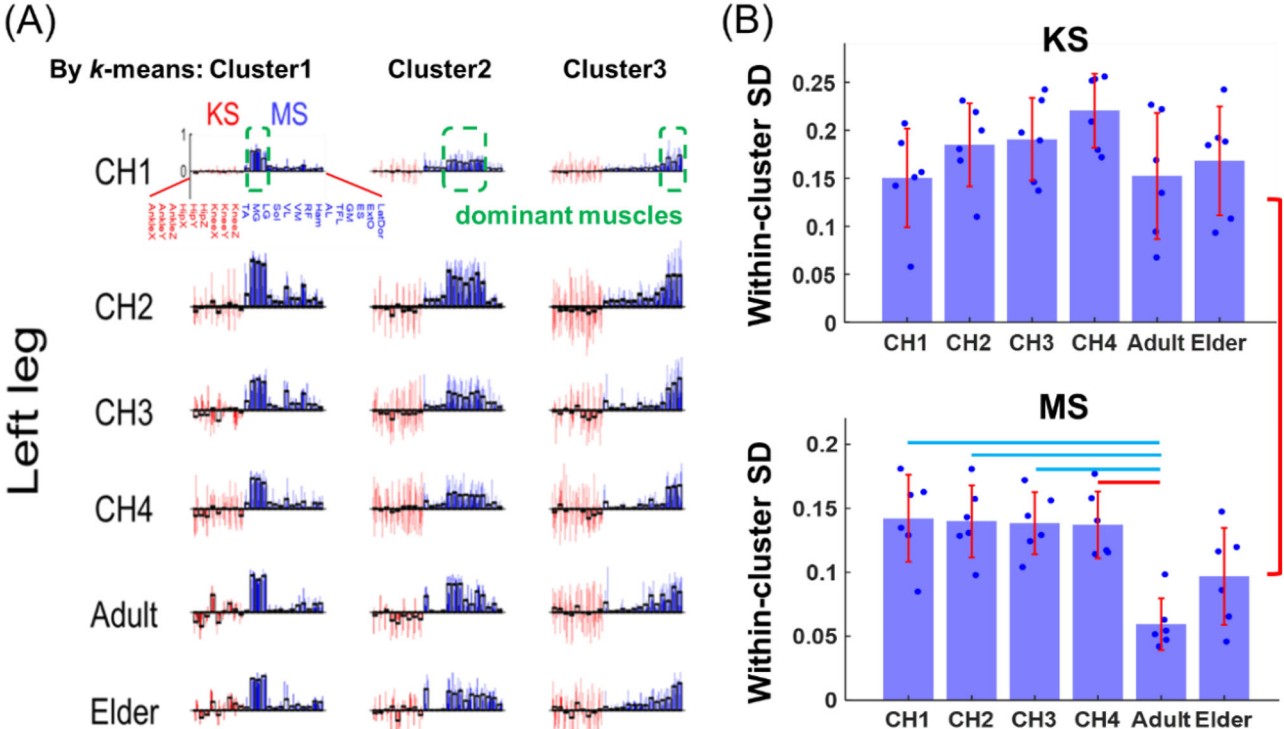

**Fig. 4 | Changes in within-cluster subject variability of KS and MS across the lifespan.** To investigate whether similar MSs lead to similar KSs at the population level, $k$-means clustering was performed for each lower limb at each stage, grouping all subjects' KMSs based on MS similarity and showing the variability of the associated KSs. **A** For the left leg, three clusters were identified at each stage and aligned across all six stages (cluster labels shown on the left). KMSs within each cluster are represented by bars (KS in red, MS in blue), with transparent bars indicating cluster centroids. **B** For each cluster, within-cluster subject variability was quantified for each KMS component by computing its standard deviation (SD) across subjects in the cluster. SDs were then averaged over the nine KS components, and the dominant muscles of MSs indicated in green dotted rectangles in (**A**), to represent within-cluster subject variability in KS and MS, respectively. SDs from six clusters ($n = 6$) across both limbs were compared across stages, and between KS and MS. Within-cluster KS variability was significantly higher than that of MS ($p < 0.001$). Statistically significant differences between stages are indicated above the bars (red: $p < 0.001$; blue: $p < 0.05$).

MS and limb biomechanical parameters. To validate the relationship between these three variabilities, we regressed the within-cluster KS variability of the infantile stages on the within-cluster subject variability of a specific muscular parameter and/or that of the MS, as follows. For each KMS cluster, which comprised KMSs from different infants at each stage (Fig. 4A), we computed the within-cluster SD for each KMS component, and averaged the SD over the dominant MS components to quantify MS variability, but over the 9 KS components to quantify KS variability. For each muscle parameter, we likewise averaged the SD of the parameters for the cluster's dominant muscles to quantify the parameters' variability. For each parameter, there were 6 clusters from both limbs for each of the 4 stages (Fig. 4A, Fig. S4), and thus, variabilities from 24 ($=6 \times 4$) clusters were regressed. From this regression, we found that the within-cluster subject variability of KS significantly increased ($p = 0.03$) when the variability of the StrengthCoefficient increased ($p = 0.03$) (Fig. 7D). When performing regression on data from each stage, we found that at CH4, the within-cluster subject variability of KS significantly increased ($p < 0.05$) when the variability of the MS ($p < 0.05$) decreased, and those of the OptimalFibreLength ($p = 0.009$) (Fig. 7A) or TendonSlackLength ($p = 0.003$) (Fig. 7B) or StrengthCoefficient ($p = 0.002$) (Fig. 7C) increased. These results suggest that across the infants of our cohort, the observed KS variability originates from the variability of both the MS and limb biomechanical properties. The data relevant to the above regressions are provided in the supplementary file, Supplementary Data 4-Muscle-tendon Parameters.xlsx.

**Increasing complexity and stability of the KMSs' activations over the lifespan**

The analyses above concern just the developmental changes of the static KMS vectors. The temporal AC of the KMSs, on the other hand, describe how the MSs and KSs are modulated together within the gait cycle to realize specific biomechanical requirements for walking. It would therefore be instructive to analyze how the ACs change across the developmental stages.

Before independent and stable walking emerges, infants may walk with diverse AC patterns that modulate the MSs. This AC variability may underpin the motor exploration needed for the developing motor system to determine the optimal control scheme for muscle activations and joint movements[6,29]. To characterize the diversity of AC patterns, we performed $k$-means clustering on the ACs for all KMSs from all gait cycles of each subject (Fig. S7A). The number of AC clusters increased significantly from 2 to 5 over the infantile stages, and then decreased slightly from CH4 to the Adult and Elder stages ($p < 0.001$, Fig. 8A). If we interpret the number of AC patterns as a measure of the complexity of motor control signals[10], this observation demonstrates the increase in complexity of the temporal control patterns for the KMSs during infantile locomotor development.

Though the overall diversity of the ACs increased, we were curious to know whether each KMS was also modulated by more or less variable AC patterns across gait cycles over the walking stages. We measured the $r$ between the AC of each gait cycle and the averaged AC across cycles for each KMS of each subject, and found that this $r$ increased steadily across the six stages ($p < 0.001$, Fig. 8B, Gait Stability). To further characterize the level of AC diversity of each KMS, we measured how the ACs of all gait cycles of each KMS were distributed among the AC clusters obtained above (Fig. S7). At the earlier infantile stages, the cycle ACs of each KMS distributed nearly evenly across all AC clusters, but this distribution evolved steadily into one that was concentrated onto a single cluster for each KMS at the later walking stages ($p < 0.001$, Fig. 8B, Pattern Stability; Fig. S7B, C). In addition, the inter-limb $r$ between the ACs for the corresponding KMS's showed an increase from the infantile to Adult stages ($p < 0.001$, Fig. 8B, Gait

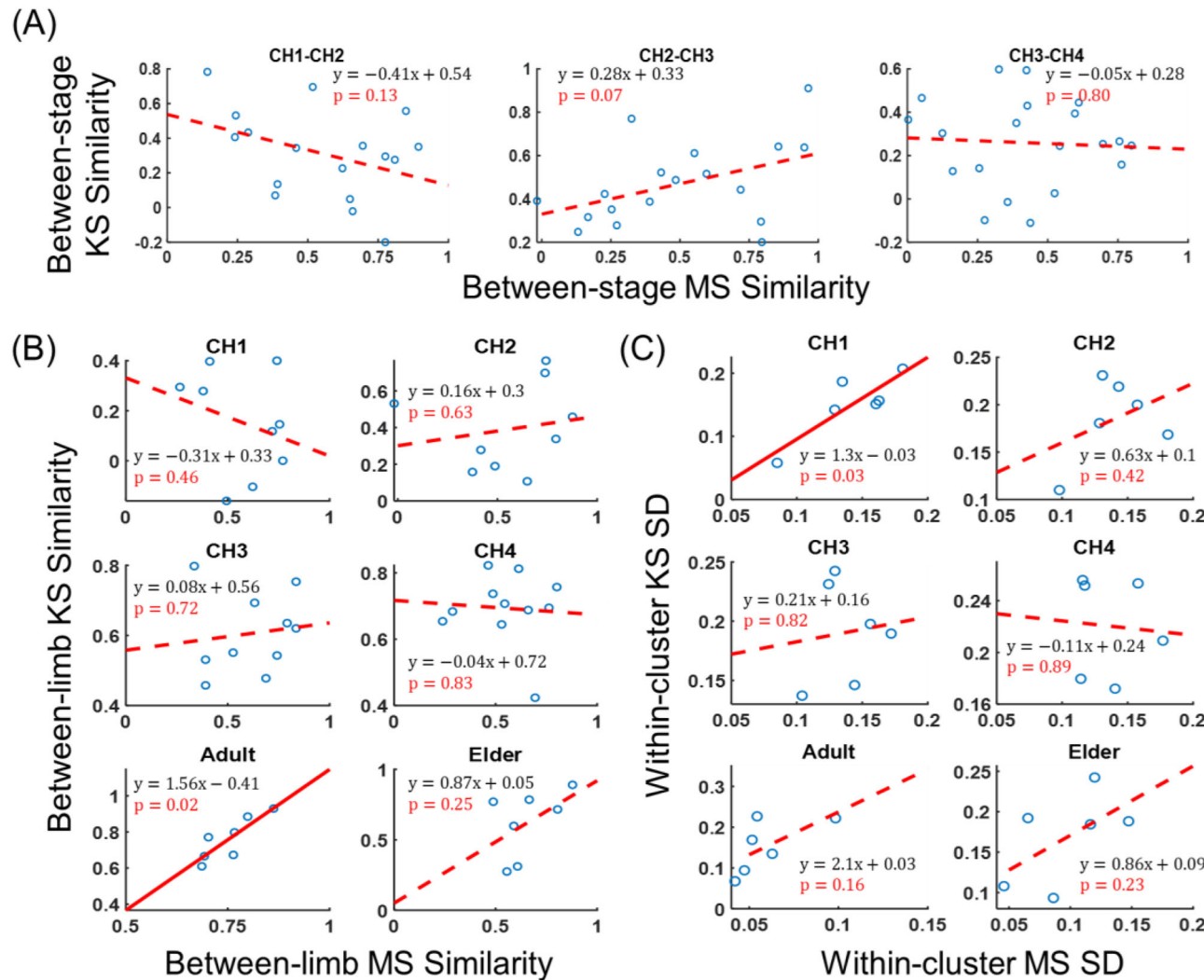

**Fig. 5 | KS variability cannot be explained by MS variability alone across stages, limbs, and subjects.** To directly assess whether changes in KS could be explained solely by changes in MS, KS similarity was regressed on MS similarity across stages, limbs, and subjects. Each subplot follows the same format as Fig. 2D. **A** At each stage transition (CH1–CH2: $n = 17$, CH2–CH3: $n = 18$, CH3–CH4: $n = 20$), between-stage KS similarity (y-axis) could not be predicted by MS similarity (x-axis) alone. **B** At each stage ($n = 9, 9, 10, 11, 7, 7$), between-limb KS similarity could not be explained by MS similarity, except in adults, where a significant relationship was observed ($p = 0.02$). **C** At each stage ($n = 6$), within-cluster subject variability of KS could not be explained by MS variability, except at CH1, where a significant regression was found ($p = 0.03$).

Symmetry). Also, the system chaos embedded within the ACs, as indicated by the maximal Lyapunov exponent (MLE), decreased across the stages ($p < 0.05$, Fig. 8C). All in all, during locomotor development, the ACs became more stable across gait cycles and more symmetric between limbs, with each KMS associated with a more consistent AC temporal pattern that was at the same time less chaotic.

While the above analysis focuses on comparing the KMSs and ACs of the individual subjects at different walking stages, we also performed analogous cluster analysis that considers the KMSs from all subjects as a whole. For each leg, we k-means-clustered the KMSs of all individuals, and found relatively more consistent MS but more variable KS across the six developmental stages, even though the number of KMS clusters were slightly different for each stage (Fig. S8). For the ACs of these clusters, because of the populational variability of the ACs, the mean AC temporal profiles at CH1 and CH2 had no distinct peaks. However, towards the later developmental stages, the ACs became more consistent over the gait cycles, thus resulting in mean profiles with clear and consistent peaks. Across the later stages, the averaged ACs of the matched KMS clusters also showed similar temporal profiles, indicating robust temporal modulations of the KMS clusters. Overall, results from this cluster analysis are broadly consistent with those from individual-based

analysis. The data relevant to the AC analyses presented above are provided in the supplementary file, Supplementary Data 5-AC comparison.xlsx.

## Discussion

In this study, we characterized changes of MS, KS, and limb biomechanical properties across the whole lifespan, and examined how these changes may be associated with each other during gait development. We demonstrated that KS exhibited variability that could not be fully explained by MS variability alone (Fig. 5) for comparisons between stages, limbs, and subjects. By building customized NMS model for each infant at each stage, we revealed age-related changes of several biomechanical parameters (Fig. 6), and showed that within-cluster subject variability of Tendon-SlackLength, OptimalFibrelength and StrengthCoefficient of certain muscles, together with MS variability, could explain the observed KS variability (Fig. 7). Notably, in spite of the observed KS variability, bilateral KS similarity increased significantly as the subjects aged into adulthood (Fig. 3A, C). Overall, our data and analyses have shed light on the functional significance of lifelong MS changes, and how the development of MS, KS, and limb biomechanical properties may be dependent on each other.

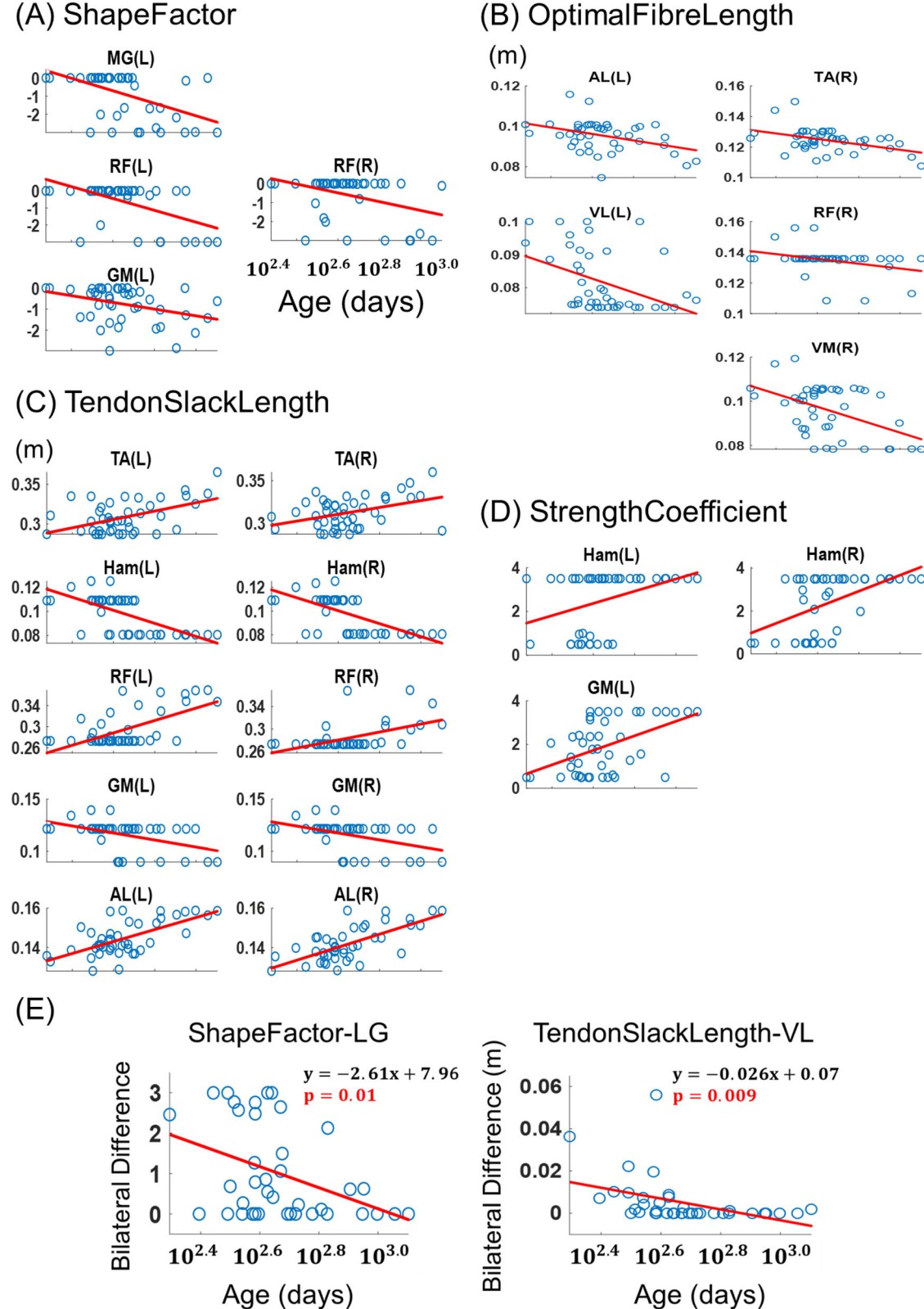

**Fig. 6 | Age-related changes in muscle-tendon parameters across infantile stages.** Using personalized neuromusculoskeletal (NMS) models for each infant at each stage, we quantified 10 muscle-tendon parameters (Table 3) to assess changes in limb biomechanical properties as a function of the infant's age ($\log_{10}$ of days). Of these, four parameters showed significant age-related effects (**A–D**). For each subplot, the corresponding muscle is indicated at the top with the side of the muscle shown in parentheses (left, L; right, R). Data from the infant cohort are represented by blue dots ($n = 44$), and regression lines are shown in red. Regression formulas and $p$-values for the whole model are provided in Table S1. OptimalFibreLength and TendonSlackLength were measured in meters (m). **E** To assess whether muscle-tendon parameters of the two lower limbs become symmetrical with age, we regressed the between-limb difference for each parameter on the infant's age ($\log_{10}$ days, $n = 39$) and reported those parameters whose $p$-values were less than 0.05. Subplots follow the same format as Fig. 2D.

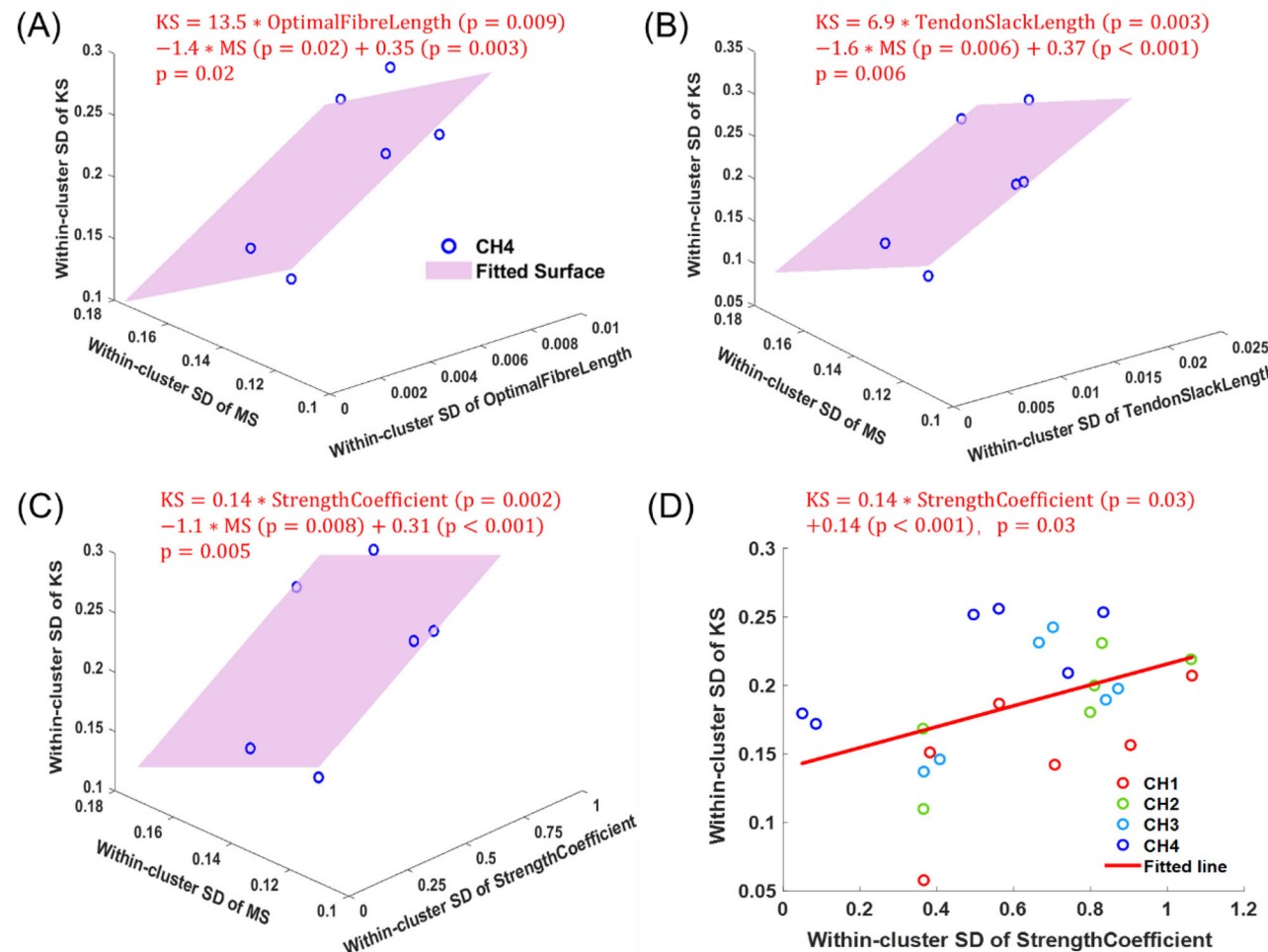

**Fig. 7 | Within-cluster subject variability of KS can be explained by variability of MS and muscle-tendon parameters.** To investigate the influence of limb biomechanical properties on the MS-KS relationship, simulated muscle-tendon parameters (Table 3) across four infantile stages were incorporated into the regression analysis of within-cluster subject variability (Fig. 5C). For each cluster, within-cluster subject variability for each muscle-tendon parameter was quantified by averaging its inter-subject SDs across the cluster's dominant muscles (Fig. 4A). **A–C** At CH4, OptimalFibreLength, TendonSlackLength, and StrengthCoefficient were identified as significant predictors of MS-KS association. Data points for six clusters at CH4 are shown as dark blue circles ($n = 6$), with the fitted regression surface displayed in semi-transparent purple. **D** StrengthCoefficient demonstrated a significant effect on inter-subject KS variability across four infantile stages ($n = 24$), with the regression line shown in red. Observations from the four infantile stages are color-coded (CH1: red; CH2: green; CH3: light blue; CH4: dark blue), with each stage comprising six data points (six clusters from both limbs; Figs. 4A and S4). Regression coefficients and the $p$-values for the whole model and each factor are indicated at the top of each subplot.

## Changes of MS over the entire lifespan

We monitored the development of MS in each infant by comparing the MSs across the walking stages, and found that the MSs underwent significant alterations from the first to last infantile stages. For instance, the Pearson similarity of MS between CH1 and CH4 was only 0.4, which suggests that over half of the CH1 MSs were modified by the CH4 stage (Fig. S1B). Also, through the walking stages, the MSs became sparser in structure with fewer number of muscles active in each MS (Fig. S2). These observations of MS changes are consistent with the prior result that neonatal stepping MSs changed during locomotor development[6,8,29,48], so that by the toddler stage, the initial MSs fractionated into more fragmented MSs, with each MS controlling a more specific set of muscles[6,10,27,48,49]. The above findings argue against the viewpoint that locomotor MSs are entirely unchanging entities that are specified very early in life[24,50]. Such MS changes may arise from either time-dependent genetically determined processes[29] and/or experience-dependent plasticity[49].

Even though the MSs changed between walking stages within each individual, at the population level the dominant muscles of MSs appeared to be somewhat consistent across subjects in all walking stages, as indicated by the small within-cluster SD across MS components (Fig. 4B). This

consistency may reflect genetically encoded processes that specify the MSs' course structures[29], or simply from sensorimotor experience shared by all subjects in the cohort during development.

Overall, our MS findings suggest the following picture of human locomotor MS development. Neonates are born with a "generic" set of MSs that are similar across individuals[29], but not tailored to the individuals' limb biomechanical properties (see below) required for stable gait. As the infants progress through their trajectories of neuro-musculoskeletal development together with their individually unique sensorimotor experiences[48], their MSs are adjusted[51,52] to suit the biomechanical properties of both limbs to primarily support postural balance and locomotor stability[49], which may be achieved by symmetrizing the KSs (see below). The MSs may also be adjusted to fulfill other secondary goals, such as minimizing energy expenditure[34,49].

## Changes of limb biomechanical properties during gait development

To understand how the infants' limb biomechanical properties change across stages, we developed a personalized NMS model for each infant at every stage and found that 4 muscular properties exhibited age-related

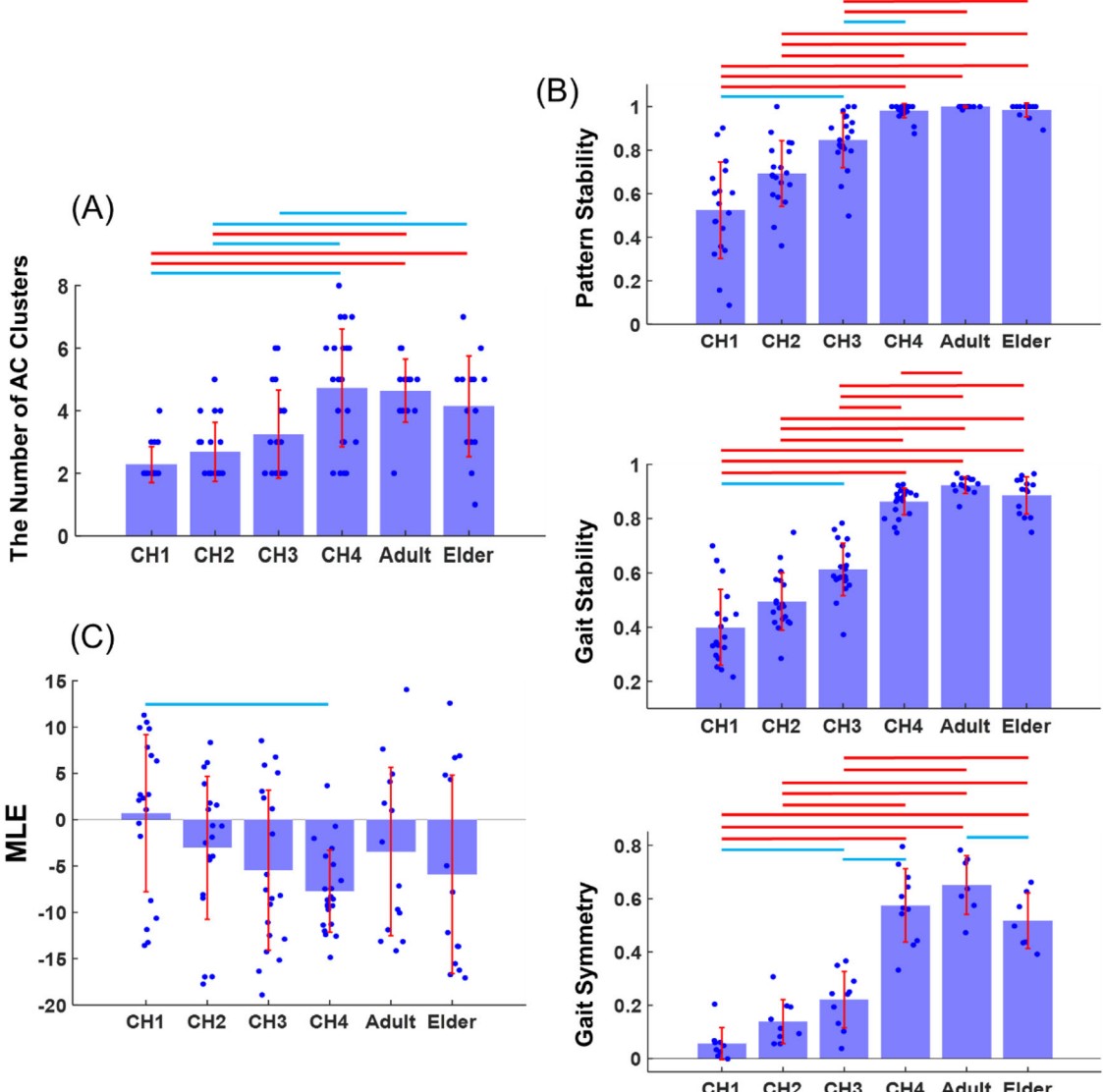

**Fig. 8 | Increasing complexity and stability of AC across the lifespan. A** The complexity of AC patterns was quantified as the number of clusters derived from multi-gait-cycle ACs of all KMSs from each limb of each subject at each stage ($n$ = 18, 19, 20, 22, 14, 14). At the populational level, the number of AC clusters increased significantly from 2 to 5 across the infantile stages, then decreased slightly from CH4 to Adult and Elder stages. **B** To assess whether AC patterns become more stable across gait cycles and between lower limbs, we measured (i) **Pattern Stability** (top panel), the distribution of cycle ACs for each KMS across the identified AC clusters (Fig. S7A) for each subject at each stage ($n$ = 18, 19, 20, 22, 14, 13). Early infantile stages showed a nearly even distribution across the AC clusters, which evolved into one that was concentrated onto a single cluster at later walking stages (Fig. S7B, C). (ii) **Gait Stability** (middle panel), quantified by the correlation ($r$) between the AC of each cycle and the mean AC for each KMS and subject. Gait stability increased steadily across the six stages ($n$ = 18, 19, 20, 22, 14, 14). (iii) **Gait Symmetry** (bottom panel), quantified by the inter-limb correlation ($r$) between ACs for corresponding KMSs. Symmetry increased from the infantile to Adult stages ($n$ = 9, 9, 10, 11, 7, 7). **C** System Chaos: The maximal Lyapunov exponent (MLE) of ACs was computed to quantify system chaos, showing a decreasing trend across stages ($n$ = 18, 19, 20, 22, 14, 14). All subplots follow the same format as Fig. 3A.

changes (Fig. 6). Through such changes, the infants may exert more stable and efficient control of muscle activations, thereby enhancing force generation over time to facilitate the maturation of walking pattern. For instance, as defined in the CEINMS toolbox, the ShapeFactor is a constant that determines the nonlinearity in the relationship between neural activation and muscle activation. Specifically, it modulates how changes in neural input are translated into muscle activation. The reduction in ShapeFactor in muscles GM, RF, and MG (Fig. 6A) indicates that for these muscles, fluctuations in neural activation have a reduced effect on muscle activation. This implies that the muscle activation level becomes relatively more stable and less sensitive to small, step-to-step variations in neural control. Consequently, this greater stability likely contributes to more consistent force generation from these muscles during walking, even in the presence of neural variability[53]. The increase in StrengthCoefficient in the Ham and GM (Fig. 6D) suggests an amplified maximum isometric force and enhanced muscle strength for these muscles. Also, the decrease in the OptimalFibreLength for maximal force generation in AL, quadriceps, and TA (Fig. 6B) implies that their force levels may be maintained even as they are shortened with contractions[53]. Finally, the increase in TendonSlackLength in TA, RF, and AL of both legs (Fig. 6C) contributes to a broader range of joint motion and more flexibility in their activations[53], while the decrease in TendonSlackLength in Ham and GM implies that smaller fiber activations are needed to eliminate the slack from these muscles.

Besides, the biomechanical parameters discussed above changed similarly in both legs. Specifically, there were significant increases in the bilateral symmetry of ShapeFactor for LG and TendonSlackLength for VL as the

infants progressed through the developmental stages (Fig. 6E). Muscle LG contributes to ankle stabilization and support during early stance, while muscle VL plays a pivotal role in knee extension and stabilization throughout stance. Enhanced symmetry in LG's ShapeFactor and VL's TendonSlackLength is expected to facilitate the generation of similar force amplitudes and patterns in lower limb muscles of both sides[53], thereby improving knee and ankle stability as well as the infants' weight-bearing capabilities as they acquire independent walking[54]. This observation indicates a tendency for the infants to increase the symmetry of certain limb biomechanical parameters, likely for facilitating postural balance during gait.

## Emergence of bilateral KS symmetry during gait development

The KSs we extracted from the joint-motion data summarize the synergistic joint coordination patterns embedded within the data, and may be considered as one representation of the kinematic functions implemented by the extracted MSs[22,23]. However, in addition to the muscular compositions and activations of the MSs, the geometry and properties of the musculoskeletal system contribute to the generation of the net joint torques that drive joint motions. Hence, the limb biomechanical properties, serving as a crucial intermediary, potentially influence the MS-KS relationship. We therefore studied how the MSs were paired with the KSs thoroughly to understand the role of limb biomechanical properties in specifying MSs' kinematic functions.

Our comparison of the KSs of each individual across the 4 infantile stages revealed that the KS underwent more substantial time-dependent changes than the MS (Figs. 2C and S1). Also, the rate of KS change decreased more prominently with aging than the rate of MS change (Fig. S3). The greater variability of KS as compared with that of MS may reflect the sensitivity of the KS to any immediate biomechanical constraints and postural demands of the task, such as any stride length or step width adjustments for balance, as well as fluctuations in subject's overall activity level, walking speed, and other environmental conditions such as the texture of the walking surface or the presence of obstacles[24,55,56]. But most notably, through the stages, the KS exhibited a more pronounced increase in bilateral symmetry than the MS (Fig. 3), consistent with the prior result that gait symmetry measured by KS displayed more age-related changes than that by MS during child development[46]. Since the structure of any KS presumably emerges from the interaction between its corresponding MS and limb biomechanical properties, any across-stage KS changes must result from changes of either the limb biomechanical properties, as discussed above (Fig. 6), and/or the MS. It is possible that during gait development, the MSs for both limbs are finetuned by a process that accommodates limb-specific changes in musculoskeletal properties to gradually symmetrize the KSs bilaterally, so that more stable gait can be achieved. Concurrently, the increase in symmetry of certain biomechanical parameters discussed above (Fig. 6E) should also contribute to this process. Alternatively, we cannot exclude the possibility that some KS changes, especially those from CH1 and CH2 to CH3 and CH4, may be attributed to the weight support provided to the infants during the former but not the latter stages, even though some prior studies suggest that both joint angles and kinetics of the lower extremities should not be affected by weight support[57,58].

Additionally, to directly examine the MS-KS association, we performed regression analyses to reveal that throughout the lifespan, KS variability could not be solely elucidated by MS variability alone (Fig. 5), thus further demonstrating that variability of limb biomechanical properties contributes to the observed KS variability. Interestingly, when comparing the MSs and KSs between the two legs, among the 6 life stages analyzed only in adults did we observe a significant correlation between MS similarity and KS similarity (Fig. 5B). Possibly, this finding reflects a high degree of bilateral symmetry of certain biomechanical properties in the adult but less in the other life stages, so that only across the adult subjects can the inter-limb KS similarity be directly related to MS similarity. Presumably, these bilaterally symmetric parameters must be critically relevant to the determination of the KSs in adults. This finding is consistent with the previous result that in adults, the KSs and MSs of two-digit grasping[22] and locomotion[23] could be linked with one-to-one correspondence.

To identify the exact limb biomechanical parameters that are the most relevant to KS determination, we regressed the within-cluster subject variability of KS on the variability of both the MS and simulated muscle-tendon parameters of a large number of muscles, and found that OptimalFibreLength, TendonSlackLength and StrengthCoefficient exerted the most impact on MS-KS association (Fig. 7). Indeed, a previous gait simulation study[59] demonstrated that the TendonSlackLength was the primary determinant of muscle-tendon functions of the plantar flexors as these muscles' metabolic demand was 28 times more sensitive to changes in TendonSlackLength than in tendon stiffness. Thus, the developmental increase in bilateral symmetry of the VL TendonSlackLength suggested by our personalized NMS model (Fig. 6E) may play a critical role in the increase in bilateral KS symmetry (Fig. 3A, C) and the emergence of the direct relationship between the inter-limb MS and KS similarity in adults (Fig. 5B).

## Dynamic co-development of MS, KS, and limb biomechanical properties

Our data suggest that the KS, MS, and limb biomechanical properties of infants exhibited distinct dynamics of developmental changes, but how did they all become more bilaterally symmetrical at the adult stage? At the initial CH1 stage, both the asymmetrical limb biomechanical properties and the "untuned" MS result in the asymmetrical KS ($r = 0.18$; Fig. 3A). Such KS asymmetry would introduce bilaterally imbalanced sensory feedback during gait, which would then drive the motor system to refine the MSs on both sides to adapt to any inter-limb differences in limb biomechanical properties[60,61], thereby increasing KS symmetry. But as the MSs are fine-tuned, the limb properties concurrently develop towards higher bilateral symmetry (Fig. 6E). Indeed, as suggested by this and other studies[24–27], limb biomechanical properties change rapidly in the first two years of life (Fig. 6), a period that coincides with the presumed critical period of human motor development[34]. Such symmetrization of limb properties in turn demands more MS symmetry for an increase in KS symmetry, and indeed, we observed that both KS and MS symmetry increased rapidly from CH1 to CH3 (Fig. 3A, B). The limb biomechanical condition should also contribute to the shaping of the MS through modulations of neuronal properties, network connectivity, and nerve conduction velocity[27,62]. Therefore, during gait development, the MS, limb biomechanical properties, and KS interact reciprocally and co-develop in an organic way to eventually achieve higher levels of bilateral symmetry of all three at adulthood (Fig. 3).

Thus, as walking matures, both the MS and KS are modified[52] in a way that the MS-KS association is adapted to the varying limb biomechanical properties to result in more bilateral symmetry of KS. Consistent with this interpretation, symmetry is considered critical in shaping the neural circuitry structures of locomotor central pattern generators and the development of the brain and the neuro-fascia-musculoskeletal system[30]. Moreover, 5-month-old infants could perceive bilateral motion and realize balance control more readily than 3-month-old infants, who could only respond to the absolute and relative motions of a single limb[31], thus indirectly suggesting that responding to and producing bilaterally symmetric motion is a developmental trait. It is possible that such changes are predominantly driven by a process that enhances locomotor stability. Prior studies have likewise suggested that increasing stability is the primary factor that dictates how motor patterns change during locomotor adaptation[30,63,64]. In a study that delivered robotic perturbation to the leg, participants were found to adapt more strongly to step-length perturbation than to step-height perturbation, so that locomotor stability could be maintained even at higher mechanical costs[34]. Also, when facing a velocity-dependent resistance, subjects adapted with a fast motor response to ensure locomotor stability without restoring the baseline kinematics[35]. A similar stability-maintaining driving force may also be at work during gait development.

## Increasing complexity and stability of the KMSs' activations over the lifespan

The AC of KMS reflect how the MSs and KSs are modulated together within the gait cycle. To examine how the AC patterns varied among the

population, we performed *k*-means clustering on the ACs of each subject at each stage, and revealed that the number of AC clusters significantly increased from 2 to 5 across the four infantile stages (Fig. 8A). The increased AC complexity suggests that the MSs are controlled with finer temporal precision and more flexibility as walking matures[29]. Such refinement of MS activations may well be necessary to adapt the MSs' activations to the changing limb biomechanical properties.

We also examined whether each KMS for each individual was modulated by more or less diverse patterns over the developmental stages by calculating the similarity between the ACs of multiple gait cycles and the AC's maximal Lyapunov exponent (MLE). Stages CH1 and CH2 were characterized by prominent between-cycle AC variability[48] (Fig. 8B, Gait Stability) and strong systematic chaos (Fig. 8C), suggesting that the motor system may be engaged in an exploratory search of the optimal AC by varying the activation timing of the MSs. Stages CH3 and CH4 were characterized by a more stable and unique association between the KMS and AC[6] (Figs. 8B and S7, Pattern Stability), suggesting that as infants gradually manage walking better, the motor system tends to activate specific MSs with more stable rhythm. We also found that the AC between limbs became more similar from CH1 to Adult (Fig. 8B, Gait Symmetry), which may contribute to an enhanced stability of walking. Overall, motor exploration in early stages enabled the motor system to search for optimal AC and gradually converge on stable and symmetrical gait patterns.

### KMS as a tool for examining functional relationships between MS and KS

The extraction of KMS is a powerful method for examining the direct link between MS and KS, potentially representing an improvement over the conventional practice of analyzing the MS and KS separately[22,23,36]. The KMS approach may reveal certain functional MS-KS connections that would remain hidden with separate MS and KS extractions because of noise and other peculiarities in the data structure. Also, KMS analysis should mitigate the difficulty of matching the separately extracted MSs to their corresponding KSs.

To further support the utility of KMS, it has been shown that the functional connection between MS and KS revealed by KMS is consistent with current understanding of the biomechanical functions of the MSs for multi-directional reaching[36], hand grasping[65], and adult locomotion both in experiments[14,66] and simulation[15]. As a further validation, our MS and AC results derived from KMS (Figs. S2 and 8A, B) are consistent with the prior findings[6,10,29,48] that the sparsity of MS and number of AC increased across the gait developmental stages. Our study demonstrates that KMS is a valid approach of dissecting the biomechanical functions of neuromotor modules derived from either EMG and/or neuronal recordings.

### Limitations and future work

This study has the following limitations. First, concerning our experimental design, the parents' means of providing external support to the infants as they walked in CH1 and CH2 were not subject to any standardized requirements, thus potentially introducing variability to the synergies and parameters identified. Future studies would benefit from the use of a consistent, quantifiable weight support system. Additionally, there were a few recording sessions (5 out of 44 sessions, for $11 \times 4 = 44$ time points) in which the infant performed functional walking tests indicative of two developmental stages on the same day. This situation arose either because we missed the ideal recording time window for a stage, or because these infants developed with a trajectory characterized by considerable overlap between stages. An alternative scheme of staging locomotor development that would permit a more unambiguous characterization may be explored in the future. Also, it would be helpful to further increase the sample size, and to record the exact onset time of both supported and independent walking of the infants for a more complete and precise analysis.

Second, concerning the extraction KMS, before extraction, we introduced a delay of 50 ms to the measured EMGs relative to the joint kinematics data to account for the electromechanical delay, which was assumed to be constant in all stages. However, this delay value is based on previous findings from adult humans[36,67], and infants may have longer delays due to their immature NMS systems. To the best of our knowledge, the lower limb electromechanical delay time for human infants has not been properly documented in the literature. But in a preliminary analysis, KMS extraction from a data set with even a 40-ms deviation of the delay time introduced yielded KMSs that correlated significantly with the original KMSs ($r = 0.90$).

Third, concerning the estimation of muscle-tendon parameters, due to limitations in our experimental setup, the ground reaction force and other external force were not recorded during our sessions. This omission may affect the accurate calculation of joint moments using inverse dynamics and thus impact the calibration of muscle-tendon parameters in our infant cohort. However, as shown in our model evaluation results (Fig. S6), the model's performance in predicting muscle activations remained robust, and most importantly, was also consistent across all four infantile developmental stages. This suggests that, while the accuracy of the absolute values of the parameters may be affected, their relative trends across the infant developmental stages are at least still reliably captured. But future infant NMS modeling should include the ground reaction forces and other support forces to further improve accuracy of the model.

Lastly, we note that the decreasing trend in OptimalFibreLength observed across the four infantile stages (Fig. 6B) is indeed surprising given the concurrent increases in body height and muscle length. We suspect that this decrease is physiologically plausible in the following way. In the rat, muscles of neonates contain primarily type I (slow-twitch) and type IIC muscle fibres[68], with the latter type being a primitive form of type II fibres that differentiates into type IIA and IIB fast-twitch fibres later in life. It is possible that in humans, the proportions of slow- and true fast-twitch fibres likewise change during development, so that when the muscles mature, there are relatively more of the latter type. Interestingly, a previous study finds that when compared with type I slow-twitch fibres, type II fast-twitch fibres could produce similar forces at shorter sarcomere lengths[69]. Thus, the increasing proportion of true fast-twitch fibres in the muscles may lead to a decrease in the overall OptimalFibreLength of the muscles. Further studies are required to confirm this conjectural mechanism.

## Conclusion

In this study, we argue that as infants acquire independent walking, the MSs, limb biomechanical properties, KSs, and the MSs' ACs co-develop in a way to ultimately promote bilateral symmetry of the KSs for ensuring locomotor stability. Thus, enhancing locomotor stability through achieving symmetrical gait patterns may be the major driving force that dictates how the mature MSs of both sides are structured at the end of the developmental learning process. An understanding of how this developmental force interacts with other potential driving forces, such as genetic programs[29] and other typical or atypical sensorimotor experience[70], awaits future research. The details of the neural circuits that underpin the MS-KS co-development also remain to be dissected in future studies.

## Materials and methods
### Subject recruitment and Tasks
We enrolled 11 healthy children through word of mouth and tracked their locomotor development longitudinally across four stages of walking (CH1 to CH4; Fig. 1 and Table 1). Subjects were excluded if they had a history of significant medical conditions that could impact motor development or render them unable to complete the study protocol. At CH1 (age 6.3–14.7 months; weight-support walking), infants walked with weight support provided by their parents, who lifted them by the shoulders. At CH2 (10.3–15.8 months; hand-support walking), children walked with partial weight support from their parents' hands. Once children could walk independently, they were evaluated at CH3 (12.5–17.9 months; early independent walking) and then CH4 (17.5–42.3 months; later independent walking). For some subjects, two functional walking tests indicative of two consecutive developmental stages were recorded on the same day either because we missed the ideal recording time window for a stage, or because

these infants developed with considerable overlap between stages (Subject 6, tests for CH1 and CH2; Subjects 4, 5, 7, and 8, tests for CH2 and CH3). In addition, we reanalyzed data originally recorded for a previous study[12] from two subject groups: Adults (age 23.5 ± 2.6 years; $n = 7$, 3 females) and Elders (65.7 ± 5.6 years; $n = 7$, 4 females). These participants walked overground independently at self-selected speed. Prior to the experiment, all procedures were approved by the Joint Chinese University of Hong Kong-New Territories East Cluster Clinical Research Ethics Committee (protocols 2017.690 and 2019.498). Written informed consent was obtained from all participants, or their guardians for those under age 18.

### Data recording and preprocessing

To capture lower limb movements and their corresponding muscle activations, we recorded both kinematic and surface EMG signals. Lower limb kinematics were recorded using a three-dimensional motion capture system (Vicon, Oxford Metrics, Oxford, UK; 100 Hz). Infrared reflective markers ($n = 18$) were attached to anatomical landmarks following the lower-body plug-in gait model[71,72] to track limb movement. Simultaneously, EMG were recorded from 28 lower limb and trunk muscles (14 muscles on each side) using two wireless EMG systems (Trigno, Delsys, Boston, USA; 2000 Hz). Surface electrodes were placed on the muscle bellies and oriented according to the fiber direction. The recorded muscles included TA, MG, LG, soleus (Sol), VL, VM, RF, Ham, AL, tensor fascia latae (TFL), GM, erector spinae at L2 (ES), external oblique (ExtO), and latissimus dorsi (LatDor) (Fig. 1). Attachment positions of the sensors followed the guideline of Surface Electromyography for the Non-Invasive Assessment of Muscles —European Community project (SENIAM)[73]. Prior to attachment, the relevant skin surfaces were cleaned using alcohol. To minimize motion artifacts, white surgical tapes (3 M™ Transpore™, USA) and self-adherent bandage wraps (3 M™ Coban™, USA) were used to securely stabilize the sensors. For each trial, the EMG data were acquired as inputs to the Vicon system and exported as hpf files. All EMG and kinematic data were time-synchronized.

Following data collection, the kinematic data were analyzed using Visual3D (V3D, C-Motion, Washington DC, USA), which provides model-based reconstruction of the hip, knee, and ankle angles based on the marker trajectories. For each joint, the three-dimensional angles were decomposed into the X- (+ for extension, − for flexion), Y- (+ for abduction, − for adduction), and Z-axes (+ for external rotation, − for internal rotation), respectively (Fig. 1). Gait cycles were identified based on the height of the heel markers using the O'Connor algorithm[74] integrated within V3D. To clarify the joint-angle profiles, marker trajectories of the extracted cycles were low-pass filtered using a Butterworth filter (cutoff frequency of 6 Hz). Only the gait cycles with good data quality were selected for further analysis. The number of selected gait cycles for the subjects are reported in Table 2. Joint angular accelerations (JAC) were then calculated for each subject, gait cycle, and joint using MATLAB (Math-Works, Natick, MA, USA). To ensure comparability across subjects, gait cycles, and joints, the JACs of each joint were normalized between −1 and 1 by dividing them by the maximum absolute values. Additionally, the length of the gait cycles was resampled to 100 time points for ease of subsequent analyses.

To preprocess the raw EMG, a high-pass window-based finite impulse response (FIR) filter (50th order; cutoff frequency of 50 Hz)[12,49,75,76] was first applied to remove motion artifacts and clarify the envelope of muscle activity. The signals were then rectified, low-pass filtered using a window-based FIR filter (50th order; cutoff frequency of 20 Hz) to remove noise, and then integrated over 20-ms intervals. To further eliminate occasional high-amplitude spiky noise, we applied the Savitzky-Golay filter[77] to the EMG. High-amplitude spikes were identified using a thresholding criterion, where values exceeding 3–5 times the standard deviation above mean were flagged as noise. The filter then smoothed the signal by fitting a polynomial of a specified order to a moving window of data points, effectively reducing transient noise while maintaining the physiological features of the signal.

Here, we set the polynomial order to be 2, window size to be 3, and fixed the noise threshold using Otsu's method[78]. After the spike removal, each muscle's EMG was normalized to unit variance for each gait cycle, ensuring comparability across muscles and subjects. The entire EMG trial was segmented based on the identified gait events[74] (such as heel-strike and toe-off). To correct for the electromechanical delay[36,67] associated with force generation, a delay of 50 ms was introduced to the EMG relative to the kinematics. For each subject and gait cycle, the amplitude of EMG envelopes of each muscle was normalized to the range of 0 to 2 to match the range of JAC, and resampled to contain 100 time points, thus ensuring consistent length across cycles for subsequent analyses.

### Kinematic-muscular synergy extraction

In the literature[66,79], muscle synergies (MS) were often identified from multi-muscle EMG using factorization algorithms such as the non-negative matrix factorization (NNMF)[79,80]. While the extracted MSs may be regarded as modules of muscle coordination utilized by the CNS for movement construction[2–4,18], it is often unclear, from the structures of the MSs alone, whether their recruitment consistently executes certain kinematic or biomechanical functions associated with the intended motor behavior. Here, to overcome this limitation, we concurrently extracted both the MSs and the synergistic JAC patterns consistently associated with the activations of the same MSs from the combined, time-synchronized EMG-JAC data. Each extracted synergy consists of time-invariant components across both the 3 JAC dimensions of the hip, knee, and ankle joints (i.e., the KS) and the 14 recorded muscles (i.e., the MS). It can hence be referred to as a KMS, a vector in $(3 \times 3) + 14 = 23$ dimensions with 9 KS components and 14 MS components (Fig. 2B).

To identify KMS, we note that while the MSs are constrained to be non-negative because rectified EMG are never negative by definition, the KSs are unconstrained in sign because joint angles can change in multiple directions, such as extension and flexion. Thus, NNMF is not sufficient for KMS extraction. To this end, we utilized the mixed matrix factorization algorithm (MMF) proposed by Scano et al.[36]. As an extension of NNMF based on gradient descent, MMF extracts a low-dimensional spatial matrix ($W$) and a time-varying coefficient matrix ($C$) from a data matrix ($X$) so that $X \approx WC$. Alternatively,

$$x(t) \approx \sum_{i=1}^{n} w_i c_i(t) \tag{1}$$

where $n$ is the number of KMS extracted. Unlike NNMF, MMF imposes different constraints on different components of $W$ during extraction. Specifically, the first $k$ rows of $W$ representing the KS are left unconstrained, allowing for positive and negative values, while the remaining $m$ rows representing the MS are constrained to be non-negative. With these constraints, each extracted KMS from MMF would contain $k$ unconstrained components, followed by $m$ non-negative components (i.e., $w_i$ with $w_{ij} \in \mathcal{R}$ for $j = 1, \ldots, k$, and with $w_{ij} \in \mathcal{R}^+$ for $j = k+1, \ldots, k+m$).

In MMF, decomposition of $X$ into $W$ and $C$ is realized by minimizing the cost function:

$$\varepsilon = \frac{1}{2} \sum_{s=1}^{S} \left[ ||x(s) - Wc(s)||^2 \right] + \frac{1}{2} \lambda ||W||^2 \tag{2}$$

where the first term is the squared reconstruction error and the second a regularization term that enforces minimum norm solutions. Taking the gradient of the error with respect to components of $W$ and $C$, the following iterative update rules can be derived (with learning rates $\mu_w = 0.001$, $\mu_C = 0.001$, and the weight for regularization $\lambda = 1$):

$$\Delta W = -\mu_w \frac{\partial \varepsilon}{\partial W} = \mu_w \left[ (X - WC)C^T - \lambda W \right] \tag{3}$$

**Table 2 | Number of gait cycles collected per subject across stages**

| | | Infant | | | | | | | |
|---|---|---|---|---|---|---|---|---|---|
| Number | Gender | CH1 | | CH2 | | CH3 | | CH4 | |
| | | Left | Right | Left | Right | Left | Right | Left | Right |
| 1 | F | 19 | 20 | 18 | 22 | 19 | 20 | 9 | 13 |
| 2 | F | 0 | 0 | 23 | 28 | 20 | 32 | 13 | 17 |
| 3 | M | 4 | 1 | 32 | 26 | 57 | 57 | 21 | 22 |
| 4 | M | 3 | 8 | 48 | 43 | 34 | 27 | 25 | 29 |
| 5 | F | 5 | 8 | 15 | 21 | 7 | 9 | 27 | 29 |
| 6 | M | 13 | 16 | 0 | 0 | 39 | 40 | 9 | 6 |
| 7 | F | 20 | 16 | 9 | 9 | 3 | 4 | 40 | 38 |
| 8 | F | 26 | 23 | 12 | 13 | 13 | 13 | 36 | 37 |
| 9 | F | 16 | 25 | 23 | 20 | 28 | 31 | 44 | 39 |
| 10 | M | 15 | 20 | 19 | 21 | 26 | 25 | 12 | 13 |
| 11 | F | 28 | 27 | 18 | 17 | 0 | 0 | 33 | 34 |
| Mean ± SD | | 15 ± 9 | 16 ± 8 | 22 ± 11 | 22 ± 9 | 25 ± 16 | 26 ± 16 | 24 ± 13 | 25 ± 12 |

| | | Adult | | | | | Elder | | |
|---|---|---|---|---|---|---|---|---|---|
| Number | Gender | Independent walking | | | Number | Gender | Independent Walking | | |
| | | Left | Right | | | | Left | Right | |
| 1 | M | 12 | 11 | | 1 | M | 4 | 3 | |
| 2 | M | 20 | 12 | | 2 | M | 9 | 10 | |
| 3 | F | 6 | 6 | | 3 | F | 7 | 5 | |
| 4 | F | 20 | 10 | | 4 | F | 29 | 23 | |
| 5 | M | 15 | 12 | | 5 | F | 24 | 20 | |
| 6 | F | 11 | 13 | | 6 | F | 7 | 14 | |
| 7 | M | 13 | 19 | | 7 | M | 16 | 16 | |
| Mean ± SD | | 14 ± 5 | 12 ± 4 | | Mean ± SD | | 14 ± 10 | 13 ± 7 | |

$$\Delta C = -\mu_C \frac{\partial \varepsilon}{\partial C} = \mu_C W^T (X - WC) \qquad (4)$$

$R^2$ values (determinant coefficients) were used to measure how well the reconstructed data matrix ($WC$) approximated the original input matrix, and a threshold of $R^2 = 0.8$ was set as a termination criterion for the algorithm.

To extract KMS from our data, we concatenated multiple preprocessed gait cycles of JAC and EMG for each limb and subject, forming a data matrix with 9 rows representing kinematic trajectories, and 14 rows representing EMG. To initialize the MMF, we first utilized the NNMF to extract muscle synergies from the EMG alone. These NNMF-derived muscle synergies were regarded as prior knowledge with information on the structure of the MS of KMS[36]. They were then combined with randomized kinematic components to form the initial estimate of KMS for MMF. By incorporating the prior MS, which captured the variability present in the EMG signals, we ensured a qualified starting point for the algorithm that should lead to an optimized solution with less iterations. Indeed, we observed that the MSs of the extracted KMS were usually very similar to the prior MS. To avoid convergence onto local minima, we repeated the MMF extraction 100 times. For each repetition, the algorithm proceeded until the $R^2$ reached 0.8.

The number of KMS extracted ($n$) was determined by the number of muscle synergies in the prior MS, which in turn was identified by the minimum number of synergies needed for achieving an EMG reconstruction $R^2$ of $\geq 0.8$[81]. For each subject at each walking stage, once $n$ was determined, all KMS solutions at $n$ KMSs with EMG-JAC reconstruction $R^2 \geq 0.8$ from among the 100 extraction repetitions were selected, and the KMSs from this selection were $k$-means clustered, with the number of clusters determined by the number that yielded the maximum silhouette value. The centroids of the clusters were then regarded as the representative KMS set for the subject. Instead of directly selecting the solution with the

highest $R^2$, this clustering approach ensures the robustness and reliability of the extracted KMSs. The KMS centroids were then $l^2$-normalized for downstream analysis.

To derive the AC of the KMS ($c_i(t)$ in Eq. (1)), we utilized the KMS centroids from $k$-means to find the AC that reconstructed the concatenated JAC and EMG for each gait cycle. The KMSs were kept fixed during this process while the ACs were updated iteratively from randomized initializations using Eq. (4). This updating was repeated 50 times for each gait cycle. In each repetition, updating was terminated when the EMG-JAC reconstruction $R^2$ reached 0.8, or when the change in $R^2$ was $<10^{-4}$ for 20 iterations, or when the number of iterations exceeded 30,000. The AC from the repetition that yielded the highest $R^2$ was selected as an optimal one for each gait cycle for downstream analysis.

### K-means clustering

At each stage, we applied $k$-means clustering to sets of KMS ($R^2 \geq 0.8$ from 100 extraction repetitions) to derive representative sets for each limb of each subject, to AC across multiple gait cycles to quantify the major activation patterns for each limb and subject (Fig. 8A), and to KMSs from the subject cohorts to derive representative clusters at the populational level (Figs. 4A, S4, S5 and S8). The number of clusters considered ranged from 2 to 20. To calculate the similarity between KMSs and between KSs, we used cosine similarity, which performed better than the Pearson's correlation coefficient ($r$) due to the presence of negative values within KMS and KS. For AC and MS, however, we used Pearson's $r$ to measure similarity. $K$-means was performed using the kmeans function in MATLAB with 20 repetitions and a maximum of 500 iterations in each repetition. For each number of clusters, for all input data points, we calculated the silhouette value ($sv$) (silhouette.m in MATLAB), which measures how similar a given data point is to the other data points in its own cluster when compared to data points belonging to the

other clusters, with higher values indicating better fits. When computing the representative KMSs and ACs for each limb of each subject at each stage, we ensured the robustness of each resulting cluster by requiring that the proportion of elements with $sv \leq 0.2$ (indicating poor fit) was less than 15%[6]. If this criterion was not met, the cluster was then removed. For the remaining clusters, elements with $sv \leq 0.2$ were removed to improve the overall clustering quality. For each number of clusters, we then recalculated the average silhouette values for all clusters and re-derived the centroids. The optimal number of clusters was selected based on the one that yielded the highest $sv$, indicating the best overall fit. As for computing the representative clusters for each subject cohort at each stage, either based on MS similarity (Figs. 4A and S4) or based on KS similarity (Fig. S5) and KMS similarity (Fig. S8), we did not remove any KMS or cluster, and allowed multiple KMSs from the same subject to be assigned to the same cluster.

### KMS and AC comparisons

To characterize the developmental changes of KMS and AC, it is necessary to compare their patterns and profiles across the walking stages to identify any potential tendencies or trends.

**KMS matching.** Before comparing the KMS sets of any two walking stages, every synergy in one set must first be matched to its corresponding synergy in the other set so that all the paired-up synergies could be compared. To establish this correspondence between any two stages, we identified the matching pattern that maximized the overall between-stage KMS similarity across the matched KMS pairs. Specifically, we considered all possible one-to-one matching patterns and selected the pattern that yielded the highest total correlation coefficient ($r$) across pairs. If the numbers of KMSs in the two sets were unequal, after the first round of KMS pairing, the unmatched KMSs in the larger set were paired to their most similar vectors in the other set. After this matching, the similarity between any two KMS sets can then be quantified by the average $r$ values across all matched vector pairs.

To unambiguously establish KMS correspondences across all four stages, the KMSs of CH1 and CH2 were first matched as described above. Then, KMSs of CH3 were matched to those of CH1 and CH2 separately, thereby generating two average $r$ values. The matchings of CH3 to both CH1 and CH2 were then dictated by the matching that yielded the higher of the two $rs$ (i.e., if CH3 is more similar to CH1 than to CH2, CH3 KMS is matched to its corresponding CH1 KMS). Similarly, the KMSs of CH4 were first matched to those of CH1, CH2, and CH3 separately, and the matching that produced the highest $r$ dictated the matching of CH4. After these alignments, the KMSs for each infant could then be compared across the four walking stages.

**Between-stage KMS variability.** We initially examined the changes in KMS between two adjacent developmental stages (Fig. 2B). After aligning the KMSs across the four stages, we proceeded to measure three types of synergy similarities ($SSI$)—KMS, KS, and MS similarity—by calculating the correlation coefficient ($r$) of the corresponding KMS, KS, or MS components between any two consecutive stages (Fig. 2C). We likewise measured the changes of KMS, KS, or MS between CH1 and CH4 (Fig. S1B). In addition, to quantify the rate of change of these components over time, we divided the between-stage synergy dissimilarity ($1 - SSI$) by the exact time that elapsed between the two stages (Fig. S3A). To further examine whether the above changes in the components may themselves be age-dependent, we linearly regressed the synergy similarity and rate-of-change measures against the between-stage increase in the infants' age (Figs. 2D and S3B), represented by the logarithm (base 10) of the number of days that elapsed between two adjacent stages. The data points with a duration of zero day were excluded from the regression. We investigated whether the between-stage KS similarity could be associated with between-stage MS similarity by regressing the KS similarity on the MS similarity for each stage transition (Fig. 5A).

**Between-limb KMS variability (gait symmetry).** To investigate the potential changes in gait symmetry as the infants learned to walk, we first

calculated the similarity of KS and MS between the two lower limbs at each stage of development (Fig. 3A, B). After the KMS of the two limbs were matched with the highest $r$ (see "KMS Matching" above), we calculated the bilateral KS and MS similarities between the matched synergies. We regressed the bilateral KS and MS similarities on the subject's age (in logarithmic scale) with a quadratic polynomial (Fig. 3C, D). For each walking stage, we also regressed the between-limb KS similarity on the between-limb MS similarity (Fig. 5B). For the AC, to assess the bilateral AC correlation, we calculated the correlation between each cycle of the left limb with each of the right limb and then calculated the mean correlation over all pairs (Fig. 8B, Gait Symmetry), which indicates the overall correlation of AC between the two lower limbs.

**Within-cluster subject variability of KMS.** Recent results suggest that the MS patterns may exhibit higher stability than KS patterns during development[28,46]. To examine whether stable MSs may be associated with variable KSs among subjects, at each walking stage, we grouped the KMSs of all subjects into clusters by performing $k$-means over only the MS components of the KMSs, so that within each cluster, the extent of KS variability associated with a set of similar MSs could be revealed (Fig. 4A). Inter-subject KMS variability was quantified by calculating the SD of KMSs over the subjects within each cluster. Specifically, within-cluster SD values were computed for each muscular and kinematic component, and then averaged across the 9 JAC and the dominant muscular components for each cluster to derive the within-cluster SD of KS and MS, respectively. Three clusters of KMSs were identified. For each cluster, muscles with the most contributions to the MS vector were selected as the cluster's dominant muscles. For cluster 1, these dominant muscles were MG, LG, Sol, and Ham; for cluster 2, TA, VL, VM, RF, Ham, AL, TFL, and GM; for cluster 3, TA, ES, ExtO, and LatDor. There were 6 clusters from both limbs for each stage, and we included the within-cluster KS SD and MS SD of these 6 clusters for a regression of the former variable on the latter (Fig. 5C). Besides, we compared the within-cluster KS SD with MS SD across clusters and stages (Fig. 4B). We likewise measured the within-cluster subject variability of MS by performing $k$-means clustering over only the KS components to observe the variability of the MSs associated with similar KSs (Fig. S5). Here, no dominant muscular components were identified for each cluster. We averaged the SD over the 14 MS components to derive the within-cluster MS SD, and then compared it with within-cluster KS SD for each cluster at each stage.

**Between-cycle AC variability (gait stability).** To quantify the changes in inter-cycle gait variability throughout development, we calculated the $r$ between the AC of each gait cycle and the averaged AC of all cycles (Fig. 8B). For each KMS of each subject, we then obtained the mean $r$ over the gait cycles.

**Convergence of AC pattern (complexity and pattern stability).** Following the approach of Sylos-Labini et al.[6], we investigated whether the ACs of the gait cycles of a KMS may converge, over the developmental stages, towards specific patterns that are stably employed across gait cycles only in that KMS. For each subject, limb, and stage, we first identified the major AC patterns by performing $k$-means clustering on all gait-cycle ACs of KMSs (see section 4 of $K$-means Clustering, Fig. 8A). A greater number of clusters indicated a higher complexity of temporal activation patterns observed in the subject. Then, for each KMS of each stage, we calculated the proportion of gait-cycle ACs grouped to each cluster, and derived a vector that described the distribution of the gait-cycle ACs over the clusters (Figs. S7 and 8B). We further calculated the sparseness of this vector to quantify the spread of this distribution pattern[82]:

$$sparseness(x) = \frac{\sqrt{n} - \left(\sum |x_i|\right) / \sqrt{\sum x_i^2}}{\sqrt{n} - 1} \qquad (5)$$

## Table 3 | Muscle-tendon parameters in the calibrated NMS model

| Parameter | Interpretation |
|---|---|
| c1 | First recursive coefficient for estimating past neural activation from muscle excitation |
| c2 | Second recursive coefficient for estimating past neural activation from muscle excitation |
| ShapeFactor | Nonlinear factor describing the transformation from neural activation to muscle activation |
| ActivationScale | Scaling coefficient for muscle activation |
| OptimalFibreLength | Fiber length at which maximum activation occurs |
| PennationAngle | Pennation angle of the fiber at maximum activation |
| TendonSlackLength | Tendon length when the tendon is slack (not under tension) |
| MaxContractionVelocity | Maximum velocity at which a muscle fiber can contract |
| MaxIsometricForce | Maximum force a muscle can generate during isometric contraction |
| StrengthCoefficient | Multiplicative factor applied to max isometric force for muscles within the same group |

where $x_i$ represents the proportion of ACs from KMS $x$ grouped to cluster $i$, and $n$ is the number of clusters. Higher sparseness values mean more gait-cycle ACs are disproportionately assigned to particular cluster patterns at a developmental stage, thus indicating more stability in the AC patterns across cycles. For instance, if the ACs from the first KMS are all grouped to the same cluster, its sparseness value is 1. To summarize the AC pattern stability defined as above for each stage, limb and subject, the sparseness values across all KMS vectors were averaged.

**Muscular co-contraction**. To assess whether muscular co-contraction decreased during development (Fig. S2), we quantified the sparseness of MS for each KMS using Eq. (5), with $x_i$ being $i^{th}$ muscle component of synergy $x$, and $n$, the number of MS. To obtain an overall measure of sparseness, for each limb, subject, and walking stage, we calculated the mean sparseness across all KMSs. Statistical analysis was performed to reveal any significant change of sparseness during development.

**System chaos of AC**. Given the recent finding that neural commands may exhibit changes in their dynamics during infant development, especially when new activating patterns emerge[6,29], we also examined how the stability of the system dynamics embedded within the temporal AC changed across the walking stages. We hypothesize that infants exhibit a more stable, less chaotic neural control of the muscle synergies as they become capable of walking with more robustness and balance. To investigate this hypothesis, we utilized the Maximum Lyapunov Exponent (MLE) to quantify the system chaos and local dynamic stability of the AC (Fig. 8C). The MLE is a mathematical measure that characterizes the rate of separation between infinitesimally close trajectories in the presence of small perturbations. Larger MLE values indicate lower dynamic stability. For each KMS, we computed the MLE of AC for each gait cycle and then calculated the mean value over cycles. The MLE was calculated using the MATLAB function lyapunovExponent.m. The mean MLE values for each limb, subject, and walking stage were then collected for statistical analysis.

### Building neuromusculoskeletal models for infants across developmental stages

Our observation that even similar MSs could be associated with variable KSs led us to consider whether this KS variability might be attributed to differences in limb biomechanical properties across subjects and developmental stages. To evaluate the potential impact of limb biomechanics on KSs, we first quantified each subject's limb biomechanical properties by constructing a NMS model for each infant at each walking stage, allowing us to track the changes across longitudinal development. If the biomechanical parameters derived from the NMS models demonstrate substantial variability across stages, the observed KS variability may then be attributed to differences in both MSs and the relevant biomechanical properties. The

simulated NMS models incorporated ten muscle-tendon parameters (Table 3) characterizing each muscle's musculotendon physiology, as well as activation and contraction dynamics. The modelling process comprised four steps. First, we pre-processed the experimental EMG and kinematic data to ensure data quality and reliability. Second, by integrating these individual biomechanical and kinematic data, we constructed a personalized NMS model with an initial set of ten muscle-tendon parameters. Third, we refined the muscle-tendon parameters through calibration with experimental EMG data from the subjects' walking trials. Fourth, the robustness of the calibrated model was then assessed by comparing the model-predicted muscle activations with experimental EMGs from walking trials not used for parameter calibration.

**Step 1: data preprocessing**. We utilized the MATLAB software package MOTONMS (MATLAB MOtion data elaboration TOolbox for NeuroMusculoSkeletal applications, version 2.2)[83] to preprocess the raw data obtained from the Vicon motion capture system. The package's C3D2MAT module was first used to transform the Vicon data stored in c3d files into MATLAB-readable mat files. Then, the MOTONMS's Data Processing module was utilized to preprocess the kinematic and EMG data in the dynamic trials to ensure that both data modalities were formatted appropriately and stored in suitable files (.trc files for kinematic trajectories of joints and limbs; mot for EMG) for subsequent analyses. The third step, called Static Elaboration, was identical to the second except that it was implemented for the static trials. By employing MOTONMS, we transformed the raw motion capture data into formats compatible with OpenSim and enabled their further analysis in a MATLAB-based environment.

**Step 2: NMS model construction**. Next, we utilized OpenSim (version 3.3)[84] to generate personalized NMS model for each infant at every developmental stage. The first step involved scaling the standard adult model (gait2392_simbody) to create a customized model for each infant. This scaling took into account the infant's mass, the marker attachment protocol, and the marker trajectories obtained from an experimental static trial. The infant's mass was measured by a weighing scale after each recording session. To simulate and optimize the infant's walking patterns, we employed inverse kinematics and inverse dynamics based on the marker trajectories recorded during the walking trials. Inverse kinematics estimated the joint angles and positions from the recorded marker trajectories, while inverse dynamics calculated the joint moments. Then, we performed static optimization to minimize the sum of squared muscle activations that could execute the calculated joint moments. Additionally, we calculated the lengths of muscle-tendon units (MTU) and the moment arms of each muscle by running OpenSim's muscle analysis, which provided initial estimates of the musculoskeletal properties within the NMS model.

**Step 3: EMG-Informed calibration of muscle-tendon parameters.** We further refined the personalized NMS models by calibrating muscle-tendon parameters using experimental EMG data from walking trials, employing the CEINMS toolbox (Calibrated EMG-Informed NMS Modelling Toolbox, version 0.10.0)[44,53]. CEINMS requires trial-based EMG input in mot file format (exported from MOTONMS), as well as joint moments, muscle moment arms, and MTU lengths estimated by OpenSim. The calibration process aims to optimize muscle-tendon parameters to accurately represent muscle activation and contraction dynamics during walking (Table 3). This is achieved by comparing the joint moments estimated by the model with those computed using OpenSim's inverse dynamics module. The toolbox iteratively adjusts the muscle-tendon parameters using a simulated annealing optimization algorithm, minimizing the error between estimated and experimental joint moments (i.e., $\alpha = 1, \beta = 0, \gamma = 0$ in Eq. (6); see below). The optimization converges when the change of joint moment error over four consecutive iterations is less than 0.001 (Nm)$^2$ or when the maximum number of iterations (2,000,000) is reached. For each infant and each developmental stage, EMG data from five walking trials were used for parameter calibration. The default parameter values and algorithmic settings from the CEINMS examples were used to initiate calibration across all subjects. Importantly, this calibration imposes no restrictions on the number of degrees of freedom or MTU, allowing the model to be tailored to individual biomechanical profiles.

**Step 4: evaluating model accuracy.** Following calibration, the accuracy of the refined model was evaluated by comparing the model-predicted muscle activation with experimental EMG in walking trials not used in the calibration. For this evaluation, activations of the model muscles were predicted, using the EMG-assisted mode of CEINMS, by minimizing a cost function ($F_{obj}$) that combines the error between the estimated and experimental joint moments, the error between the estimated and experimental muscle activations, and penalization of excessive muscle activations:

$$F_{obj} = \alpha * \sum_{k \in DOFs} \left( \tau_k - \widetilde{\tau}_k \right)^2 + \beta * \sum_{j \in MTUs} \left( e_j - \widetilde{e}_j \right)^2 + \gamma * \sum_{j \in MTUs} \left( e_j^2 \right) \quad (6)$$

where $\tau_k$ is the moment of joint $k$ as estimated by CEINMS, $\widetilde{\tau}_k$ is the experimental moment of joint $k$, $e_j$ is the estimated activation for MTU $j$, $\widetilde{e}_j$ is the experimental activation for MTU $j$, and $\alpha, \beta, \gamma$ are weights for the three terms, set here as $\alpha = 1, \beta = 2$, and $\gamma = 3$. In essence, this optimization uses the muscle-tendon parameters from step 3 to find a muscle pattern, one that is as close to the experimental EMG as possible, that at the same time approximates the experimental joint moment. A good alignment between the predicted muscle activation and experimental EMG implies that the model parameters are accurate.

To compare the predicted muscle activation and experimental EMG, we first aligned the two traces temporally with a time delay that minimized their cross-correlation. Model accuracy was then quantified by computing the Pearson correlation coefficient between the time-aligned traces. For this evaluation, 1 to 3 independent walking trials were used per subject per stage. For each muscle, we reported the Pearson correlation coefficients across all validation trials and all 11 infants (Fig. S6A) as well as the proportion of trials in which the Pearson correlation $p$-value was <0.05 (Fig. S6B).

**Muscle-tendon parameters calibrated in NMS model.** In CEINMS calibrations, the activation dynamics of a muscle is modeled as the muscle's twitch response, in which parameters c1 and c2 (Table 3) quantify how quickly the muscle responds to neural signals, contracts, and then relaxes back to its resting state. With the provided EMG, this calibration estimates the past neural activations with a linear second-order recursive system. Parameter ShapeFactor is the non-linear factor that describes the relation between neural excitation and muscle activation. Parameter ActivationScale serves as a coefficient that scales the muscle activation. Based on the estimated muscle activation, fibers would then show contraction dynamics that generate forces. Within this step, parameters OptimalFibreLength, PennationAngle, and MaxContractionVelocity are then estimated under the maximal activation, and then used to normalize the fibers' force-length and force-velocity relationships. Strain of the tendon is modeled to distribute and transmit the generated forces to the associated bone and hence to realize joint movements. Parameter TendonSlackLength of the tendon is estimated to formulate this force-strain relationship. Another parameter, MaxIsometricForce, is also estimated for each muscle. For muscles with close anatomical locations and similar biomechanical functions, their MaxIsometricForce values were scaled as a group by another parameter, StrengthCoefficient, to reduce computational complexity. For c1, c2, ShapeFactor, ActivationScale, and StrengthCoefficient, there are no specific physical units. The ranges for these parameters were initially defined as absolute values prior to calibration, following the default settings of the CEINMS toolbox. Parameters TendonSlackLength and OptimalFibreLength are measured in meters (m), with their ranges set by default as multiplying factors applied to their initial values for each subject. PennationAngle is measured in radian (rad), MaxContractionVelocity in m/second, and MaxIsometricForce in Newton (N).

With the above steps, for each subject at each walking stage, we created a calibrated, personalized, and quantitative NMS model, which we used to investigate the changes in the muscular and biomechanical parameters as the infants progressed through their walking stages. We regressed each of the NMS model-derived muscle-tendon parameter on the infants' age (10-based logarithm of days) at the 4 stages to identify which ones exhibited significant age-related changes (Fig. 6). We also regressed their bilateral differences on the age (Fig. 6E) to reveal the ones that may affect the MS-KS relationship through their effects on the bilateral KS similarity.

To further understand the sources that contributed to the observed KS variability, we regressed the within-cluster subject variability of KS on the within-cluster subject variability of both the muscle-tendon parameters and the MS, across all KMS clusters of both limbs from all infantile stages. In this regression, for each KMS cluster (Figs. 4A and S4), within-cluster subject variability of MS was quantified by the average within-cluster SD of the dominant muscles in the cluster, and KS variability, by the average within-cluster SD of the 9 JAC components (Fig. 7). Within-cluster subject variability of each parameter in each cluster was quantified by first computing its SD, for each muscle, across those subjects represented in that KMS cluster, and then averaging the SD over the cluster's dominant muscles. For cluster 1 of the 3 KMS clusters identified from the infants' single lower limbs, the dominant muscles were MG, LG, Sol, and Ham; for cluster 2, TA, VL, VM, RF, Ham, AL, TFL, and GM; for cluster 3, TA, ES, and ExtO. LatDor was not used here as this muscle was not included in the NMS model. For each muscular parameter, variability values from 6 clusters from both limbs of four infantile stages were considered, and thus $6 \times 4 = 24$ values were included in each regression. To examine their relationship at each stage, we also applied this regression to variability values from the 6 clusters at each infantile stage. Through these analyses, we identified the muscle-tendon parameters that exerted the most effect on within-cluster variability of KS, and should therefore have the most influence on MS-KS association.

## Statistics

As we compared the metrics across 4 time points for the infant cohort, specific statistics for these repeated measurement data were demanded. We first examined the assumption of data normality using the Kolmogorov–Smirnov Test, and then examined data sphericity using the Mauchly's test. If these assumptions were met, we employed the repeated measures Analysis of Variance (ANOVA), or the non-parametric Friedman test if not met, to test for statistical significances of the differences across time points. If a significant difference was found, we further performed a post hoc

multiple comparison test with Bonferroni correction for identifying specific group pairs that exhibited significant differences. When we separately compared the metrics of each infantile stage with the Adult and Elder stages, we employed one-way ANOVA and Bonferroni correction post hoc for normally distributed data, or the Kruskal–Wallis test with the pairwise Wilcoxon rank-sum test post hoc for non-normal data.

To examine the correlation between any two vectors, we computed the Pearson correlation coefficients ($r$), which indicates the strength and direction of the correlation. To statistically compare the $r$ values derived from comparisons of KSs, MSs, and KMSs, the Fisher's z transformation[85] was first applied to normalize the distributions of $r$ values. When employing a linear regression analysis to investigate the relationship between factors, we reported the $p$-values of the F-statistics to assess whether the fitted model differed significantly from a constant (null) model, as well as the $p$-values of the t-statistics to determine whether each factor coefficient was significantly different from zero. The significance level for hypothesis testing was set at 5%. All statistical analyses were performed using MATLAB.

## Data availability
All data collected and analyzed in the study are available to any researcher upon request sent to the corresponding author.

## Code availability
All code developed in the analysis are available to any researcher upon request sent to the corresponding author.

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

## Acknowledgements
This work is supported by The Hong Kong Research Grants Council (14114721, 14119022, N_CUHK456/21, R4022-18F, STG1/M-401/24-N), CUHK RSFS (3133184), and CUHK Research Committee (Impact Case C7).

## Author contributions
J.Y.L., S.C.W.H., C.Y.C., R.T.H.C., V.C.K.C. contributed to the study design; S.C.W.H., J.H.Z., Z.Y.S.C., X.Y.G., B.R.H., K.Y.S.L., R.H.M.C., R.T.H.C. contributed to the collection of experimental data; J.Y.L., K.D.F., V.C.K.C. contributed to the analysis of data; J.Y.L., V.C.K.C. contributed to writing of the manuscript; V.C.K.C. supervised all aspects of this work.

## Competing interests
The authors declare no competing interests.
