## [Transparent Peer Review file · Communications Biology]

Functional Symmetrization of Neuromotor Modules During Locomotor Development in Human Infants

Corresponding Author: Professor Vincent Cheung

Version 0:

Reviewer comments:

Reviewer #1

(Remarks to the Author)

This paper investigates the development of motor synergies in early childhood by integrating the parallel maturation of biomechanical properties. This approach addresses an important gap in the literature, as previous work has primarily focused on neural and muscular aspects without fully accounting for the evolving biomechanics of the developing body. The authors present a highly valuable longitudinal dataset of muscular and kinematic data collected across four infantile walking stages. Using a rigorous methodology to extract kinematic-muscular synergies (KMSs), they propose a range of analyses aimed at characterizing the co-evolution of muscular control and limb biomechanics.

The main conclusion is that inter-individual variability in kinematic synergies cannot be fully explained by variability in muscular synergies alone. The authors argue that developmental changes in biomechanical properties—and their inter-individual variability—play a key role in shaping kinematic outputs. Additionally, they report distinct developmental trajectories for the symmetry of muscular and kinematic synergies across the lifespan, with kinematic symmetry increasing earlier than muscular symmetry. The authors suggest that these differences in timing may be explained by developmental changes in the symmetry of biomechanical parameters.

While I highly appreciate the longitudinal dataset and the integrative nature of the work—bridging muscular synergies, kinematic synergies, and biomechanical modeling—I have several methodological concerns detailed below.

Regression linking KS variability to MS and biomechanics (Line 293-314):

While the link between KS variability, MS variability, and biomechanical parameters appears to be one of the central conclusions of the study, some aspects of the regression analysis are not entirely clear.

First, the regression is based on variability indices computed within KMS clusters (Fig. 4A), but it is not clear whether all subjects could be included in this analysis, and if so, whether each synergy from a given subject was necessarily assigned to a different cluster, or whether multiple synergies from the same subject could end up in the same cluster (in which case, how did the authors proceed to reattribute KMSs to the good clusters). Clarifying this seems important to assess how representative the clusters are of the full cohort, and would itself provide valuable insight into inter-individual variability within the cohort.

Second, the main conclusion does not appear to be fully coherent with the regression models displayed above the plots in Fig. 7 (specifically, in Fig. 7A, the effect of MS variability appears to be not significant ($p = 0.278$), and in Figure 7B, only marginally significant ($p = 0.048$), especially in a context of multiple comparisons). As such, it is unclear which p-value the authors refer to in the summary sentence (" $p < 0.05$ ") — is it for the full model, or for the individual predictors? Can the authors clarify which statistical results exactly support the claim?

Additionally, only two biomechanical parameters (TendonSlackLength and StrengthCoefficient) are shown. It would be helpful to either provide results for all tested parameters, or to clarify the rationale behind the selection of these two. It would also be useful to comment on the choice to test each biomechanical parameter in a separate regression model, and whether a joint model was considered.

Finally, the 3D scatter plots in Figure 7 are difficult to interpret visually. It is unclear whether the individual data points lie above, below, or on the regression plane, making it hard to assess the quality of the fit. If the graph can't be improved in 3D, I suggest adding 2D projections (MS and biomechanical parameters).

Complexity and stability of activation coefficients over the lifespan (Line 315-331 – 740-759):

How complexity was computed lacks methodological details (maybe a methodological section could be added with this keyword, as it is the case for Gait Stability and Pattern Stability?). Specifically, I am not sure I understand which input was given to the clustering. Isn't the number of AC per subject the same as the number of KMS? If so, does this mean the AC clustering is based on pooled data with uneven numbers of ACs across individuals and stages? Is this complexity index different from the number of KMS?

Moreover, if I'm not mistaken, how many gait cycles were available for each infant and each stage is not reported. Was this number fixed, including across stages? If not, how did the authors ensure comparability for complexity and stability analyses?

Limitations

Although the study is methodologically rich, potential limitations are not currently discussed. It would be valuable to include a reflection on how certain methodological choices may have influenced the measurement of variability and symmetry, particularly: (i) the fact that children were held by their parents rather than by a standardized experimenter during stepping tests, (ii) the methodological choice of analyzing functional developmental stages despite the wide age ranges within each group (including how these functional stages were defined), and (iii) the use of the O'Connor algorithm for gait event detection, which was developed for adult gait and may not be appropriate for the less stable and more variable gait patterns observed in infants.

Minor comments

Line 42-45: This sentence suggests a strong established relationship between muscle synergy asymmetry, biomechanical asymmetry, and resulting kinematic symmetry. While the study presents interesting evidence that all three factors evolve during development, it remains unclear whether the evolution of MS symmetry explicitly compensates for limb biomechanics to drive KS symmetry. I would recommend adjusting the abstract to align more closely with the cautious and well-balanced interpretation given in the discussion section (line 449-466).

Line 184-197: The comparison of Pearson correlation values across KS, MS, and KMS might provide useful insights into the relative developmental stability of these components, but I wonder if they are comparable given the differences in dimensionality between the vectors (KS: 9, MS: 14, KMS: 23). Could the authors comment on whether vector dimensionality might influence Pearson correlation values and affect comparability?

Line 335: I'm not sure I understand the term "correlation between gait-cycle pairs". In the Methods, it is stated that correlations were calculated between each gait-cycle AC and the average AC across cycles. Are the authors referring to the same computation?

Line 394-395: Could the authors clarify which analysis (or/and how Fig.4B) demonstrates consistency in dominant muscles across individuals?

Line 591: The reported age ranges for stages appear inconsistent with the values in Table 1 (e.g., CH1 includes two subjects younger than 6.3 months).

Figure 3B: the colors of CH2 and CH3 cannot be well distinguished

Figure 4: Clusters are very small and difficult to read. Considering the richness of the dataset—including synchronized EMG and kinematic recordings across multiple developmental stages—it would be valuable to present clearer visualizations of the KMSs across development (with labeled angles and muscles). In the same vein, adding an illustration showing how EMG and joint angle signals evolve across development—prior to dimensionality reduction—would provide helpful context for interpreting the extracted KMSs and for visualizing variability/complexity in the original, observable motor patterns.

Reviewer #2

(Remarks to the Author)

In this paper, the author examines changes in synergy during the developmental period of walking, and argues that synergy changes in order to improve bilateral symmetry in movement. The attempt to clarify the factors behind changes in muscle synergy during development from the data is interesting, and the hypothesis that muscle synergy changes in order to resolve physical changes and movement asymmetry during development is convincing. The idea of using a musculoskeletal model that deals with physical development to investigate the relationship between movement and muscles is also convincing. On the other hand, when it comes to whether the hypothesis was sufficiently verified by the results, I felt that there were some unclear points in the current paper. The following are specific points of questions and problems.

1: Regarding the data on KMS synergy

One of the features of this paper is that it uses a synergy that combines the characteristics of kinematics and muscle synergy, called KMS synergy. I felt that there were two problems or questions regarding the results of this KMS.

* [1.1: Only a part of the KMS synergy is presented] In the paper, only the representative example in Figure 2A is presented as the result of the KMS. To understand how the KMS synergy is structured, I would like to see more KMS synergies, not just the representative example. Can you provide the results for each subject (even only for the infants) as supplementary data? Also, if possible, I think it would be helpful to understand the results in Figure 4A if the authors could attach labels for the data corresponding to Cluster 1, Cluster 2, and Cluster 3 etc. in the cluster analysis.

* [1.2: Joint angle acceleration data for analyzing KMS] The paper uses JAC for all three axes to calculate KMS. Here, I have some questions about this approach. The movements being targeted are mainly composed of joint flexion/extension (x-axis),

and most of the muscle activity being measured is also from muscles that compose this movement. Nevertheless, the data from the y-axis and z-axis are normalized and treated in the same way as the x-axis, in the paper. Is this processing appropriate? As a result of this process, for example, in KMS2 for CH4 in Figure 2A, the z-axis of the ankle is the main component of the activity. Here, I have some doubts about whether this result is appropriate for representing the activity at this stage. If possible, I felt it would be better to verify whether there is any difference in the results when KMS is calculated using only the x-axis data.

2: Changes in asymmetry with age: Regarding Figure 3B

* 2.1: Since KS can take both positive and negative values, and MS is defined as only positive, KS correlations can be both positive and negative, while MS correlations can only be positive. In fact, in Figure 3B, only KS shows negative values. At this point, it is somewhat questionable whether it is fair to compare the values of the correlations between KS and MS. If you can justify this point, please write about it.

* 2.2: The authors have performed a regression analysis of KS and MS using cubic and quadratic functions, respectively, and compared the results at the peak ages, but is it fair to compare the peaks of regressions in different dimensions? It would be better to verify whether the same results are obtained when the order of the regression is the same.

3: Changes in between-subject variability of KS and MS (Fig. 4A, Fig. S5)

* 3.1: I agree with the method used in the paper to evaluate variability by collecting similar patterns as clusters. I would like to know the overall picture of the clusters, such as how many clusters there are (as the number of clusters vary for each subject) and how many KMS synergies belongs to each cluster. As I commented in 1.1, it might be possible to show the overall picture by including all KMS data in the Supplementary and labeling them.

* 3.2: I would like more explanation about Figure S5. The MS results for the data in Figure S5 show the same pattern for all clusters, but did other patterns not belong to any cluster?

4: Interpretation of the musculoskeletal model

In order to investigate the relationship between muscle synergy and kinematics, the paper estimates muscle activity using inverse dynamics and static optimization with a musculoskeletal model based on motion data, and then uses the measured muscle activity to make corrections and examines the body parameters. In interpreting the results, I would like to know the following three points regarding the quality of the results.

* 4.1: To what extent do the estimated (post-correction) muscle activities match the measured muscle activities? Since the estimation is carried out using optimization with a musculoskeletal model, it is desired to indicate the degree of error in the results when considering the quality of the results.

* 4.2: Are the floor reaction forces and the support forces (of CH1 and CH2) taken into account? When performing inverse dynamics, it is possible to calculate torque without including reaction forces, but if force data is not included, the torque estimated with inverse dynamics and the muscle activity estimated from that will be off by that amount. How much consideration is given to the relationship between these forces? If they are not taken into account, it seems necessary to discuss how they might affect the body parameters.

* 4.3: The data for CH2 and CH3 in Number 20 and 24 in Table 1 are the same. Please check whether this is a simple mistake in the description.

In addition, there were the following points that were unclear about how to interpret the results.

* 4.4: Both height and weight have increased by about 1.5 times between CH1 and CH4, and accordingly, muscle length and the required torque have increased. Therefore, the results showing that muscle length and the strength coefficient increase are understandable. On the other hand, is it possible that the optimal fiber length decreases even though the height increases? Are these values normalized in some way? If they are not normalized, then the vertical axis of Figure 6 should include the unit as much as possible.

* 4.5: The change in shape factor is written as one of the characteristics of the results, but it was difficult to get a clear picture from the paper of how shape factor is specifically defined and how it changes with age. I felt that this part needed to be explained. Also, in line with this, I felt that a more detailed explanation was needed for the explanation in P11, Line 416-417: "thus potentially leading to more stability in the force generated across step cycles despite the step-to-step variability in neural activations."

* 4.6: The results of the musculoskeletal model, which the paper claims to show symmetry between the left and right sides, are shown in Supplementary Figure S6, and these results are often referred to in the Discussion as supporting the change in symmetry. Therefore, I felt that this figure should be included in the main text rather than in the Supplementary (not mandatory). Also, if anatomical evidence could be referenced regarding the changes in tendon slack length during development, I felt that the paper's claims would be strengthened (also not mandatory).

Reviewer #3

(Remarks to the Author)

This manuscript offers a novel and valuable contribution to understanding developmental changes in motor control through the analysis of muscle and kinetic synergies. The manuscript investigates how muscle synergies evolve throughout human development, particularly comparing kinetic synergies (KS) and muscle synergies (MS) across different life stages, from infancy to adulthood. The authors claim that KS show higher variability and a more rapid rate of change in early stages compared to MS, which remain relatively stable and become sparser and more independently controlled in adulthood. This manuscript presents an interesting and valuable contribution to understanding motor control development. The study offers novel insights into the differential dynamics of kinetic and muscle synergies, which could influence future research in developmental neuroscience, motor control, and rehabilitation. In addition, the concept of comparing kinetic and muscle

synergies across development is interesting and novel. The conclusions are generally supported by presented evidence. However, ensuring robust statistical correction and clearly articulating methodological choices will significantly enhance the manuscript's strength and reproducibility.

General comments:

1. Lines 590-594: The authors explicitly define four distinct developmental walking stages (CH1: weight-support; CH2: hand-support; CH3: early independent walking; CH4: later independent walking), each characterized by specific chronological and developmental criteria. However, Table 1 highlights critical inconsistencies regarding participant age across these stages: participant 25 has identical ages (0.93 years) reported for both CH1 (weight-support) and CH2 (hand-support). And participants 20, 24, and 27 also show identical ages for CH2 (hand-support walking) and CH3 (early independent walking). This seems to indicate that, at least in these cases, different walking conditions were recorded at the same developmental time rather than representing separate, chronologically distinct stages. Could you please clarify how children were distinguished and assigned to the different walking stages? If this is indeed the case—that is, if some children were assessed in multiple walking conditions simultaneously rather than distinct developmental stages—this should be clearly stated. Importantly, this distinction significantly impacts the manuscript's primary claim of investigating developmental trajectories across four clearly separated walking stages. Therefore, it is crucial that the authors explicitly address and clarify this inconsistency in detail. Specifically, they should explain clearly whether each participant indeed passed through four separate developmental stages, as described in the methods, or if instead they were assessed across multiple conditions within the same developmental stage. This clarification is essential for interpreting the study's findings and validating the conclusions drawn regarding developmental progression in walking stages.
2. Figure 2C and Table 1: Could the authors clarify precisely what is shown in Figure 2C? According to the legend, the X-axis represents the number of days elapsed between two developmental stages for each subject (CH1-CH2;CH2-CH3;CH3-CH4). However, Table 1 reports identical ages for several participants across different stages (e.g., participant 25 has the same age in CH1 and CH2, and participants 20, 24, and 27 have the same age in CH2 and CH3). If the difference in days between recordings was truly zero in these cases, how is it that the smallest value plotted in Figure 2C is 32 days? Were these cases excluded from the regression? Or is the variable on the X-axis representing something different than the actual time between recordings? Please clarify this apparent inconsistency between the figure and the subject data.
3. Given the high variability typically observed in walking patterns among infants and toddlers, the number of young participants included appears relatively low. Additionally, the number of participants included in both the adult and elderly groups (n=7 each) is also limited. Increasing participant numbers or providing clear justification for the sample sizes chosen would strengthen the reliability and generalizability of the findings.
4. Please clearly state the exact criteria used for determining the number of clusters in the k-means clustering analyses (e.g., silhouette scores are mentioned, but explicit thresholds or procedures for selecting cluster number should be elaborated).
5. Statistical analysis: The statistical approaches employed (e.g., Pearson similarity, regression analyses, and sparseness calculations) are appropriate. However: Clarify whether corrections for multiple comparisons were performed. If not, this should be addressed explicitly, considering the multiple statistical tests conducted across life stages.
6. Statistical analysis: Provide more detailed descriptions or justifications for chosen statistical tests in the main manuscript text.
7. Interpretation of Results: The interpretation regarding the stability of MS vs. variability of KS is compelling but could benefit from additional evidence or discussion: are there potential confounding factors (e.g., activity level differences, walking speeds, or environmental influences) that might impact the variability observed in KS?
8. Could the authors comment further on why kinetic synergies might exhibit higher variability relative to muscle synergies?
9. Figures and Supplementary Material: The supplementary figures provided robust additional support for main findings. However, it would be beneficial if some key supplementary results (e.g., Fig. S1, S3, and S5 regarding synergy similarity and rate of change) could be integrated or briefly discussed in the main text to strengthen the core narrative of developmental trajectories.
10. The authors might consider explicitly stating potential future research directions or implications of their findings for clinical applications or theoretical models of motor development.

Minor / Specific Points:

11. Lines 590-591: The authors mention that during the CH1 (weight-support walking) condition, "children walked with weight support from their parents lifting their body from the shoulder." Could the authors specify more clearly the exact method used for providing support? Specifically, were parents instructed to hold the infants around the waist, under the shoulders, or elsewhere, and what specific instructions were given to parents regarding the amount or proportion of body weight to support? Several studies have demonstrated significant effects of the amount of body-weight support on kinematic patterns and muscle synergies, both in toddlers and adults (Pang and Yang, 2000; Dominici et al., 2007; Kerkman et al., 2022; see review by Apte et al., 2018). It is crucial to clearly describe and address this methodological detail for correct interpretation of the data and comparison with previous literature.
12. Data Recording: The manuscript does not specify important details regarding the data collection protocol. In particular, it is unclear how many trials were recorded per participant, what was the size of the recording volume, and how many strides or gait cycles were collected in total. Providing this information is essential to assess the robustness and representativeness of the dataset. Please include these details to improve transparency and reproducibility.
13. Line 608: For transparency and reproducibility, please explicitly specify the anatomical locations where markers were placed on the participants
14. Lines 615-616: Could the authors clarify the rationale for recording muscle activity from the external oblique and latissimus dorsi muscles, given that the primary focus is on lower-body walking patterns? Particularly in the CH1 (weight-support walking) condition, there might be significant muscle-activation artifacts introduced by the researcher's support. Please discuss how this potential issue was addressed or justify the selection of these muscles.
15. Line 616 and line 630: The reference to Fig. 2A in these lines is unclear, as Fig. 2A does not explicitly illustrate the EMG electrode placement or muscle selection, nor does it clearly depict the joint or 3D angles mentioned. Please revise the figure

- reference or provide an appropriate figure that visually clarifies the EMG electrode placements and joint angles analyzed.
16. Line 641: The selected cutoff frequency (50 Hz) for the high-pass filter applied to EMG data appears unusually high compared to standard practices. Could the authors clarify or justify explicitly why such a high cutoff frequency was selected?
17. Line 644-645: The description of applying the Savitzky-Golay filter to remove occasional high-amplitude noisy spikes from the EMG signal is not entirely clear. Could the authors briefly clarify the steps implemented for this process? Specifically, how were high-amplitude spikes identified—was there a threshold or criterion used? Providing a concise but clear explanation of these preprocessing steps would greatly assist readers in understanding and potentially reproducing the methodology.
18. Line 649: The authors apply a fixed electromechanical delay of 50 ms to shift EMG signals. However, this value is based on adult data and may not be appropriate for infants and toddlers, who usually show longer electromechanical delay due to their immature neuromuscular systems. Please clarify why a constant 50 ms delay was used for all age groups, and consider discussing whether age-specific electromechanical delay values would be more accurate.
19. Line 705: Please clarify how many gait cycles were ultimately included per participant in the concatenated matrix used to extract KMS.
20. Figure 1 (section: longitudinal - learn to walk): The image used to represent the 'supported walking' condition could be improved by showing a toddler clearly supported by an adult (e.g., holding hands while walking). This would help distinguish more clearly between supported and independent walking, improving the visual clarity of the developmental stages depicted. Additionally, the figure and data collection section mention 'BodyWeight'—could the authors clarify what specific body weight information was collected and how? This detail does not appear to be reported elsewhere in the manuscript.
21. Figure 2A is very small and difficult to evaluate clearly. Please consider enlarging the figure or improving its resolution to ensure that all elements—especially EMG profiles and joint angles—are clearly visible and interpretable by the reader. Additionally, while Figure 2A illustrates an example from a single infant, it appears that a larger number of strides were recorded for CH1 compared to CH4. It would be helpful if the authors could report the average number of strides (and/or the range) used for the analysis at each developmental stage, as well as for adults and elderly participants. This information is important to assess the representativeness and balance of the dataset across conditions.
22. Lines 184–188 / Figure 2B and Fig. S1A: The manuscript reports statistical comparisons of between-stage synergy similarity values (e.g., KS vs. MS) across developmental transitions, with associated p-values (e.g., $p = 0.039$, $p = 0.023$). However, the specific statistical test used to assess these differences is not stated. Since the comparisons appear to be based on within-subject, paired data (i.e., comparing KS and MS similarity for the same subjects), it is important to clarify whether a paired t-test, Wilcoxon signed-rank test, or another method was used. Additionally, since the similarity values are Pearson correlation coefficients, it is worth noting that these are not normally distributed. A Wilcoxon signed-rank test would be more appropriate unless the authors applied a Fisher z-transformation before performing a parametric test. Please clarify the statistical approach used and consider whether a non-parametric alternative might be more suitable. More broadly, the analysis appears to involve repeated measurements across both synergy type (KS, MS, KMS) and developmental stage transitions (CH1→CH2, CH2→CH3, CH3→CH4) for the same subjects. This suggests that a two-way repeated measures ANOVA or a linear mixed-effects model may be more appropriate to account for within-subject structure and potential interactions between factors. Could the authors clarify the overall statistical strategy used for these comparisons? Were pairwise tests applied independently, or was a multi-factorial model considered? Also, were any corrections for multiple comparisons performed? Explicitly outlining the statistical approach would improve the transparency and robustness of the analysis.
23. Line 363 Discussion : The discussion could be enriched by further exploring possible interpretations of the observed differences between kinetic (KS) and muscle synergies (MS) across development. While the authors describe KS as more variable and changing more rapidly than MS, it may be helpful to consider how these patterns could relate to underlying neural or biomechanical mechanisms. For instance, could KS be more influenced by body dynamics and postural demands, whereas MS might reflect more stable, spinally organized motor modules? Expanding on these possibilities—even as hypotheses—could help contextualize the findings within broader models of motor development.
24. Lines:1013-1014 Figure 1 legend: should be 'weight-supported walking (CH1), 158 hand-supported walking (CH2)'
25. Table 1: For the children in the CH3 (early independent walking) group, please consider adding information regarding their age at walking onset. This detail is important for clearly determining each child's walking experience
26. Information about participant recruitment procedures is missing. Please specify how participants were recruited (e.g., inclusion/exclusion criteria, recruitment setting, consent procedures), particularly for infants and toddlers. This information is important for assessing the generalizability and ethical conduct of the study.
27. Figure 8B: there are multiple lines and statistical comparison that overlap with the axes and plotted data, making the figure somewhat difficult to interpret. The visual clarity is reduced, especially where lines intersect with bars or axis labels. Please consider revising the layout or presentation of the comparisons to enhance readability and improve figure clarity.

Version 1:

Reviewer comments:

Reviewer #1

(Remarks to the Author)

I thank the authors for this thorough revision. I feel that all of my main concerns have been addressed, and I would like to emphasize the strong contribution this work makes to the literature. I only have one remaining issue regarding my very first point.

I understand the methodological choice of not constraining cluster assignment by subject, as it allows for more representative groupings of similar MSs across the cohort. However, in this context, it remains unclear why the metric derived from this clustering (within-cluster SD) is often referred to as “between-subject variability” in the paper (e.g., l226–253, l311–337, Fig. 4, Fig. 7, etc). If multiple synergies from the same subject can fall within a single cluster, this metric seems to capture a mixture of intra- and inter-individual variability. For clarity, I would suggest either adopting terminology that more directly reflects the methodology (e.g., “within-cluster variability” or “between-KS variability”), or explicitly showing how subjects are distributed across clusters to justify the use of “between-subject variability”. In addition, I believe that the paper would be strengthened by explicitly discussing that regressions were performed on cluster-level variability, and therefore that whether KS, MS, and biomechanical parameters vary together at the individual level remains unknown.

Reviewer #2

(Remarks to the Author)

The revised manuscript largely addresses the concerns raised in the previous manuscript. In particular, the modifications to the comparison method in Figure 3 and the revisions regarding fitness consistency in the musculoskeletal model have made the arguments more clearly supported.

However, concerns remain regarding the age-related changes in OptimalFibreLength in Figure 6B, which were also pointed out previously. It seems difficult to accept that optimal fiber length decreases despite increases in height and muscle length due to development. Since sarcomere increases with development, optimal fiber length should increase when considering the muscle alone. Admittedly, as the authors' response noted, pennation angle and tendons do influence this, making it difficult to definitively state that optimal fiber length always increases. Nevertheless, the result that optimal fiber length uniformly decreases with development is hard to believe.

One interpretation I considered for the property observed in Figure 6B is that body support during walking might have influenced it. When walking with body support, muscles may be in a stretched state during large muscle activities like foot contact or lift-off. Therefore, it could be interpreted that optimal fiber length was particularly longer during the early infant stage when body support is substantial. If this is the case, the results in Figure 6B would represent a property distinct from motor control strategies.

I am not entirely confident in the correctness of this interpretation. Furthermore, I do not believe the concerns regarding Figure 6B are significant enough to substantially lower the overall evaluation of the paper. However, I felt the results in Figure 6B require somewhat careful handling. Therefore, I would like to request one of the following from the authors:

1: If the results in Figure 6B are considered correct, I hope the authors can provide a more convincing explanation or cite literature demonstrating a decrease in optimal fiber length with development.

2: If it is difficult to support the results in Figure 6B, I believe it would be better to revise the text to treat this result as less significant results, such as by stating limitations regarding Figure 6B. (In that case, the claims regarding the properties using OptimalFibreLength in Figure 7 may also require some revision.)

Below are minor comments.

* In Table S1, only values where $P < 0.05$ are listed, with the rest left blank. Does this mean the relationship for the blank cells was not evaluated? If they were evaluated, it would be preferable to include all values.

* Line 1060: tk, Line 1062: alpha, beta, gamma, etc., should be italicized but are currently in roman type.

Reviewer #3

(Remarks to the Author)

I thank the authors for their careful revision. They have addressed my comments thoroughly. The manuscript is now improved, and I have no further concerns.

Version 2:

Reviewer comments:

Reviewer #2

(Remarks to the Author)

I appreciate the authors' careful and thorough responses to the points I raised.

In this revision, the authors have addressed all my concerns, particularly those related to the inverse dynamics analysis, through careful and detailed discussion, including the addition of relevant references. These revisions have strengthened the persuasiveness of the authors' arguments.

I have no further concerns.

All line numbers referenced in this rebuttal refer to lines in the redline version of the revised manuscript (i.e., manuscript file with Track Changes).

Responses to Reviewer No. 1

This paper investigates the development of motor synergies in early childhood by integrating the parallel maturation of biomechanical properties. This approach addresses an important gap in the literature, as previous work has primarily focused on neural and muscular aspects without fully accounting for the evolving biomechanics of the developing body. The authors present a highly valuable longitudinal dataset of muscular and kinematic data collected across four infantile walking stages. Using a rigorous methodology to extract kinematic-muscular synergies (KMSs), they propose a range of analyses aimed at characterizing the co-evolution of muscular control and limb biomechanics.

The main conclusion is that inter-individual variability in kinematic synergies cannot be fully explained by variability in muscular synergies alone. The authors argue that developmental changes in biomechanical properties—and their inter-individual variability—play a key role in shaping kinematic outputs. Additionally, they report distinct developmental trajectories for the symmetry of muscular and kinematic synergies across the lifespan, with kinematic symmetry increasing earlier than muscular symmetry. The authors suggest that these differences in timing may be explained by developmental changes in the symmetry of biomechanical parameters.

While I highly appreciate the longitudinal dataset and the integrative nature of the work—bridging muscular synergies, kinematic synergies, and biomechanical modeling—I have several methodological concerns detailed below.

We thank the reviewer for the supportive and encouraging comments. In this revision we have tried our very best to further improve all aspects of the manuscript. Hopefully insights from our data will contribute to our understanding of the co-development of locomotor muscle synergies, kinematic synergies and the biomechanical properties.

Regression linking KS variability to MS and biomechanics (Line 293-314):

While the link between KS variability, MS variability, and biomechanical parameters appears to be one of the central conclusions of the study, some aspects of the regression analysis are not entirely clear.

First, the regression is based on variability indices computed within KMS clusters (Fig. 4A), but it is not clear whether all subjects could be included in this analysis, and if so, whether each synergy from a given subject was necessarily assigned to a different cluster, or whether multiple synergies from the same subject could end up in the same cluster (in which case, how did the authors proceed to reattribute KMSs to the good clusters). Clarifying this seems important to assess how representative the clusters are of the full cohort, and would itself provide valuable insight into inter-individual variability within the cohort.

Thank you for highlighting this point of confusion. In our *k*-means clustering analysis across the entire subject cohort, we intentionally allowed multiple synergies from the same subject to be assigned to the same cluster. Our primary goal in this step was to group similar MSs together and assess the variability of their associated KSs; thus, ensuring sufficient similarity among MSs within each cluster was our main priority. For this reason, we did not impose a constraint that would require MSs from the same subject to be assigned to different clusters, and we only tested cluster numbers ranging from 2 to 20 and then selected the optimal number based on the number that yielded the highest silhouette value. In addition, if we had required that each subject's KMSs be placed in separate clusters, this would likely have resulted in a much larger number of clusters, thereby complicating the identification of representative groupings of similar MSs across the cohort. We have provided further clarification on this clustering process with additional details in lines 848-851. Relevant data were organized in an Excel file (Data3-BetweenSubject Comparison.xlsx).

Second, the main conclusion does not appear to be fully coherent with the regression models displayed above the plots in Fig. 7 (specifically, in Fig. 7A, the effect of MS variability appears to be not significant ($p = 0.278$), and in Figure 7B, only marginally significant ($p = 0.048$), especially in a context of multiple comparisons). As such, it is unclear which p-value the authors refer to in the summary sentence (" $p < 0.05$ ") — is it for the full model, or for the individual predictors? Can the authors clarify which statistical results exactly support the claim?

We have carefully reanalyzed the data and revised Fig. 7 to more clearly present the p values associated with each fitted model. Our updated results demonstrate that at CH4, inter-subject variability in OptimalFibreLength, TendonSlackLength, and StrengthCoefficient interacted with variability in MS, thus collectively influencing the inter-subject variability observed in KS (Fig. 7A-C). But during CH1-CH4, the inter-subject KS variability was mainly affected by variability in StrengthCoefficient (Fig. 7D). We have provided the p values of the F-statistics to assess whether the fitted model differed significantly from a constant (null) model, as well as the p values of the t-statistics to determine whether each factor coefficient was significantly different from zero. Related content has been updated accordingly in the revised manuscript at the indicated lines 327-334.

Additionally, only two biomechanical parameters (TendonSlackLength and StrengthCoefficient) are shown. It would be helpful to either provide results for all tested parameters, or to clarify the rationale behind the selection of these two. It would also be useful to comment on the choice to test each biomechanical parameter in a separate regression model, and whether a joint model was considered.

Each of the ten biomechanical parameters was tested in separate regression models, and the data and results were organized in an Excel file (Data4-Muscle-tendon Parameters.xlsx). In Fig. 7, we have presented regression models only for those parameters whose p values were less than 0.05. For the joint model, we have

included all ten biomechanical parameters in the regression and did not find any significant effect for each parameter involved.

Finally, the 3D scatter plots in Figure 7 are difficult to interpret visually. It is unclear whether the individual data points lie above, below, or on the regression plane, making it hard to assess the quality of the fit. If the graph can't be improved in 3D, I suggest adding 2D projections (MS and biomechanical parameters).

Thank you for the suggestion. In the updated Fig. 7 (A-C), we have reduced the transparency of the fitted surface. This should allow the reader to see whether any point lies above or below the surface based on its color. Points below the surface are now slightly darker or shaded. Furthermore, when the influence of muscle-tendon parameters was not considered, the 2D projections comparing within-subject KS variability to that of MS variability across six stages have been presented in Fig. 5C.

Complexity and stability of activation coefficients over the lifespan (Line 315-331 – 740-759): How complexity was computed lacks methodological details (maybe a methodological section could be added with this keyword, as it is the case for Gait Stability and Pattern Stability?). Specifically, I am not sure I understand which input was given to the clustering. Isn't the number of AC per subject the same as the number of KMS? If so, does this mean the AC clustering is based on pooled data with uneven numbers of ACs across individuals and stages? Is this complexity index different from the number of KMS?

For each subject and stage, we used the number of AC clusters to represent the complexity of the neural commands that activate the MSs, which is distinct from the number of KMSs. A single KMS can be associated with multiple ACs across different gait cycles, and these ACs may display variable activation patterns. To quantify such variability of ACs at each stage for each subject, we pooled the ACs of all KMSs together and applied *k*-means clustering to identify the appropriate number of AC clusters. Additional methodological details on calculating AC complexity have been added into the section, 'Convergence of AC Pattern', in lines 941–945.

Moreover, if I'm not mistaken, how many gait cycles were available for each infant and each stage is not reported. Was this number fixed, including across stages? If not, how did the authors ensure comparability for complexity and stability analyses?

The number of gait cycles varied across stages and subjects, as detailed in Table 2. The average number of gait cycles per stage and per limb across the 11 infants ranged from 15 to 26. For the adult and elder groups, the average in adults was 12 for the left lower limbs, 14 for the right lower limbs, while for the elders, 13 for the left, and 14 for the right. These relatively close numbers of gait cycles across subject groups allow for reasonable across-group comparability of the metrics for AC complexity and stability. Additionally, these metrics were calculated from the AC of each gait cycle and then averaged across all gait cycles of each subject at

each stage, which helps represent any statistical tendency of these metrics across developmental stages.

Limitations

Although the study is methodologically rich, potential limitations are not currently discussed. It would be valuable to include a reflection on how certain methodological choices may have influenced the measurement of variability and symmetry, particularly: (i) the fact that children were held by their parents rather than by a standardized experimenter during stepping tests, (ii) the methodological choice of analyzing functional developmental stages despite the wide age ranges within each group (including how these functional stages were defined), and (iii) the use of the O'Connor algorithm for gait event detection, which was developed for adult gait and may not be appropriate for the less stable and more variable gait patterns observed in infants.

Thank you for the suggestion of including a reflection on the potential limitations of our methodologies, which we have added in the revised manuscript (lines 616-652).

- (i) Parental support during stepping tests: The means of providing external support from the parents in stages CH1 and CH2 was variable across subjects and not subject to specific constraints or standardized measures, thus potentially introducing variability in the extracted KMSs and ACs both within and across experimental sessions. Since we did not measure the amount of support provided, we could not assess how this variability may impact on the patterns and changes of MS and KS. Future studies would benefit from a more controlled environment with a consistent support system controlled by trained experimenters.
- (ii) Functional developmental stages: While we tracked the development of infants based on their functional capability of walking, we acknowledge that the wide age range within each stage may enlarge the between-subject variability of the observed MS and KS. Indeed, this issue of wide age range arose partly from the logistical difficulty of scheduling experiments that could accommodate the limited availability of both the parents' time slots and laboratory resources, a challenge that we will try to overcome in our future work. Moreover, the four functional stages used here (CH1-CH4), so far defined based on the observed behaviors and developmental milestones of the infants, may be more precisely redefined, perhaps based on quantifiable neurophysiological or biomechanical measures.
- (iii) O'Connor's algorithm for gait event detection: We initially used the O'Connor algorithm integrated within Visual3D to automatically extract gait cycles, then manually selected the relatively complete cycles. This was done by ensuring that the height of the heel marker transitioned from one minimum to the next same minimum for each selected cycle. With our manual examination, we are confident that gait event detection should be accurate and robust for all age groups. We have further clarified this point in lines 714-715.

Minor comments

Line 42-45: This sentence suggests a strong established relationship between muscle synergy asymmetry, biomechanical asymmetry, and resulting kinematic symmetry. While the study presents interesting evidence that all three factors evolve during development, it remains unclear whether the evolution of MS symmetry explicitly compensates for limb biomechanics to drive KS symmetry. I would recommend adjusting the abstract to align more closely with the cautious and well-balanced interpretation given in the discussion section (line 449-466).

Thank you for this suggestion. We have updated the sentences in lines 44-46 accordingly.

Line 184-197: The comparison of Pearson correlation values across KS, MS, and KMS might provide useful insights into the relative developmental stability of these components, but I wonder if they are comparable given the differences in dimensionality between the vectors (KS: 9, MS: 14, KMS: 23). Could the authors comment on whether vector dimensionality might influence Pearson correlation values and affect comparability?

Varying vector dimensionalities can indeed influence the comparison of Pearson correlation coefficients (r) as higher dimensions may introduce greater data variance and potentially lower r values. To address the impact of these differences, in the revised manuscript in lines 1147-1149, we applied Fisher's z transformation to the r values prior to statistical analysis. Fisher's z transformation helps normalize the distribution of correlation coefficients and stabilize their variances, thereby improving their suitability for statistical comparison across conditions. Nevertheless, we recognize that this approach cannot fully eliminate the effects of differing dimensionalities, but it remains the most effective method currently available to minimize the associated bias.

Line 335: I'm not sure I understand the term "correlation between gait-cycle pairs". In the Methods, it is stated that correlations were calculated between each gait-cycle AC and the average AC across cycles. Are the authors referring to the same computation?

Yes, "gait-cycle pairs" means the pair that includes the AC of each cycle and the mean AC across cycles. We have further clarified the description in lines 359-360.

Line 394-395: Could the authors clarify which analysis (or/and how Fig.4B) demonstrates consistency in dominant muscles across individuals?

In Fig. 4A, we clustered all the KMSs of each subject cohort based on the similarity of MS (i.e., only the MS portion of each KMS vector was used in the clustering) and derived three clusters in each developmental stage. For each cluster, we calculated the within-cluster standard deviation (SD) of the synergy weight of each muscle, and then averaged these SD values across the dominant muscles to quantify the between-subject variation of the MSs. Fig. 4B shows that the average within-cluster SD values of the MSs were less than 0.15 across all six stages, indicating that the dominant MS patterns in the population were quite stable. We have

clarified this procedure in lines 421-422.

Line 591: The reported age ranges for stages appear inconsistent with the values in Table 1 (e.g., CH1 includes two subjects younger than 6.3 months).

We have rechecked the data and updated the subject information in Table 1.

Subject 1, at CH1: 6 months 16 days (0.54 year).

Subject 8, at CH1: 0.85 year, 8.3kg, 69cm; CH2: 1.05 year, 8.7 kg, 71 cm.

Figure 3B: the colors of CH2 and CH3 cannot be well distinguished

We have updated the colors in Fig. 3B accordingly.

Figure 4: Clusters are very small and difficult to read. Considering the richness of the dataset—including synchronized EMG and kinematic recordings across multiple developmental stages—it would be valuable to present clearer visualizations of the KMSs across development (with labeled angles and muscles). In the same vein, adding an illustration showing how EMG and joint angle signals evolve across development—prior to dimensionality reduction—would provide helpful context for interpreting the extracted KMSs and for visualizing variability/complexity in the original, observable motor patterns.

Thank you for the suggestion. We have enlarged the KMS clusters in different figures and added an illustration of the pre-processed EMG and joint angle signals across the four infantile stages in Fig. 2A.

Responses to Reviewer No. 2

In this paper, the author examines changes in synergy during the developmental period of walking, and argues that synergy changes in order to improve bilateral symmetry in movement. The attempt to clarify the factors behind changes in muscle synergy during development from the data is interesting, and the hypothesis that muscle synergy changes in order to resolve physical changes and movement asymmetry during development is convincing. The idea of using a musculoskeletal model that deals with physical development to investigate the relationship between movement and muscles is also convincing. On the other hand, when it comes to whether the hypothesis was sufficiently verified by the results, I felt that there were some unclear points in the current paper. The following are specific points of questions and problems.

1: Regarding the data on KMS synergy

One of the features of this paper is that it uses a synergy that combines the characteristics of kinematics and muscle synergy, called KMS synergy. I felt that there were two problems or questions regarding the results of this KMS.

* [1.1: Only a part of the KMS synergy is presented] In the paper, only the representative example in Figure 2A is presented as the result of the KMS. To understand how the KMS synergy is structured, I would like to see more KMS synergies, not just the representative example. Can you provide the results for each subject (even

only for the infants) as supplementary data? Also, if possible, I think it would be helpful to understand the results in Figure 4A if the authors could attach labels for the data corresponding to Cluster 1, Cluster 2, and Cluster 3 etc. in the cluster analysis.

Thank you for this suggestion. Given the large number of KMSs identified from the infants across four stages and both limbs, clear visualization of all of them would be challenging. Hence, we have organized the KMS clusters in the supplementary data (Data3-BetweenSubject Comparison.xlsx), labeling each KMS from each stage with its corresponding subject number. This file should provide a more comprehensive presentation of the KMS synergies.

* [1.2: Joint angle acceleration data for analyzing KMS] The paper uses JAC for all three axes to calculate KMS. Here, I have some questions about this approach. The movements being targeted are mainly composed of joint flexion/extension (x-axis), and most of the muscle activity being measured is also from muscles that compose this movement. Nevertheless, the data from the y-axis and z-axis are normalized and treated in the same way as the x-axis, in the paper. Is this processing appropriate? As a result of this process, for example, in KMS2 for CH4 in Figure 2A, the z-axis of the ankle is the main component of the activity. Here, I have some doubts about whether this result is appropriate for representing the activity at this stage. If possible, I felt it would be better to verify whether there is any difference in the results when KMS is calculated using only the x-axis data.

Thank you for your valuable comment. We believe that normalization of all axes is essential for a comprehensive analysis of infants' walking patterns and for accurately evaluating the functional outcomes of their corresponding muscle activations. The rationale behind this approach is the same as that for applying normalization to each muscle prior to across-muscle analyses. Without normalization, the algorithm may overemphasize axes with larger amplitudes, inadvertently treating those with smaller amplitudes as noise and thus biasing the results. Moreover, unlike adult gait—which typically exhibits consistent, predominantly x-axis variability—infant walking is characterized by substantial variability across multiple planes. As demonstrated in Figure 2B, for example, the z-axis of the ankle emerged as the principal component in KMS2 during CH4, underscoring the multidimensional and variable nature of infant movement patterns. Normalization ensures that all axes contribute equally, thereby enabling a more accurate and unbiased assessment of gait patterns in infants.

2: Changes in asymmetry with age: Regarding Figure 3B

* 2. 1: Since KS can take both positive and negative values, and MS is defined as only positive, KS correlations can be both positive and negative, while MS correlations can only be positive. In fact, in Figure 3B, only KS shows negative values. At this point, it is somewhat questionable whether it is fair to compare the values of the correlations between KS and MS. If you can justify this point, please write about it.

It is reasonable to question whether the broader range of KS as compared with that of the MS might introduce more variance into components of the KSs, potentially

leading to biases when comparing r values derived for KS vs. MS comparisons. To mitigate the effects of unequal ranges in KS and MS, in this revision we applied Fisher's z transformation to the r values prior to statistical analysis. Fisher's z transformation helps normalize the distribution of correlation coefficients and stabilize their variances, thereby improving their suitability for statistical comparison across conditions. The revised methodology is now described in lines 1147-1149. Notably, the MS correlations also display negative values in Fig. 3B.

* 2.2: The authors have performed a regression analysis of KS and MS using cubic and quadratic functions, respectively, and compared the results at the peak ages, but is it fair to compare the peaks of regressions in different dimensions? It would be better to verify whether the same results are obtained when the order of the regression is the same.

We appreciate your suggestion of applying the same degree of polynomial functions to both KS and MS symmetry analyses. We applied quadratic modeling to both datasets and updated the results in Fig. 3C. However, the quadratic model did not provide a statistically significant fit to the MS symmetry development curve ($p = 0.23$). Notably, these results indicate that KS symmetry peaked prior to adulthood at about 11 years of age, whereas MS symmetry demonstrated a more subtle increase with age without a specific peak. However, we acknowledge that at this stage, our primary objective is to characterize the overall trend of evolution of KS and MS symmetry rather than to rigorously compare their peak times.

3: Changes in between-subject variability of KS and MS (Fig. 4A, Fig. S5)

* 3.1: I agree with the method used in the paper to evaluate variability by collecting similar patterns as clusters. I would like to know the overall picture of the clusters, such as how many clusters there are (as the number of clusters vary for each subject) and how many KMS synergies belongs to each cluster. As I commented in 1.1, it might be possible to show the overall picture by including all KMS data in the Supplementary and labeling them.

The k -means clustering in the said figure was performed on the KMSs collected from each subject cohort rather than from each subject. Three clusters were required in each stage. The figure's associated KMSs and their corresponding subject labels were organized in an Excel file uploaded as Supplementary Materials (Data3-BetweenSubject Comparison.xlsx).

* 3.2: I would like more explanation about Figure S5. The MS results for the data in Figure S5 show the same pattern for all clusters, but did other patterns not belong to any cluster?

In Fig. S5, k -means clustering was applied to categorize KMSs based only on the similarity of KS. Because of the variable association between KS and MS, this clustering of the KMSs by grouping similar KSs together leads to significant MS variability within each KMS cluster. We believe this within-cluster MS variability results in the cancellation between the larger and smaller MS components in many muscles when the MS cluster average was calculated, thus yielding similar average

MSs across the clusters. All KMSs were included in this clustering process.

4: Interpretation of the musculoskeletal model

In order to investigate the relationship between muscle synergy and kinematics, the paper estimates muscle activity using inverse dynamics and static optimization with a musculoskeletal model based on motion data, and then uses the measured muscle activity to make corrections and examines the body parameters. In interpreting the results, I would like to know the following three points regarding the quality of the results.

* 4.1: To what extent do the estimated (post-correction) muscle activities match the measured muscle activities? Since the estimation is carried out using optimization with a musculoskeletal model, it is desired to indicate the degree of error in the results when considering the quality of the results.

Thank you for raising this important point. In our study, muscle-tendon parameter calibration was performed by fine-tuning the default parameters to minimize the error between the estimated and experimental joint moments. Experimental joint moments were calculated using the inverse kinematics and inverse dynamics modules in OpenSim. For parameter calibration, we utilized a simulated annealing algorithm integrated within CEINMS. The algorithm was regarded as converged when the error between estimated and measured joint moments was less than 0.001 (Newton-meter)². Parameter variations were also constrained within predefined physiological boundaries to ensure realizations of biomechanically realistic muscle functions^{1,2}. In the revised manuscript, we continued to perform the execution step of CEINMS, which evaluates the calibrated model's capability to predict muscle activations in independent trials not used for calibration. Model accuracy was assessed by calculating the Pearson correlation between the predicted muscle activations and the measured EMG signals for each muscle channel, validation trial, and subject. To account for potential temporal misalignment between the predicted and recorded signals, we utilized a cross-correlation approach to identify and correct for the optimal time lag. Additional methodological details can be found in lines 1052-1075. The corresponding results were presented in lines 302-310 and illustrated in Fig. S6. Following temporal alignment, the Pearson correlation between predicted and measured EMGs was 0.60 ± 0.31 for right-limb muscles and 0.58 ± 0.31 for left-limb muscles. In comparison, without applying temporal alignment, the correlations were 0.37 ± 0.29 (right) and 0.40 ± 0.28 (left), respectively. This high correlation therefore argues for the reliability of our parameter calibration and directly addresses your concerns regarding the accuracy of our personalized NMS models for the infant cohort.

* 4.2: Are the floor reaction forces and the support forces (of CH1 and CH2) taken into account? When performing inverse dynamics, it is possible to calculate torque without including reaction forces, but if force data is not included, the torque estimated with inverse dynamics and the muscle activity estimated from that will be off by that amount. How much consideration is given to the relationship between these forces? If they are

not taken into account, it seems necessary to discuss how they might affect the body parameters.

Thank you for your insightful question regarding the potential inclusion of ground reaction forces and support forces (from CH1 and CH2) in our analysis. Due to limitations in our experimental setup, these force data were not collected during our data recording sessions. We fully agree that this omission may affect the accurate calculation of joint moments using inverse dynamics and also impact the calibration of muscle-tendon parameters in our infant cohort. However, as shown in our model evaluation results (Fig. S6), the model's performance in predicting muscle activations remained robust, and most importantly, was also consistent across all four infantile developmental stages. This suggests that, while the accuracy of the absolute values of the parameters may be affected, their relative trends across the infant developmental stages are at least still reliably captured.

* 4.3: The data for CH2 and CH3 in Number 20 and 24 in Table 1 are the same. Please check whether this is a simple mistake in the description.

Specifically, the data for CH2 and CH3 from infants 20 (No. 4) and 24 (No. 5) were collected at the same age. We have presented the result in lines 675-679 and discussed this issue in lines 621-628.

In addition, there were the following points that were unclear about how to interpret the results.

* 4.4: Both height and weight have increased by about 1.5 times between CH1 and CH4, and accordingly, muscle length and the required torque have increased. Therefore, the results showing that muscle length and the strength coefficient increase are understandable. On the other hand, is it possible that the optimal fiber length decreases even though the height increases? Are these values normalized in some way? If they are not normalized, then the vertical axis of Figure 6 should include the unit as much as possible.

Thank you for this valuable feedback. It is correct that both the height and weight increased significantly between CH1 and CH4, and such increases are consistent with the observed increases in muscle length and strength modulation. However, we think that OptimalFibreLength can still decrease even as height increases since it is influenced by various architectural and functional adaptations, including alterations in pennation angle and tendon length. Besides, we did not normalize the muscle-tendon parameter values in this analysis. ShapeFactor and StrengthCoefficient do not have specific physical units. The ranges for these parameters were initially defined as absolute values prior to calibration, following the default settings of the CEINMS toolbox. On the other hand, TendonSlackLength and OptimalFibreLength are measured in meters (m), and we have now included these units in Fig. 6. The ranges for these parameters were set by default as multiplying factors of their initial values, which were derived from each subject's scaled OpenSim musculoskeletal model. In the revised manuscript, we have clarified this point in lines 1094-1101.

* 4.5: The change in shape factor is written as one of the characteristics of the results, but it was difficult to get a clear picture from the paper of how shape factor is specifically defined and how it changes with age. I felt that this part needed to be explained. Also, in line with this, I felt that a more detailed explanation was needed for the explanation in P11, Line 416-417: "thus potentially leading to more stability in the force generated across step cycles despite the step-to-step variability in neural activations."

Thank you for pointing out the need for a clearer explanation of the shape factor and its age-related changes. As defined in the CEINMS toolbox, the ShapeFactor is a constant that determines the nonlinearity in the relationship between neural activation and muscle activation. Specifically, it modulates how changes in neural input are translated into muscle activation. The pattern of ShapeFactor change with age varied across different muscles, as illustrated in Fig. 6A. A decrease in the ShapeFactor means that fluctuations in neural activation have a reduced effect on muscle activation. This implies that the muscle activation level becomes relatively more stable and less sensitive to small, step-to-step variations in neural control. Consequently, this greater stability likely contributes to more consistent force generation during walking, even in the presence of neural variability. To address your comments, we have added further explanation of the ShapeFactor and its functional implications to lines 441–450 in the revised manuscript.

* 4.6: The results of the musculoskeletal model, which the paper claims to show symmetry between the left and right sides, are shown in Supplementary Figure S6, and these results are often referred to in the Discussion as supporting the change in symmetry. Therefore, I felt that this figure should be included in the main text rather than in the Supplementary (not mandatory). Also, if anatomical evidence could be referenced regarding the changes in tendon slack length during development, I felt that the paper's claims would be strengthened (also not mandatory).

Thank you for these constructive suggestions. We have relocated Fig. S6 to Fig. 6E as appropriate, to support our argument that the increased symmetry of biomechanical properties contributes to the improved symmetry of MS and KS, and that there is developmental fine-tuning of limb biomechanics possibly driven by the feedback from asymmetrical KS in the early stages. Lastly, there is currently a lack of references regarding the possible anatomical changes in TendonSlackLength during such an early stage of motor development (and especially regarding its left-right symmetry). This underscores the novelty of our findings, indicating that further studies are necessary for a more complete characterization of the development of muscle-tendon properties.

Responses to Reviewer No. 3

This manuscript offers a novel and valuable contribution to understanding developmental changes in motor control through the analysis of muscle and kinetic

synergies. The manuscript investigates how muscle synergies evolve throughout human development, particularly comparing kinetic synergies (KS) and muscle synergies (MS) across different life stages, from infancy to adulthood. The authors claim that KS show higher variability and a more rapid rate of change in early stages compared to MS, which remain relatively stable and become sparser and more independently controlled in adulthood. This manuscript presents an interesting and valuable contribution to understanding motor control development. The study offers novel insights into the differential dynamics of kinetic and muscle synergies, which could influence future research in developmental neuroscience, motor control, and rehabilitation. In addition, the concept of comparing kinetic and muscle synergies across development is interesting and novel. The conclusions are generally supported by presented evidence. However, ensuring robust statistical correction and clearly articulating methodological choices will significantly enhance the manuscript's strength and reproducibility.

General comments:

1. Lines 590-594: The authors explicitly define four distinct developmental walking stages (CH1: weight-support; CH2: hand-support; CH3: early independent walking; CH4: later independent walking), each characterized by specific chronological and developmental criteria. However, Table 1 highlights critical inconsistencies regarding participant age across these stages: participant 25 has identical ages (0.93 years) reported for both CH1 (weight-support) and CH2 (hand-support). And participants 20, 24, and 27 also show identical ages for CH2 (hand-support walking) and CH3 (early independent walking). This seems to indicate that, at least in these cases, different walking conditions were recorded at the same developmental time rather than representing separate, chronologically distinct stages. Could you please clarify how children were distinguished and assigned to the different walking stages? If this is indeed the case—that is, if some children were assessed in multiple walking conditions simultaneously rather than distinct developmental stages—this should be clearly stated. Importantly, this distinction significantly impacts the manuscript's primary claim of investigating developmental trajectories across four clearly separated walking stages. Therefore, it is crucial that the authors explicitly address and clarify this inconsistency in detail. Specifically, they should explain clearly whether each participant indeed passed through four separate developmental stages, as described in the methods, or if instead they were assessed across multiple conditions within the same developmental stage. This clarification is essential for interpreting the study's findings and validating the conclusions drawn regarding developmental progression in walking stages.

Thank you for your question. In the instances you mentioned, the infants were indeed evaluated, at the same age, with two functional walking tests whose performances would ideally be indicative of two different developmental stages, respectively. During those data recording sessions, the infants could perform the walking trials while being subjected to the task requirements for both developmental stages. We suspect that this situation arose either because we missed the ideal time window for a specific walking stage, or because these infants developed with a trajectory characterized by considerable overlap between stages. At the very least, according to their parents, these subjects showed typical gait

development. We believe the inclusion of these subjects would not alter the overall conclusions of our study, as this phenomenon was observed only in five out of a total of 44 cases (across 4 stages for 11 infants). We have clarified this point in both the Methods and Discussion (under “Limitations”) sections, as indicated in lines 675-679, lines 891-892, and lines 621-628.

2. Figure 2C and Table 1: Could the authors clarify precisely what is shown in Figure 2C? According to the legend, the X-axis represents the number of days elapsed between two developmental stages for each subject (CH1-CH2;CH2-CH3;CH3-CH4). However, Table 1 reports identical ages for several participants across different stages (e.g., participant 25 has the same age in CH1 and CH2, and participants 20, 24, and 27 have the same age in CH2 and CH3). If the difference in days between recordings was truly zero in these cases, how is it that the smallest value plotted in Figure 2C is 32 days? Were these cases excluded from the regression? Or is the variable on the X-axis representing something different than the actual time between recordings? Please clarify this apparent inconsistency between the figure and the subject data.

Thank you for pointing out this confusion. The X-axis variable represents the actual time elapsed between two consecutive recordings. Indeed, the cases with between-stage duration of zero were excluded from the regression analysis. We have clarified this point in both the Methods and Discussion (under “Limitations”) sections, as indicated in lines 675-679, lines 891-892, and lines 621-628.

3. Given the high variability typically observed in walking patterns among infants and toddlers, the number of young participants included appears relatively low. Additionally, the number of participants included in both the adult and elderly groups (n=7 each) is also limited. Increasing participant numbers or providing clear justification for the sample sizes chosen would strengthen the reliability and generalizability of the findings.

We acknowledge that the sample sizes for the different subject groups are relatively small, especially considering the variability observed in the walking patterns of infants and toddlers. However, for the infant group, our primary focus was on characterizing within-subject longitudinal changes over time rather than inter-individual variability, which lessens somewhat the absolute necessity of a larger sample size. Regarding the adult and elderly groups, we believe that a smaller sample may not impact on the general conclusions, as inter-individual variability in these populations is typically smaller than that in infants. Our relatively smaller sample sizes notwithstanding, several of our findings are consistent with results of previous studies, thus lending general credibility to our results. In particular, those concerning changes in MS and AC throughout infant development indicated in lines 574-597 are consistent with Dominici *et al.*, 2011³; Sylos-Labini *et al.*, 2020⁴; Zandvoort *et al.*, 2022⁵. Nonetheless, we have acknowledged this limitation of sample sizes in lines 628–630 and recommended that future research should include a larger number of participants.

4. Please clearly state the exact criteria used for determining the number of clusters in the k-means clustering analyses (e.g., silhouette scores are mentioned, but explicit thresholds or procedures for selecting cluster number should be elaborated).

For each clustering analysis, we first analyzed with the number of clusters varying successively from 2 to 20, and then identified the optimal number based on the one yielding the highest average silhouette value. This methodological detail has been further clarified in lines 826-851.

5. Statistical analysis: The statistical approaches employed (e.g., Pearson similarity, regression analyses, and sparseness calculations) are appropriate. However: Clarify whether corrections for multiple comparisons were performed. If not, this should be addressed explicitly, considering the multiple statistical tests conducted across life stages.

6. Statistical analysis: Provide more detailed descriptions or justifications for chosen statistical tests in the main manuscript text.

22. Lines 184–188 / Figure 2B and Fig. S1A: The manuscript reports statistical comparisons of between-stage synergy similarity values (e.g., KS vs. MS) across developmental transitions, with associated p-values (e.g., $p = 0.039$, $p = 0.023$). However, the specific statistical test used to assess these differences is not stated. Since the comparisons appear to be based on within-subject, paired data (i.e., comparing KS and MS similarity for the same subjects), it is important to clarify whether a paired t-test, Wilcoxon signed-rank test, or another method was used. Additionally, since the similarity values are Pearson correlation coefficients, it is worth noting that these are not normally distributed. A Wilcoxon signed-rank test would be more appropriate unless the authors applied a Fisher z-transformation before performing a parametric test. Please clarify the statistical approach used and consider whether a non-parametric alternative might be more suitable. More broadly, the analysis appears to involve repeated measurements across both synergy type (KS, MS, KMS) and developmental stage transitions (CH1→CH2, CH2→CH3, CH3→CH4) for the same subjects. This suggests that a two-way repeated measures ANOVA or a linear mixed-effects model may be more appropriate to account for within-subject structure and potential interactions between factors. Could the authors clarify the overall statistical strategy used for these comparisons? Were pairwise tests applied independently, or was a multi-factorial model considered? Also, were any corrections for multiple comparisons performed? Explicitly outlining the statistical approach would improve the transparency and robustness of the analysis.

Thank you for your three thoughtful comments regarding the statistical analyses. We would like to address them collectively below. As we compared the metrics across 4 time points for the infant cohort, specific statistics for these repeated measurement data were demanded. We first examined the assumption of data normality using the Kolmogorov-Smirnov Test, and then examined data sphericity using the Mauchly's test. If these assumptions were met, we employed the repeated measures Analysis of Variance (ANOVA), or the non-parametric Friedman test if

not met, to test for statistical significances of the differences across time points. If a significant difference was found, we further performed a post hoc multiple comparison test with Bonferroni correction for identifying specific group pairs that exhibited significant differences. When we separately compared the metrics of each infantile stage with the Adult and Elder stages, we employed one-way ANOVA and Bonferroni correction post hoc for normally distributed data, or the Kruskal-Wallis test with the pairwise Wilcoxon rank-sum test post hoc for non-normal data.

To examine the correlation between any two vectors, we computed the Pearson correlation coefficients (r), which indicates the strength and direction of the correlation. To statistically compare the r values derived from comparisons of KSs, MSs, and KMSs, the Fisher's z transformation⁶ was first applied to normalize the distributions of r values. When employing a linear regression analysis to investigate the relationship between factors, we reported the p values of the F-statistics to assess whether the fitted model differed significantly from a constant (null) model, as well as the p values of the t-statistics to determine whether each factor coefficient was significantly different from zero. The significance level for hypothesis testing was set at 5%. We have provided a justification for the selection of statistical tests in the revised manuscript, as detailed in lines 1131–1154.

7. Interpretation of Results: The interpretation regarding the stability of MS vs. variability of KS is compelling but could benefit from additional evidence or discussion: are there potential confounding factors (e.g., activity level differences, walking speeds, or environmental influences) that might impact the variability observed in KS?

8. Could the authors comment further on why kinetic synergies might exhibit higher variability relative to muscle synergies?

23. Line 363 Discussion: The discussion could be enriched by further exploring possible interpretations of the observed differences between kinetic (KS) and muscle synergies (MS) across development. While the authors describe KS as more variable and changing more rapidly than MS, it may be helpful to consider how these patterns could relate to underlying neural or biomechanical mechanisms. For instance, could KS be more influenced by body dynamics and postural demands, whereas MS might reflect more stable, spinally organized motor modules? Expanding on these possibilities—even as hypotheses—could help contextualize the findings within broader models of motor development.

Given that these three questions all call for a deeper exploration of the greater variability of KS as compared with that of MS, as observed in our analyses of between-stage, between-limb, and between-subject MS and KS variations, we shall address them collectively below.

In human locomotion, MSs are understood as neuromotor control modules, implemented by spinal neuronal networks, which coordinate the activations of multiple muscles as discrete control units to enable stable walking. On the other hand, KSs reflect the resulting joint movement patterns produced by this

coordination. For any single motor action, the translation from muscle activation to observable joint movement is influenced by multiple intermediary factors and sources of variability, as noted by the reviewer, which may include differences in activity level, walking speed, environmental conditions, and postural demands. We have briefly addressed these contextual factors in the revised discussion (see lines 486-491). However, we did not explore these aspects in depth, as our study primarily focuses on the overall developmental progress of the MS and KS rather than the hierarchy between MS and KS needed for locomotor control. Additionally, infants in our study performed natural walking, with or without external support, in a relatively controlled laboratory setting. Therefore, some of the factors mentioned above may not be directly examined in our design. We sincerely appreciate these insightful suggestions and intend to investigate these relationships in greater detail in future research.

9. Figures and Supplementary Material: The supplementary figures provided robust additional support for main findings. However, it would be beneficial if some key supplementary results (e.g., Fig. S1, S3, and S5 regarding synergy similarity and rate of change) could be integrated or briefly discussed in the main text to strengthen the core narrative of developmental trajectories.

Thank you for your suggestion. The supplementary figures mentioned have been briefly discussed in the main text. Specifically, lines 405-409 and 483-485 for Fig. S1, and lines 485-486 for Fig. S3. Since Fig. S5 serves as a supplementary material for the findings in Fig. 4, we did not discuss it further in the main text.

10. The authors might consider explicitly stating potential future research directions or implications of their findings for clinical applications or theoretical models of motor development.

Thank you for your comment. We have incorporated these contexts into the revised Discussion ("Limitations") (lines 616-652).

Minor / Specific Points:

11. Lines 590-591: The authors mention that during the CH1 (weight-support walking) condition, "children walked with weight support from their parents lifting their body from the shoulder." Could the authors specify more clearly the exact method used for providing support? Specifically, were parents instructed to hold the infants around the waist, under the shoulders, or elsewhere, and what specific instructions were given to parents regarding the amount or proportion of body weight to support? Several studies have demonstrated significant effects of the amount of body-weight support on kinematic patterns and muscle synergies, both in toddlers and adults (Pang and Yang, 2000; Dominici et al., 2007; Kerkman et al., 2022; see review by Apte et al., 2018). It is crucial to clearly describe and address this methodological detail for correct interpretation of the data and comparison with previous literature.

Thank you for the comment. In CH1, parents were instructed to support the infants with gentle lifting from their armpits, while in CH2, they held the infants' hands. In both instances, parents adjusted their support to help the infants walk as steadily as possible. We have discussed this between-stage difference in how support was provided as a limitation (lines 617-621), highlighting the importance of a standardized experimental paradigm for providing external support.

12. Data Recording: The manuscript does not specify important details regarding the data collection protocol. In particular, it is unclear how many trials were recorded per participant, what was the size of the recording volume, and how many strides or gait cycles were collected in total. Providing this information is essential to assess the robustness and representativeness of the dataset. Please include these details to improve transparency and reproducibility.

The number of gait cycles for each subject at each stage is reported in Table 2. The cycle count varied across stages and infants, and was influenced by the duration of the data collection session and data quality. We selected the relatively complete gait cycles for further analysis. The average numbers of gait cycles for the four infantile stages are as follows: CH1: 15±9 (Left leg), 16±8 (Right leg); CH2: 22±11 (L), 22±9 (R); CH3: 25±16 (L), 26±16 (R); CH4: 24±13 (L), 25±12 (R). For adults, the mean numbers of gait cycles are 14±5 (L), 12±4 (R); for elders, they are 14±10 (L), 13±7 (R).

13. Line 608: For transparency and reproducibility, please explicitly specify the anatomical locations where markers were placed on the participants

The exact anatomical locations where the markers were placed were detailed in the revised manuscript (lines 691-693).

14. Lines 615-616: Could the authors clarify the rationale for recording muscle activity from the external oblique and latissimus dorsi muscles, given that the primary focus is on lower-body walking patterns? Particularly in the CH1 (weight-support walking) condition, there might be significant muscle-activation artifacts introduced by the researcher's support. Please discuss how this potential issue was addressed or justify the selection of these muscles.

These two muscles are biomechanically important to bipedal walking, in that external oblique helps stabilize the pelvis and spine, and latissimus dorsi assists in arm movement and stabilization of the trunk. Even when recorded during gait with external support, the EMGs of these muscles could still reflect their potential functional roles described above, or illuminate whether their activity patterns may be part of the modular motor programme for walking during the various developmental stages. Indeed, these same muscles were recorded in previous lower limb studies from ourselves (Rinaldi *et al.*, 2020⁷; Cheung *et al.*, 2020⁸) and others (Chvatal and Ting, 2013⁹; Bach *et al.*, 2021¹⁰).

15. Line 616 and line 630: The reference to Fig. 2A in these lines is unclear, as Fig. 2A

does not explicitly illustrate the EMG electrode placement or muscle selection, nor does it clearly depict the joint or 3D angles mentioned. Please revise the figure reference or provide an appropriate figure that visually clarifies the EMG electrode placements and joint angles analyzed.

We have updated the figure reference as Fig. 1 to avoid confusion (line 700).

16. Line 641: The selected cutoff frequency (50 Hz) for the high-pass filter applied to EMG data appears unusually high compared to standard practices. Could the authors clarify or justify explicitly why such a high cutoff frequency was selected?

Thank you for the question. We decided to use a 50-Hz cutoff frequency for the high-pass filter applied to the EMG data because similar cutoff frequencies were used in prior muscle synergy studies (references listed in line 727). In Guo *et al.* (2022)¹¹, for instance, a cutoff frequency of 40 Hz was selected because it was the minimum cutoff frequency that ensured that the number of extracted muscle synergies remained consistent. Here, we used a slightly higher 50-Hz cutoff to mitigate influences from movement artifacts and other low-frequency baseline noise, which, per our experience, are more pronounced in recordings from infants and toddlers than those from adults. While we acknowledge that this approach may attenuate some low-frequency signal content, we consider it necessary to ensure more robust and reliable analytic results derived from cleaner data.

17. Line 644-645: The description of applying the Savitzky-Golay filter to remove occasional high-amplitude noisy spikes from the EMG signal is not entirely clear. Could the authors briefly clarify the steps implemented for this process? Specifically, how were high-amplitude spikes identified—was there a threshold or criterion used? Providing a concise but clear explanation of these preprocessing steps would greatly assist readers in understanding and potentially reproducing the methodology.

The Savitzky-Golay filter was applied to mitigate influences from the occasional high-amplitude “spiky” noise in the EMG signal while preserving the overall signal structure. Such spikes can arise from the interaction between the FIR filter and certain high-frequency artifacts present in the data. We identified these spikes as noise if their amplitudes exceeded a predefined threshold, typically set at 3–5 times the signal standard deviation above the mean of the signal. The filter then smoothed the signal by fitting a polynomial of a specified order to a moving window of data points, effectively reducing transient noise while maintaining the overall physiological features of the signal. The threshold, window size, and polynomial order were optimized based on the dataset’s characteristics to ensure that the filtering process effectively removed the noise without distorting the meaningful components of the EMG signal. Here, we set the polynomial order to be 2, the window size to be 3, and fixed the noise threshold using Otsu’s method. We have provided further clarification on the related details in lines 730-737.

18. Line 649: The authors apply a fixed electromechanical delay of 50 ms to shift EMG signals. However, this value is based on adult data and may not be appropriate for

infants and toddlers, who usually show longer electromechanical delay due to their immature neuromuscular systems. Please clarify why a constant 50 ms delay was used for all age groups, and consider discussing whether age-specific electromechanical delay values would be more accurate.

Thank you for the question. We chose to apply a fixed electromechanical delay (EMD) of 50 ms across all age groups to maintain consistency and simplicity in our analysis. This value was widely reported for healthy adults and was commonly used as a standard reference in the literature¹². However, we acknowledge that this approach may not fully capture the longer EMDs typically observed in infants and toddlers, whose developing neuromuscular systems can result in delays of 70–100 ms. To evaluate whether different EMD values would impact KMS extraction, we compared KMSs derived using 50 ms, 70 ms, and 90 ms EMDs. Given that the raw EMGs were integrated over 20-ms intervals (each representing one data point in the preprocessed EMGs), we adjusted the alignment between the preprocessed EMGs and joint acceleration data for the right limb of the first subject by shifting one data point for 70 ms and two data points for 90 ms. Similarity analyses between the original KMS (at 50 ms EMD) and those generated with alternative EMDs yielded high Pearson correlation coefficients of 0.93 (70 ms) and 0.90 (90 ms), indicating that the KMSs extracted should remain relatively stable even with a 40-ms deviation of EMD (see figure below). Nevertheless, we recognize that additional data are needed to more comprehensively assess the impact of varying EMDs. We also agree that incorporating age-specific EMD values would improve the precision and validity of our findings. Future analyses should consider age-dependent EMD adjustments to better reflect the physiological characteristics of each group, thereby enhancing the accuracy of neuromuscular function assessments during movement tasks. We appreciate the reviewer’s suggestion and have addressed this point in the revised limitations section (lines 632-640).

Figure. Comparison of KMSs extracted using different EMDs for the right limb of the first infant. The first row displays the six KMSs extracted with an EMD of 50 ms, the second row with 70 ms, and the third row with 90 ms. KMSs across different EMDs were aligned based on the highest Pearson correlation. Channel information for each KMS is indicated in the lower left corner of the figure.

19. Line 705: Please clarify how many gait cycles were ultimately included per participant in the concatenated matrix used to extract KMS.

The number of gait cycles that were concatenated to extract KMSs for each subject is reported in Table 2.

20. Figure 1 (section: longitudinal - learn to walk): The image used to represent the 'supported walking' condition could be improved by showing a toddler clearly supported by an adult (e.g., holding hands while walking). This would help distinguish more clearly between supported and independent walking, improving the visual clarity of the developmental stages depicted. Additionally, the figure and data collection section mention 'BodyWeight'—could the authors clarify what specific body weight information was collected and how? This detail does not appear to be reported elsewhere in the manuscript.

Thank you for the suggestion. 'BodyWeight' indicates the body weight of the infant in kilogram (kg), which was measured by the weighing scale after each recording session. We have added this information in the legend of Fig. 1 (lines 1161-1162).

21. Figure 2A is very small and difficult to evaluate clearly. Please consider enlarging the figure or improving its resolution to ensure that all elements—especially EMG profiles and joint angles—are clearly visible and interpretable by the reader. Additionally, while Figure 2A illustrates an example from a single infant, it appears that a larger number of strides were recorded for CH1 compared to CH4. It would be helpful if the authors could report the average number of strides (and/or the range) used for the analysis at each developmental stage, as well as for adults and elderly participants. This information is important to assess the representativeness and balance of the dataset across conditions.

Fig. 2 has been updated accordingly. The number of strides is reported in Table 2. On average, the number of gait cycles ranged from 15 to 26 for the 11 infants across four stages and two limbs. For the adult and elder groups, the average was 12 for the left lower limbs, and 14 for the right in adults; 13 for the left, and 14 for the right in elders. This relatively consistent number of gait cycles across the age groups and limbs allows for reasonable comparability of the metrics.

24. Lines:1013-1014 Figure 1 legend: should be 'weight-supported walking (CH1), 158 hand-supported walking (CH2)'

The legend has been updated accordingly (line 1159).

25. Table 1: For the children in the CH3 (early independent walking) group, please consider adding information regarding their age at walking onset. This detail is important for clearly determining each child's walking experience

Thank you for the suggestion. We agree that information about the age at walking onset would provide valuable insights into the walking experience of children in the CH3 group. Unfortunately, we were unable to collect this information systematically for all participants during the study. Instead, we relied on

developmental milestones, such as observable early independent walking behavior, to categorize participants into this group. While this limitation may affect the precision of walking experience estimates, we believe the grouping still provides meaningful insights into early walking development. We have noted this limitation in the revised manuscript (lines 628-630) and discussed its potential implications for the interpretation of our findings.

26. Information about participant recruitment procedures is missing. Please specify how participants were recruited (e.g., inclusion/exclusion criteria, recruitment setting, consent procedures), particularly for infants and toddlers. This information is important for assessing the generalizability and ethical conduct of the study.

Thank you for the suggestion. We have added this information in lines 666-669 and lines 682-686. Participants were recruited by word of mouth, with efforts made to ensure a diverse sample. The inclusion criteria required the participants to be healthy infants, toddlers, or children without any known neurological, musculoskeletal, or developmental disorders. The exclusion criteria required the subjects to not have a history of significant medical conditions that could affect motor development or the inability to complete the study protocol. For infants and toddlers, their parents or legal guardians provided written informed consent after receiving a detailed explanation of the study's purpose, procedures, and potential risks. The study was conducted in accordance with ethical guidelines and was approved by the Clinical Research Ethics Committee. These steps ensure compliance with ethical standards and help support the generalizability of the findings to typically developing children.

27. Figure 8B: there are multiple lines and statistical comparison that overlap with the axes and plotted data, making the figure somewhat difficult to interpret. The visual clarity is reduced, especially where lines intersect with bars or axis labels. Please consider revising the layout or presentation of the comparisons to enhance readability and improve figure clarity.

Thank you for the comment. We have improved the clarity of Fig. 8.

References:

1. Pizzolato, C. *et al.* CEINMS: A toolbox to investigate the influence of different neural control solutions on the prediction of muscle excitation and joint moments during dynamic motor tasks. *J. Biomech.* **48**, 3929–3936 (2015).
2. Hoang, H. X., Diamond, L. E., Lloyd, D. G. & Pizzolato, C. A calibrated EMG-informed neuromusculoskeletal model can appropriately account for muscle co-contraction in the estimation of hip joint contact forces in people with hip osteoarthritis. *J. Biomech.* **83**, 134–142 (2019).
3. Dominici, N. *et al.* Locomotor primitives in newborn babies and their development. *Science* **334**, 997–999 (2011).
4. Sylos-Labini, F. *et al.* Distinct locomotor precursors in newborn babies. *Proc.*

- Natl. Acad. Sci. U. S. A.* **117**, 9604–9612 (2020).
5. Zandvoort, C. S., Daffertshofer, A. & Dominici, N. Cortical contributions to locomotor primitives in toddlers and adults. *iScience* **25**, 105229 (2022).
 6. Silver, N. C. & Dunlap, W. P. Averaging correlation coefficients: should Fisher's z transformation be used? *J. Appl. Psychol.* **72**, 146 (1987).
 7. Rinaldi, L. *et al.* Adapting to the Mechanical Properties and Active Force of an Exoskeleton by Altering Muscle Synergies in Chronic Stroke Survivors. *IEEE Trans. Neural Syst. Rehabil. Eng.* **28**, 2203–2213 (2020).
 8. Cheung, V. C. K. *et al.* Plasticity of muscle synergies through fractionation and merging during development and training of human runners. *Nat. Commun.* **11**, 4356 (2020).
 9. Chvatal, S. A. & Ting, L. H. Common muscle synergies for balance and walking. *Front. Comput. Neurosci.* **7**, 48 (2013).
 10. Bach, M. M., Daffertshofer, A. & Dominici, N. Muscle synergies in children walking and running on a treadmill. *Front. Hum. Neurosci.* **15**, 637157 (2021).
 11. Guo, X. *et al.* Age-Related Modifications of Muscle Synergies and Their Temporal Activations for Overground Walking. *IEEE Trans. Neural Syst. Rehabil. Eng.* **30**, 2700–2709 (2022).
 12. Scano, A., Mira, R. M. & d'Avella, A. Mixed matrix factorization: a novel algorithm for the extraction of kinematic-muscular synergies. *J. Neurophysiol.* **127**, 529–547 (2022).

All line numbers referenced in this rebuttal refer to lines in the redline version of the revised manuscript (i.e., manuscript file with Track Changes).

Responses to Reviewer No. 1

I thank the authors for this thorough revision. I feel that all of my main concerns have been addressed, and I would like to emphasize the strong contribution this work makes to the literature. I only have one remaining issue regarding my very first point.

We sincerely appreciate the reviewer's positive acknowledgment.

I understand the methodological choice of not constraining cluster assignment by subject, as it allows for more representative groupings of similar MSs across the cohort. However, in this context, it remains unclear why the metric derived from this clustering (within-cluster SD) is often referred to as “between-subject variability” in the paper (e.g., 1226–253, 1311–337, Fig. 4, Fig. 7, etc). If multiple synergies from the same subject can fall within a single cluster, this metric seems to capture a mixture of intra- and inter-individual variability. For clarity, I would suggest either adopting terminology that more directly reflects the methodology (e.g., “within-cluster variability” or “between-KS variability”), or explicitly showing how subjects are distributed across clusters to justify the use of “between-subject variability”. In addition, I believe that the paper would be strengthened by explicitly discussing that regressions were performed on cluster-level variability, and therefore that whether KS, MS, and biomechanical parameters vary together at the individual level remains unknown.

Thank you for highlighting the potential ambiguity stemming from our use of the term “between-subject variability.” We agree that clarification is warranted. To more accurately reflect our analysis, we had revised the terminology to “within-cluster subject variability,” emphasizing variability among individuals within each cluster. Besides, we have confirmed that all relevant regression analyses are explicitly described in the manuscript as being conducted at the cluster level (see lines 318-328). As our clustering framework captures both inter-subject and intra-subject sources of variability, it does not by itself establish that KS, MS, and biomechanical parameters co-vary at the individual level. Nevertheless, it does identify the specific biomechanical features that are influential in the MS–KS regression, thereby providing a possible explanation for how bilateral KS symmetry increased during longitudinal gait development through the co-development of KS, MS, and biomechanical parameters at the individual level. We direct the reviewer to lines 496–504 and 532–535, as well as the first paragraph of the Discussion section “Dynamic Co-development of MS, KS, and Limb Biomechanical Properties,” where this is discussed in detail.

Responses to Reviewer No. 2

The revised manuscript largely addresses the concerns raised in the previous manuscript. In particular, the modifications to the comparison method in Figure 3 and the revisions

regarding fitness consistency in the musculoskeletal model have made the arguments more clearly supported.

We sincerely appreciate the reviewer's positive recognition.

However, concerns remain regarding the age-related changes in `OptimalFibreLength` in Figure 6B, which were also pointed out previously. It seems difficult to accept that optimal fiber length decreases despite increases in height and muscle length due to development. Since sarcomere increases with development, optimal fiber length should increase when considering the muscle alone. Admittedly, as the authors' response noted, pennation angle and tendons do influence this, making it difficult to definitively state that optimal fiber length always increases. Nevertheless, the result that optimal fiber length uniformly decreases with development is hard to believe.

One interpretation I considered for the property observed in Figure 6B is that body support during walking might have influenced it. When walking with body support, muscles may be in a stretched state during large muscle activities like foot contact or lift-off. Therefore, it could be interpreted that optimal fiber length was particularly longer during the early infant stage when body support is substantial. If this is the case, the results in Figure 6B would represent a property distinct from motor control strategies.

I am not entirely confident in the correctness of this interpretation. Furthermore, I do not believe the concerns regarding Figure 6B are significant enough to substantially lower the overall evaluation of the paper. However, I felt the results in Figure 6B require somewhat careful handling. Therefore, I would like to request one of the following from the authors:

1: If the results in Figure 6B are considered correct, I hope the authors can provide a more convincing explanation or cite literature demonstrating a decrease in optimal fiber length with development.

2: If it is difficult to support the results in Figure 6B, I believe it would be better to revise the text to treat this result as less significant results, such as by stating limitations regarding Figure 6B. (In that case, the claims regarding the properties using `OptimalFibreLength` in Figure 7 may also require some revision.)

Thank you for highlighting the surprising finding of the longitudinal decrease in `OptimalFibreLength` presented in Fig. 6B, and for providing constructive suggestions. In the revised manuscript, we interpreted this result cautiously as a potential implication rather than a definitive conclusion (lines 454–456). Also, when discussing Fig. 7, we intentionally did not emphasize the effects of `OptimalFibreLength`; instead, we focused on `TendonSlackLength` as the more influential parameter affecting the MS–KS relationship (lines 528–535).

We suspect that a decrease in the muscles' `OptimalFibreLength` with development is physiologically plausible in the following way. A previous study in neonatal rats reports that muscles of newborn rats contain primarily type I (slow-twitch) and type IIC muscle fibres¹, with the latter type being a primitive form of type II fibres that differentiates into type IIA and IIB fast-twitch fibres later in life. Thus, it is possible

that in humans, the proportions of slow- and true fast-twitch fibres change during development, so that when the muscles mature, there are relatively more of the latter type. Interestingly, another previous study finds that when compared with type I slow-twitch fibres, type II fast-twitch fibres could produce similar forces at shorter sarcomere lengths². Thus, the increasing proportion of true fast-twitch fibres in the muscles may lead to a decrease in the overall OptimalFibreLength of the muscles. Of course, further studies are required to confirm this conjectural mechanism. We have incorporated this explanation into the Discussion section (under “Limitations and Future Work,” lines 656–668).

1. Brooke, M. H., Williamson, E. & Kaiser, K. K. The Behavior of Four Fiber Types in Developing and Reinnervated Muscle. *Arch. Neurol.* **25**, 360–366 (1971).
2. Granzier, H. L., Akster, H. A. & Ter Keurs, H. E. Effect of thin filament length on the force-sarcomere length relation of skeletal muscle. *Am. J. Physiol.-Cell Physiol.* **260**, C1060–C1070 (1991).

Below are minor comments.

* In Table S1, only values where $P < 0.05$ are listed, with the rest left blank. Does this mean the relationship for the blank cells was not evaluated? If they were evaluated, it would be preferable to include all values.

Thank you for raising this point. We did assess the relationships for the blank cells; however, none were statistically significant ($p \geq 0.05$). Here, we have updated Table S1 to include all values and to highlight those formulae with $p < 0.05$.

* Line 1060: τ , Line 1062: α , β , γ , etc., should be italicized but are currently in roman type.

Thank you for noting the formatting issue. We have addressed this in the revised manuscript, see lines 1075-1079.

Responses to Reviewer No. 3

I thank the authors for their careful revision. They have addressed my comments thoroughly. The manuscript is now improved, and I have no further concerns.

We sincerely appreciate the reviewer’s supportive and encouraging assessment.

Reviewer Assessment

Manuscript#: COMMSBIO-25-0823

This manuscript offers a novel and valuable contribution to understanding developmental changes in motor control through the analysis of muscle and kinetic synergies. The manuscript investigates how muscle synergies evolve throughout human development, particularly comparing kinetic synergies (KS) and muscle synergies (MS) across different life stages, from infancy to adulthood. The authors claim that KS show higher variability and a more rapid rate of change in early stages compared to MS, which remain relatively stable and become sparser and more independently controlled in adulthood. This manuscript presents an interesting and valuable contribution to understanding motor control development. The study offers novel insights into the differential dynamics of kinetic and muscle synergies, which could influence future research in developmental neuroscience, motor control, and rehabilitation. In addition, the concept of comparing kinetic and muscle synergies across development is interesting and novel. The conclusions are generally supported by presented evidence. However, ensuring robust statistical correction and clearly articulating methodological choices will significantly enhance the manuscript's strength and reproducibility.

General comments:

1. Lines 590-594: The authors explicitly define four distinct developmental walking stages (CH1: weight-support; CH2: hand-support; CH3: early independent walking; CH4: later independent walking), each characterized by specific chronological and developmental criteria. However, Table 1 highlights critical inconsistencies regarding participant age across these stages: participant 25 has identical ages (0.93 years) reported for both CH1 (weight-support) and CH2 (hand-support). And participants 20, 24, and 27 also show identical ages for CH2 (hand-support walking) and CH3 (early independent walking). This seems to indicate that, at least in these cases, different walking conditions were recorded at the same developmental time rather than representing separate, chronologically distinct stages. Could you please clarify how children were distinguished and assigned to the different walking stages? If this is indeed the case—that is, if some children were assessed in multiple walking conditions simultaneously rather than distinct developmental stages—this should be clearly stated. Importantly, this distinction significantly impacts the manuscript's primary claim of investigating developmental trajectories across four clearly separated walking stages. Therefore, it is crucial that the authors explicitly address and clarify this inconsistency in detail. Specifically, they should explain clearly whether each participant indeed passed through four separate developmental stages, as described in the methods, or if instead they were assessed across multiple conditions within the same developmental stage. This clarification is essential for interpreting the study's findings and validating the conclusions drawn regarding developmental progression in walking stages.
2. Figure 2C and Table 1: Could the authors clarify precisely what is shown in Figure 2C? According to the legend, the X-axis represents the number of days elapsed between two developmental stages for each subject (CH1-CH2;CH2-CH3;CH3-CH4). However, Table 1 reports identical ages for several participants across different stages (e.g., participant 25 has the same age in CH1 and CH2, and participants 20, 24, and 27 have the same age in CH2 and CH3). If the difference in days between recordings was truly zero in these cases, how is it that the smallest value plotted in Figure 2C is 32 days? Were these cases excluded from the regression? Or is the variable on the X-axis

- representing something different than the actual time between recordings? Please clarify this apparent inconsistency between the figure and the subject data.
3. Given the high variability typically observed in walking patterns among infants and toddlers, the number of young participants included appears relatively low. Additionally, the number of participants included in both the adult and elderly groups (n=7 each) is also limited. Increasing participant numbers or providing clear justification for the sample sizes chosen would strengthen the reliability and generalizability of the findings.
 4. Please clearly state the exact criteria used for determining the number of clusters in the k-means clustering analyses (e.g., silhouette scores are mentioned, but explicit thresholds or procedures for selecting cluster number should be elaborated).
 5. **Statistical analysis:** The statistical approaches employed (e.g., Pearson similarity, regression analyses, and sparseness calculations) are appropriate. However: Clarify whether corrections for multiple comparisons were performed. If not, this should be addressed explicitly, considering the multiple statistical tests conducted across life stages.
 6. **Statistical analysis:** Provide more detailed descriptions or justifications for chosen statistical tests in the main manuscript text.
 7. **Interpretation of Results:** The interpretation regarding the stability of MS vs. variability of KS is compelling but could benefit from additional evidence or discussion: are there potential confounding factors (e.g., activity level differences, walking speeds, or environmental influences) that might impact the variability observed in KS?
 8. Could the authors comment further on why kinetic synergies might exhibit higher variability relative to muscle synergies?
 9. **Figures and Supplementary Material:** The supplementary figures provided robust additional support for main findings. However, it would be beneficial if some key supplementary results (e.g., Fig. S1, S3, and S5 regarding synergy similarity and rate of change) could be integrated or briefly discussed in the main text to strengthen the core narrative of developmental trajectories.
 10. The authors might consider explicitly stating potential future research directions or implications of their findings for clinical applications or theoretical models of motor development.

Minor / Specific Points:

11. Lines 590-591: The authors mention that during the CH1 (weight-support walking) condition, "children walked with weight support from their parents lifting their body from the shoulder." Could the authors specify more clearly the exact method used for providing support? Specifically, were parents instructed to hold the infants around the waist, under the shoulders, or elsewhere, and what specific instructions were given to parents regarding the amount or proportion of body weight to support? Several studies have demonstrated significant effects of the amount of body-weight support on kinematic patterns and muscle synergies, both in toddlers and adults (Pang and Yang, 2000; Dominici et al., 2007; Kerkman et al., 2022; see review by Apte et al., 2018). It is crucial to clearly describe and address this methodological detail for correct interpretation of the data and comparison with previous literature.
12. Data Recording: The manuscript does not specify important details regarding the data collection protocol. In particular, it is unclear how many trials were recorded per participant, what was the size of the recording volume, and how many strides or gait

cycles were collected in total. Providing this information is essential to assess the robustness and representativeness of the dataset. Please include these details to improve transparency and reproducibility.

13. Line 608: For transparency and reproducibility, please explicitly specify the anatomical locations where markers were placed on the participants
14. Lines 615-616: Could the authors clarify the rationale for recording muscle activity from the external oblique and latissimus dorsi muscles, given that the primary focus is on lower-body walking patterns? Particularly in the CH1 (weight-support walking) condition, there might be significant muscle-activation artifacts introduced by the researcher's support. Please discuss how this potential issue was addressed or justify the selection of these muscles.
15. Line 616 and line 630: The reference to Fig. 2A in these lines is unclear, as Fig. 2A does not explicitly illustrate the EMG electrode placement or muscle selection, nor does it clearly depict the joint or 3D angles mentioned. Please revise the figure reference or provide an appropriate figure that visually clarifies the EMG electrode placements and joint angles analyzed.
16. Line 641: The selected cutoff frequency (50 Hz) for the high-pass filter applied to EMG data appears unusually high compared to standard practices. Could the authors clarify or justify explicitly why such a high cutoff frequency was selected?
17. Line 644-645: The description of applying the Savitzky-Golay filter to remove occasional high-amplitude noisy spikes from the EMG signal is not entirely clear. Could the authors briefly clarify the steps implemented for this process? Specifically, how were high-amplitude spikes identified—was there a threshold or criterion used? Providing a concise but clear explanation of these preprocessing steps would greatly assist readers in understanding and potentially reproducing the methodology.
18. Line 649: The authors apply a fixed electromechanical delay of 50 ms to shift EMG signals. However, this value is based on adult data and may not be appropriate for infants and toddlers, who usually show longer electromechanical delay due to their immature neuromuscular systems. Please clarify why a constant 50 ms delay was used for all age groups, and consider discussing whether age-specific electromechanical delay values would be more accurate.
19. Line 705: Please clarify how many gait cycles were ultimately included per participant in the concatenated matrix used to extract KMS.
20. Figure 1 (section: longitudinal - learn to walk): The image used to represent the 'supported walking' condition could be improved by showing a toddler clearly supported by an adult (e.g., holding hands while walking). This would help distinguish more clearly between supported and independent walking, improving the visual clarity of the developmental stages depicted. Additionally, the figure and data collection section mention 'BodyWeight'—could the authors clarify what specific body weight information was collected and how? This detail does not appear to be reported elsewhere in the manuscript.
21. Figure 2A is very small and difficult to evaluate clearly. Please consider enlarging the figure or improving its resolution to ensure that all elements—especially EMG profiles and joint angles—are clearly visible and interpretable by the reader. Additionally, while Figure 2A illustrates an example from a single infant, it appears that a larger number of strides were recorded for CH1 compared to CH4. It would be helpful if the authors could report the average number of strides (and/or the range) used for the analysis at each developmental stage, as well as for adults and elderly participants. This information is important to assess the representativeness and balance of the dataset across conditions.

22. Lines 184–188 / Figure 2B and Fig. S1A: The manuscript reports statistical comparisons of between-stage synergy similarity values (e.g., KS vs. MS) across developmental transitions, with associated p-values (e.g., $p = 0.039$, $p = 0.023$). However, the specific statistical test used to assess these differences is not stated. Since the comparisons appear to be based on within-subject, paired data (i.e., comparing KS and MS similarity for the same subjects), it is important to clarify whether a paired t-test, Wilcoxon signed-rank test, or another method was used. Additionally, since the similarity values are Pearson correlation coefficients, it is worth noting that these are not normally distributed. A Wilcoxon signed-rank test would be more appropriate unless the authors applied a Fisher z-transformation before performing a parametric test. Please clarify the statistical approach used and consider whether a non-parametric alternative might be more suitable. More broadly, the analysis appears to involve repeated measurements across both synergy type (KS, MS, KMS) and developmental stage transitions (CH1→CH2, CH2→CH3, CH3→CH4) for the same subjects. This suggests that a two-way repeated measures ANOVA or a linear mixed-effects model may be more appropriate to account for within-subject structure and potential interactions between factors. Could the authors clarify the overall statistical strategy used for these comparisons? Were pairwise tests applied independently, or was a multi-factorial model considered? Also, were any corrections for multiple comparisons performed? Explicitly outlining the statistical approach would improve the transparency and robustness of the analysis.
23. Line 363 Discussion : The discussion could be enriched by further exploring possible interpretations of the observed differences between kinetic (KS) and muscle synergies (MS) across development. While the authors describe KS as more variable and changing more rapidly than MS, it may be helpful to consider how these patterns could relate to underlying neural or biomechanical mechanisms. For instance, could KS be more influenced by body dynamics and postural demands, whereas MS might reflect more stable, spinally organized motor modules? Expanding on these possibilities—even as hypotheses—could help contextualize the findings within broader models of motor development.
24. Lines:1013-1014 Figure 1 legend: should be ‘weight-supported walking (CH1), 158 hand-supported walking (CH2)’
25. Table 1: For the children in the CH3 (early independent walking) group, please consider adding information regarding their age at walking onset. This detail is important for clearly determining each child's walking experience
26. Information about participant recruitment procedures is missing. Please specify how participants were recruited (e.g., inclusion/exclusion criteria, recruitment setting, consent procedures), particularly for infants and toddlers. This information is important for assessing the generalizability and ethical conduct of the study.
27. Figure 8B: there are multiple lines and statistical comparison that overlap with the axes and plotted data, making the figure somewhat difficult to interpret. The visual clarity is reduced, especially where lines intersect with bars or axis labels. Please consider revising the layout or presentation of the comparisons to enhance readability and improve figure clarity.